# Nasopharyngeal lymphatic plexus is a hub for cerebrospinal fluid drainage

Jin-Hui Yoon[1,7], Hokyung Jin[1,2,7], Hae Jin Kim[3,7], Seon Pyo Hong[1], Myung Jin Yang[1,2], Ji Hoon Ahn[1], Young-Chan Kim[1,2], Jincheol Seo[4], Yongjeon Lee[4], Donald M. McDonald[5,6], Michael J. Davis[3,8 ✉] & Gou Young Koh[1,2,8 ✉]

Cerebrospinal fluid (CSF) in the subarachnoid space around the brain has long been known to drain through the lymphatics to cervical lymph nodes[1–17], but the connections and regulation have been challenging to identify. Here, using fluorescent CSF tracers in *Prox1-GFP* lymphatic reporter mice[18], we found that the nasopharyngeal lymphatic plexus is a major hub for CSF outflow to deep cervical lymph nodes. This plexus had unusual valves and short lymphangions but no smooth-muscle coverage, whereas downstream deep cervical lymphatics had typical semilunar valves, long lymphangions and smooth muscle coverage that transported CSF to the deep cervical lymph nodes. α-Adrenergic and nitric oxide signalling in the smooth muscle cells regulated CSF drainage through the transport properties of deep cervical lymphatics. During ageing, the nasopharyngeal lymphatic plexus atrophied, but deep cervical lymphatics were not similarly altered, and CSF outflow could still be increased by adrenergic or nitric oxide signalling. Single-cell analysis of gene expression in lymphatic endothelial cells of the nasopharyngeal plexus of aged mice revealed increased type I interferon signalling and other inflammatory cytokines. The importance of evidence for the nasopharyngeal lymphatic plexus functioning as a CSF outflow hub is highlighted by its regression during ageing. Yet, the ageing-resistant pharmacological activation of deep cervical lymphatic transport towards lymph nodes can still increase CSF outflow, offering an approach for augmenting CSF clearance in age-related neurological conditions in which greater efflux would be beneficial.

CSF is essential for mechanical protection, nourishment and clearing of neurotransmitters, metabolites and protein aggregates such as amyloid-β and tau from the central nervous system[17,19]. CSF is continuously secreted by the choroid plexus into the cerebral ventricles and subarachnoid space, circulates within and around the brain and spinal cord, and turns over 3 to 5 times per day[19]. The regulation of CSF production, circulation and drainage is receiving increasing attention due to evidence that reduced CSF secretion and/or clearance during ageing could contribute to the development of Alzheimer's disease and other neurodegenerative conditions[20–23].

Among the known routes for CSF drainage from the subarachnoid space are lymphatics in the cribriform plate, where olfactory nerves pass through the ethmoid bone, and in the perineurium of cranial nerves[8,17,24]. Lymphatics in the dura mater also function as a CSF drainage route[9–17,25]. However, despite solid documentation of the contribution of lymphatics to CSF clearance, the connections between the subarachnoid space and extracranial lymphatics involved in CSF clearance have been challenging to elucidate.

A landmark study[1] provided evidence for CSF outflow to the nasopharynx by finding Richardson's blue dye in the lymphatics of the nasal and palatal mucosa after injection of the tracer into the subarachnoid space[1]. Other studies[4–7] revealed that dye or Microfil silicone rubber injected into the subarachnoid space subsequently appears in the nasal lymphatics and in the perineurium of cranial nerves. A previous study[12] reported that CSF can drain through lymphatics of the pharynx. A magnetic resonance imaging (MRI) study[26] provided further evidence of CSF outflow through the cribriform plate to the nasopharynx. A recent study[27] provided additional observations on the connections between the subarachnoid space and lymphatic vessels around olfactory nerves that cross the cribriform plate en route to the nasal mucosa. Another study[16] observed connections between intracranial lymphatics and extracranial lymphatics using light-sheet fluorescence microscopy after injection of ovalbumin–Alexa Fluor 555 (OVA–Alexa Fluor 555) conjugate into CSF or by MRI after systemic injection of gadolinium-based contrast agent. However, tracing OVA–Alexa Fluor 555 in fixed tissues was confounded by phagocytosis of the tracer by macrophages. Our

[1]Center for Vascular Research, Institute for Basic Science, Daejeon, Republic of Korea. [2]Graduate School of Medical Science and Engineering, Korea Advanced Institute of Science and Technology (KAIST), Daejeon, Republic of Korea. [3]Department of Medical Pharmacology and Physiology, University of Missouri, Columbia, MO, USA. [4]National Primates Research Center, Korea Research Institute of Bioscience and Biotechnology, Cheongju, Republic of Korea. [5]Department of Anatomy, Cardiovascular Research Institute, University of California, San Francisco, San Francisco, CA, USA. [6]Helen Diller Family Comprehensive Cancer Center, University of California, San Francisco, San Francisco, CA, USA. [7]These authors contributed equally: Jin-Hui Yoon, Hokyung Jin, Hae Jin Kim. [8]These authors jointly supervised this work: Michael J. Davis, Gou Young Koh. ✉e-mail: davismj@health.missouri.edu; gykoh@kaist.ac.kr

own MRI study revealed a strong signal at the skull base from a contrast agent in the CSF, but connections to extracranial lymphatics could not be identified owing to the limited resolution[15]. Evidence from these and other studies suggests that CSF can be drained through nasopharyngeal lymphatics, but their connections and functional properties relating to CSF outflow under normal conditions and during ageing have proven to be difficult to characterize.

To solve this problem, we performed fluorescence microscopy imaging of CSF outflow in anaesthetized prospero-related homeobox 1–green fluorescence protein (*Prox1-GFP*) reporter mice[18] after surgical exposure of the nasopharyngeal and other cervical lymphatics. This approach revealed a distinctive lymphatic plexus in the nasopharynx that functioned as a hub for CSF outflow through lymphatics from the cribriform plate and select other intracranial regions en route to deep cervical lymph nodes (dcLNs). This nasopharyngeal lymphatic plexus (NPLP) regressed and underwent transcriptomic changes with ageing. By contrast, deep cervical lymphatics, which were covered by smooth muscle and carried CSF from the plexus to lymph nodes, did not change during ageing. The contractile properties of these lymphatics were regulated by α-adrenergic and nitric oxide (NO) signalling, and CSF outflow could still be increased by pharmacological activation of CSF transport in aged mice.

## Distinctive features of the NPLP

The lymphatics in the mucosa of the nasopharynx formed a distinctive NPLP that was conspicuous in *Prox1-GFP* lymphatic reporter mice[18] after staining for LYVE1 and VEGFR3 (Fig. 1, Extended Data Fig. 1a,b and Supplementary Fig. 1a,b). The rostral end of the lymphatic plexus was connected to the flattened, condensed, highly anastomotic posterior nasal lymphatic plexus that also stained positively for PROX1–GFP and LYVE1 but stained weakly with VEGFR3 (Fig. 1a, Extended Data Fig. 1a,b and Supplementary Fig. 1a). The two lymphatic plexuses covered all surfaces of the posterior nasal cavity and nasopharynx except near the skull base (Fig. 1a, Extended Data Fig. 1a,b and Supplementary Fig. 1b).

The NPLP had 45–65 irregular, linearly shaped valves that stained positive for PROX1–GFP and laminin-α5, but no smooth-muscle coverage was evident after staining for α-smooth muscle actin (αSMA; Fig. 1a and Supplementary Fig. 1c,d). The segments between valves (lymphangions) were unusually short (Fig. 1a and Supplementary Fig. 1a–d). The plexus resembled an inverted saddle when viewed in three dimensions (Fig. 1b and Supplementary Video 1). Owing to these features, the lymphatics of the plexus were unlike individual initial lymphatics in most peripheral organs or collecting lymphatics that have semilunar valves, long lymphangions and smooth muscle coverage (Supplementary Fig. 1c). A similar lymphatic plexus was found in the nasopharynx of the primate, *Macaca fascicularis* (Extended Data Fig. 2a,b), consistent with conservation of these features among species. However, in *M. fascicularis*, the lymphatics had typical semilunar valves (Extended Data Fig. 2a,b), in contrast to those in mice.

## The NPLP as a route for CSF outflow

To determine whether the NPLP was a route for CSF outflow, we infused 3 µl of 10 kDa tetramethylrhodamine-conjugated dextran (TMR–dextran) in PBS at 1 µl min$^{-1}$ for 3 min into the subarachnoid space at the cisterna magna of anaesthetized *Prox1-GFP* mice. At 60 min, TMR–dextran fluorescence traced using a fluorescence stereomicroscope was concentrated in the olfactory bulb, cribriform plate, anterior cranial fossa (above the presphenoid bone) and middle cranial fossa (above the basisphenoid bone) adjacent to the nasopharynx (Fig. 2a,b). TMR–dextran was also visible in the CSF over the dorsal surface of the cerebellum and around the cervical spinal cord (Fig. 2b).

These findings led us to learn more about nasopharyngeal lymphatics as a CSF outflow route. At 30 min after infusion into the cisterna magna,

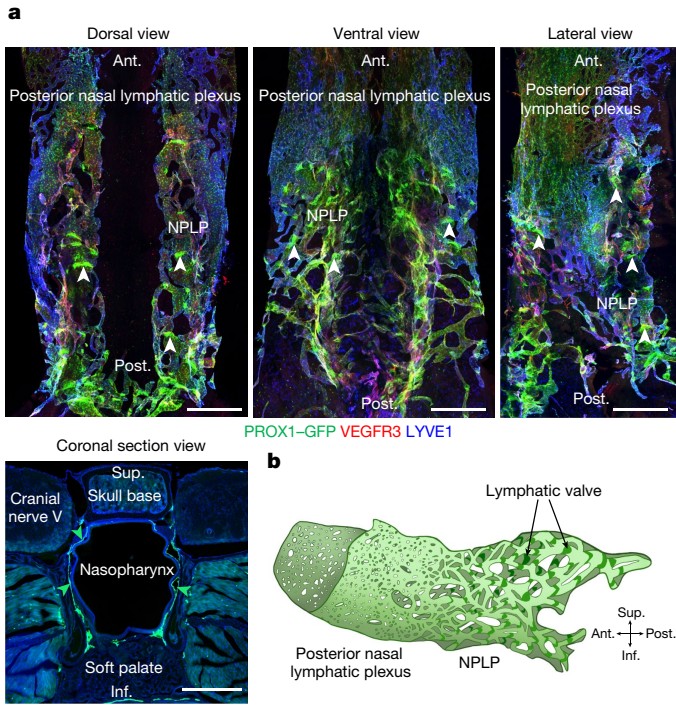

**Fig. 1 | Three-dimensional morphological features of the nasopharyngeal and posterior nasal lymphatic plexuses. a**, Immunofluorescence images of three views of whole mounts and a coronal section of the NPLP and posterior nasal lymphatic plexus of *Prox1-GFP* mice after staining for VEGFR3 and LYVE1. The flattened and condensed posterior nasal lymphatic plexus is in front of the NPLP distinguished by strong PROX1⁺, irregular and linearly shaped lymphatic valves (white arrowheads). The green arrowheads in the cross-section mark the borders of the NPLP. Scale bars, 500 µm. Similar findings were obtained from *n* = 6 mice in three independent experiments. **b**, Diagram of the inverted saddle shape of the NPLP. Anatomical positions are indicated at the bottom right. Ant., anterior; post., posterior; sup., superior; inf., inferior anatomical position.

TMR–dextran was detected in the lymphatics of the nasopharynx, deep cervical lymphatics and dcLNs, but not in the lymphatics of the oropharynx or soft palate (Fig. 2c–f, Supplementary Fig. 2a–c and Supplementary Video 2). TMR–dextran was also detected in the superficial cervical lymph nodes at 30 min (Supplementary Fig. 3a–c), but these nodes were not downstream to the NPLP and deep cervical lymphatics. This indicates that the CSF outflow that reached the superficial lymph nodes did not flow through the nasopharyngeal lymphatics.

The lymphatics of the nasopharynx had unusual PROX1–GFP⁺ valves, as described above (Fig. 2d,e, Supplementary Fig. 1a,c and Supplementary Video 2). Although PROX1–GFP⁺ lymphatics were abundant in the oropharynx, no TMR–dextran was detected there until 120 min after the intracisternal infusion (Fig. 2e,f and Supplementary Video 3). As evidence that the findings were not specific to 10 kDa TMR–dextran, we obtained similar results after intracisternal infusion of 70 kDa TMR–dextran, Texas-Red–ovalbumin or 0.5 µm FluoSpheres (Supplementary Fig. 4a–d). These findings are further evidence that CSF drains through the NPLP but not through the oropharyngeal lymphatic plexus.

## Morphological features of the NPLP

Initial lymphatics, also called lymphatic capillaries, which are specialized for the uptake of fluid and macromolecules, have button-like intercellular junctions and lack smooth-muscle-cell coverage[28,29]. By contrast, collecting lymphatics, which propel lymph downstream

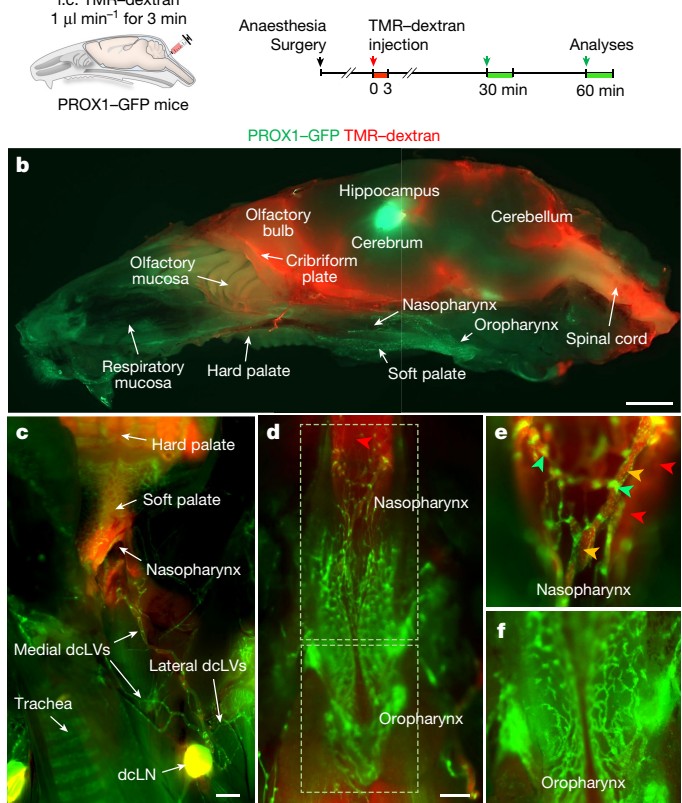

**a** i.c. TMR–dextran
1 µl min⁻¹ for 3 min

PROX1–GFP mice

Anaesthesia
Surgery / TMR–dextran injection / Analyses
0 3    30 min    60 min

PROX1–GFP TMR–dextran

**b** Hippocampus, Olfactory bulb, Cerebrum, Cerebellum, Cribriform plate, Olfactory mucosa, Nasopharynx, Oropharynx, Spinal cord, Respiratory mucosa, Hard palate, Soft palate

**c** Hard palate, Soft palate, Nasopharynx, Medial dcLVs, Lateral dcLVs, Trachea, dcLN

**d** Nasopharynx, Oropharynx

**e** Nasopharynx

**f** Oropharynx

PROX1–GFP TMR–dextran

**Fig. 2 | Preferential and selective distribution of TMR–dextran in the head and neck after intracisternal infusion. a**, Diagram of the experimental sequence for intracisternal (i.c.) infusion of TMR–dextran (molecular mass, 10 kDa) into *Prox1-GFP* mice through the cisterna magna at 1.0 µl min⁻¹ for 3 min followed by analysis of the distribution of TMR–dextran in the head and neck 30 or 60 min later. **b**, Fluorescence image showing the distribution of TMR–dextran in a mid-sagittal view of half of the head and neck at 60 min after intracisternal infusion. The PROX1–GFP signal is strong in the hippocampus and in the lymphatics in the nasopharynx, oropharynx and palate. Scale bar, 2 mm. Similar findings were obtained from *n* = 6 mice in three independent experiments. **c–f**, Fluorescence images showing the distributions of TMR–dextran in the indicated regions of dissected neck at 30 min after intracisternal infusion. TMR–dextran fluorescence (red) is strong in medial dcLVs, dcLNs (**c**) and lymphatic plexus in the nasopharynx (yellow arrowheads) (**d,e**) but not in the oropharynx (**f**). Strong PROX1⁺ lymphatic valves are indicated by green arrowheads. The red arrowheads indicate the background signal emitted from the skull base. Scale bars, 1 mm (**c**) and 500 µm (**d**). Similar findings were obtained from *n* = 10 mice in five independent experiments.

by undergoing rhythmic contractions, have zipper-like junctions, smooth-muscle coverage and valves to prevent retrograde flow[28–30]. Precollecting lymphatics have features of both types of lymphatics, in that they lack smooth-muscle coverage but have valves and a mixture of button-like and zipper-like junctions[28–30]. Some lymphatics in the nasopharyngeal plexus had little or no LYVE1 immunoreactivity, whereas others had strong LYVE1 staining. Those with weak LYVE1 staining had zipper-like junctions (Extended Data Fig. 3a,b), which fits more with transport than fluid entry. By contrast, the lymphatics in the plexus with strong LYVE1 staining had a mixture of intercellular junctions: zipper-like (47%), mixed (32%) and button-like (21%) (Extended Data Fig. 3a,b), which are features that are consistent more with fluid uptake than transport. On the basis of the mixture of button-like and zipper junctions, variable LYVE1 staining and the absence of smooth muscle, the lymphatics of the nasopharyngeal plexus had features of

both lymphatic capillaries and collecting lymphatics and, therefore, fit the description of precollecting lymphatics.

## Connections of the NPLP

We next sought to find the upstream connections of the NPLP in *Prox1-GFP* mice by tracing fluorescent 0.5 µm beads (FluoSpheres) using confocal fluorescence microscopy after infusion into the subarachnoid space (Methods). As the FluoSpheres remained in place during tissue processing, this approach enabled the identification of lymphatics near the pituitary that extended along cranial nerve V and the cavernous sinus to the NPLP (Extended Data Fig. 4a–c and Supplementary Video 4). The lymphatics contained FluoSpheres (Extended Data Fig. 4d), consistent with the connection to the subarachnoid space. Although some lymphatics in optical sections appeared to be discontinuous (Extended Data Fig. 4d), examination of confocal image stacks confirmed their continuity and also the intralymphatic location of the beads. FluoSphere-containing lymphatics along the pterygopalatine artery (PPA) also connected to the lymphatic plexus through the posterior nasal lymphatic plexus (Extended Data Fig. 5a–c and Supplementary Video 5).

In *Prox1-GFP* mice, LYVE1⁺ lymphatics in the dura near the olfactory bulb crossed the cribriform plate with sensory axons of olfactory nerves and then connected to the LYVE1⁻ lymphatic plexus in the olfactory mucosa and joined the posterior nasal lymphatic plexus (Extended Data Fig. 6a–d and Supplementary Video 6), as described in a recent report[27]. The lymphatics in these regions were found to contain FluoSpheres at 30 min after intracisternal injection into the CSF (Extended Data Fig. 6c,d), but the anatomical connections of individual lymphatics to the subarachnoid space could not be discerned. Thus, at least two groups of lymphatics appeared to be responsible for CSF outflow from the middle cranial fossa, including the pituitary and cavernous sinus, while the third carried CSF from the cribriform plate and other regions of the anterior cranial fossa (Fig. 3a).

Lymphatics downstream to the nasopharyngeal plexus included four lymphatic branches (upper right and left, and lower right and left) that were visible after removing the soft palate and other tissues around the nasopharynx in *Prox1-GFP* mice (Extended Data Fig. 7a). The upper and lower lymphatic branches on the right or left merged at the caudal end of the eustachian tube and extended to the right or left medial deep cervical lymphatic vessels (dcLVs; medial cervical lymphatics) (Extended Data Fig. 7a,b). These findings provide evidence that CSF drains through lymphatics that traverse the anterior and middle cranial fossae including the cribriform plate, join the NPLP and pass downstream through the medial cervical lymphatics en route to dcLNs (cervical lymph nodes) (Fig. 3a).

## VEGF-C-stimulated expansion of the NPLP

To determine whether the nasopharyngeal plexus can be expanded, mouse vascular endothelial growth factor-C (VEGF-C) was overexpressed in *Prox1-GFP* mice by intracisternal delivery of 3 × 10¹⁰ gene copies of adeno-associated virus serotype 9 encoding mouse VEGF-C-mCherry (AAV9-mVEGF-C-mCherry) or control AAV9-mCherry (Extended Data Fig. 8a). Then, 3 weeks after viral delivery, histological examination of mCherry fluorescence revealed mVEGF-C expression around the plexus and dural lymphatics but not around the diaphragmatic lymphatics (Extended Data Fig. 8b). The lymphatics in the plexus and dura in the AAV9-mVEGF-C-mCherry group were expanded 1.7- and 15.4-fold, respectively, compared with the AAV9-mCherry control group, but no change was found in the diaphragm (Extended Data Fig. 8b,c).

At 30 min after intracisternal infusion, TMR–dextran in the dcLNs of the AAV9-mVEGF-C-mCherry group was 3.5-fold higher than in the control group (Extended Data Fig. 8d). These findings are evidence that the

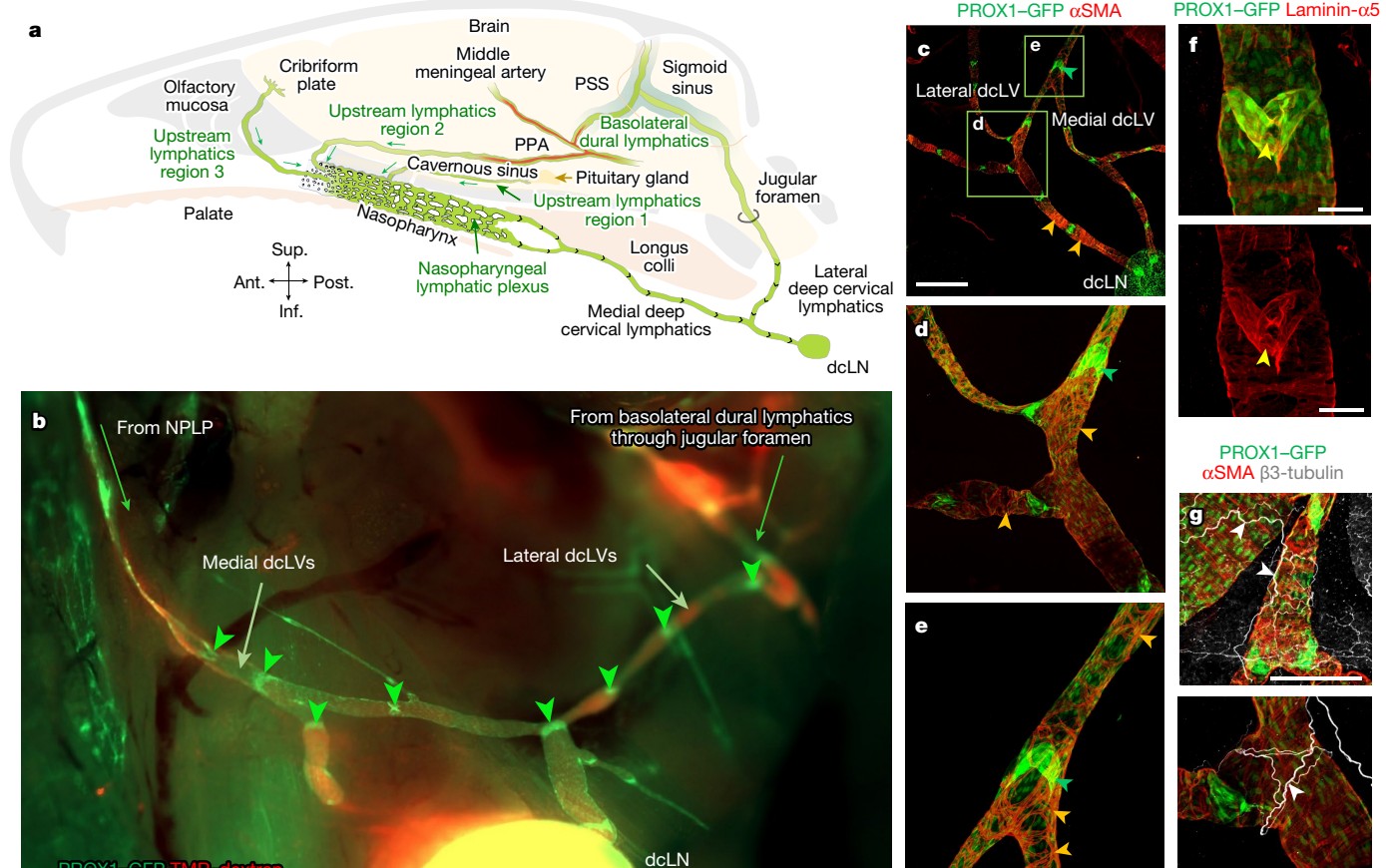

**Fig. 3 | Connections of the NPLP and features of deep cervical lymphatics.**
**a**, Diagram of intracranial upstream lymphatic regions 1, 2 and 3, which drain through the NPLP en route to medial deep cervical lymphatics and dcLNs in the neck. Upstream lymphatic region 1 includes the lymphatics near the pituitary gland and cavernous sinus that drain to the NPLP. Upstream lymphatic region 2 includes the lymphatics in the anterior region of basolateral dura near the middle meningeal artery and petrosquamosal sinus (PSS) that course along the PPA to the NPLP. Upstream lymphatic region 3 includes lymphatics near the cribriform plate that drain to the lymphatics in the olfactory mucosa en route to the posterior nasal lymphatic plexus and NPLP. By contrast, the lymphatics in the posterior region of the basolateral dura around the sigmoid sinus do not drain to the NPLP but, instead, pass through the jugular foramen to lateral deep cervical lymphatics en route to dcLNs. Anatomical positions are indicated at the bottom left. **b**, Fluorescence image showing medial dcLVs, lateral dcLVs, lymphatic valves (green arrowheads) and TMR–dextran (red) in lymphatics deep in the neck of a *Prox1-GFP* mouse. The image was obtained 30 min after i.c. infusion of TMR–dextran (molecular mass, 10 kDa) at 1.0 µl min⁻¹ for 3 min.

Medial dcLVs connect to the NPLP, and lateral dcLVs connect to the basolateral dural lymphatics through the jugular foramen. Scale bar, 1 mm. Similar findings were obtained from $n$ = 6 mice in three independent experiments. **c–e**, Immunofluorescence images of whole mounts showing the distributions of PROX1-dense, semi-lunar shaped lymphatic valves (green arrowheads) and αSMA⁺ circular smooth muscle cells (SMCs, orange arrowheads) in the medial and lateral dcLVs. **d,e**, Magnified images of the regions indicated by the green boxes in **c**. Scale bar, 1 mm (**c**). Similar findings were obtained from $n$ = 4 mice in two independent experiments. **f**, Immunofluorescence images of whole mounts showing a typical semi-lunar-shaped PROX1-dense, laminin-α5high valve (yellow arrowheads) in a medial dcLV of a *Prox1-GFP* mouse. Scale bars, 200 µm. Similar findings were obtained from $n$ = 4 mice in two independent experiments. **g**, Immunofluorescence images of whole mounts showing the distributions of β3-tubulin⁺ axons (white arrowheads) and αSMA⁺ circular smooth-muscle cells (red) along dcLVs. Scale bars, 200 µm. Similar findings were obtained from $n$ = 4 mice in two independent experiments.

NPLP can be expanded by overexpression of VEGF-C and, importantly, provide proof of concept for a strategy for increasing CSF outflow by expansion of the NPLP.

## Connection of the NPLP to the dcLNs

The nasopharyngeal plexus in *Prox1-GFP* mice was connected downstream to medial cervical lymphatics that drained to the dcLNs (Fig. 3b, Extended Data Fig. 7a,b and Supplementary Fig. 5). Medial cervical lymphatics had conventional semilunar valves that were spaced 250–750 µm apart throughout their length to lymph nodes (Fig. 3c–f and Extended Data Fig. 7b). These lymphatics had a dense but uneven layer of circular αSMA⁺ smooth-muscle cells along their entire length (Fig. 3c–e and Extended Data Fig. 7b) and were accompanied by adrenergic axons, identified by staining for tyrosine phosphatase and β3-tubulin, but not by cholinergic axons, identified by staining for

vesicular acetylcholine transporter (VAChT) and β3-tubulin (Fig. 3g and Supplementary Fig. 6a,b).

Other lymphatics, positioned laterally in the neck (lateral cervical lymphatics, lateral dcLVs), extended from the basolateral dura through the jugular foramen to dcLNs (Fig. 3b and Supplementary Fig. 7a–f). Like medial cervical lymphatics, lateral cervical lymphatics had semi-lunar valves and smooth-muscle coverage (Fig. 3b–d). These deep cervical lymphatics have contractile lymphangions (lymphangion pumps) that can potentially propel lymph flow to lymph nodes by undergoing spontaneous, cyclical contractions[31]. Consistent with their different upstream connections, medial cervical lymphatics appeared to transport CSF from the nasopharyngeal plexus en route to dcLNs, whereas lateral cervical lymphatics were routes for CSF drainage from the basolateral dural lymphatics to dcLNs (Fig. 3a).

To compare the amounts of CSF drainage through the medial and lateral cervical lymphatics, we measured TMR–dextran fluorescence

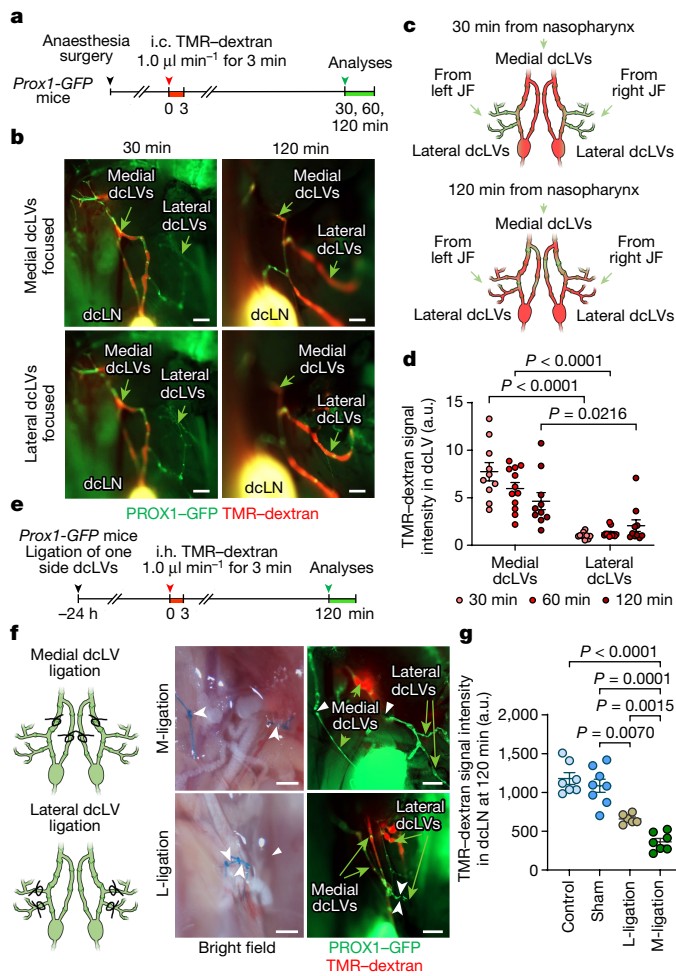

**Fig. 4 | Greater and faster drainage of TMR–dextran in CSF through the medial deep cervical lymphatics compared with the lateral deep cervical lymphatics. a**, Diagram of the experimental sequence for intracisternal infusion of TMR–dextran at 1.0 µl min⁻¹ for 3 min into *Prox1-GFP* mice followed by measurement of the fluorescence intensity in the medial and lateral dcLVs. **b–d**, Fluorescence images (**b**), diagrams (**c**) and comparison of TMR–dextran signal intensity (**d**) in the medial and lateral dcLVs at 30 min ($n = 10$), 60 min ($n = 12$) and 120 min ($n = 10$) after intracisternal infusion from five independent experiments. TMR–dextran fluorescence is stronger in medial compared with in lateral dcLVs (green arrows) from 30 to 120 min. For **b**, scale bars, 500 µm. Data are mean ± s.e.m. *P* values were calculated using two-way analysis of variance (ANOVA) followed by two-tailed Sidak's post hoc test. JF, jugular foramen. **e**, Diagram of the experimental sequence for ligation of medial dcLVs or lateral dcLVs followed 24 h later by intrahippocampal (i.h.) infusion of TMR–dextran at 0.1 µl min⁻¹ for 3 min into *Prox1-GFP* mice. The fluorescence intensity in the dcLN was measured 120 min after onset of i.h. infusion. **f,g**, Diagram, and bright-field and fluorescence images of the ligation sites of medial (M-ligation) and lateral (L-ligation) dcLVs (black and white arrowheads) (**f**) and measurements of TMR–dextran in **f** and measurements of TMR–dextran in dcLNs, with or without ligation, at 120 min after infusion (**g**). TMR–dextran accumulated in dcLVs proximal, but not distal, to the ligation. For **f**, scale bars, 500 µm. Comparison of TMR–dextran fluorescence in the dcLN after ligation of medial or lateral dcLVs. Each dot is the value for one mouse. $n = 7$ (control), $n = 8$ (sham), $n = 6$ (L-ligation) and $n = 7$ (M-ligation) from five independent experiments. Data are mean ± s.e.m. *P* values were calculated using one-way ANOVA followed by two-tailed Dunnett's T3 multiple-comparison post hoc test.

after intracisternal infusion of TMR–dextran (1.0 µl min⁻¹ of TMR–dextran for 3 min) into *Prox1-GFP* mice (Fig. 4a). At 30, 60 and 120 min after infusion, TMR–dextran fluorescence in medial cervical lymphatics was 7.1-fold, 5.2-fold and 3.4-fold the amount in lateral cervical

lymphatics (Fig. 4b–d). TMR–dextran in the anterior and middle intracranial regions that drained through medial cervical lymphatics was 2.6-fold higher than the amount of TMR–dextran in the posterior region that drained through lateral cervical lymphatics at the skull base (Supplementary Fig. 8). The findings provided evidence of drainage of CSF in the anterior and central regions of the skull base through perineural lymphatics that exit the skull en route to the NPLP, medial cervical lymphatics and dcLNs, before entering the systemic blood circulation.

We next measured TMR–dextran accumulation in the dcLNs 24 h after ligation of medial or lateral cervical lymphatics (Fig. 4e,f). In these experiments, TMR–dextran was infused into the hippocampus (0.1 µl min⁻¹ of TMR–dextran for 3 min) rather than the cisterna magna, because the accumulation of TMR–dextran in the dcLNs over 120 min was slower and less variable under the baseline conditions (Supplementary Fig. 9a–c). Compared with the sham-operated controls, ligation of medial cervical lymphatics was followed by a 70–80% decrease in TMR–dextran in lymph nodes at 120 min after infusion, whereas the reduction was significantly less (40–50%) after ligation of the lateral cervical lymphatics (Fig. 4g). This difference is evidence that CSF drainage through the medial deep cervical lymphatics to the cervical lymph nodes was on average 180% greater compared with CSF drainage through the lateral route. We therefore examined whether the CSF outflow to deep cervical lymphatics could be increased by activation of rhythmic contraction and relaxation of the lymphatics.

## Control of CSF outflow through deep cervical lymphatics

As the next step, we examined whether CSF outflow could be regulated by pharmacological activation of medial cervical lymphatics. We addressed this question by topical application of the α1-adrenergic agonist phenylephrine, or the NO donor sodium nitroprusside, to exposed medial cervical lymphatics of *Prox1-GFP* mice 30 min after intracisternal infusion of TMR–dextran. Thereafter, the lymphatic diameter and TMR–dextran fluorescence were measured in the lymphatics over 20 min (Fig. 5a), as described for other lymphatics[32]. Topical application of phenylephrine (50 µM, 500 µM or 5 mM) triggered both phasic and tonic lymphatic contractions and reduced TMR–dextran fluorescence in a concentration-dependent manner (Fig. 5b,c). By contrast, topical application of sodium nitroprusside (25 mM) increased the lymphatic diameter by 50–70% over the entire period of observation and transiently increased TMR–dextran fluorescence by 25–35% over the first 5 min (Fig. 5b,c). No changes in the diameter or TMR–dextran fluorescence were found after topical application of PBS (Fig. 5b,c). These findings provide evidence that the diameter of the medial cervical lymphatics increased with NO-mediated dilatation and decreased with α1-adrenergic-mediated constriction. Here we found that topical application of phenylephrine at a high concentration (5 mM) led to a 44% decrease in TMR–dextran fluorescence in the dcLNs, and sodium nitroprusside (3 µM) increased the fluorescence by 33% (Extended Data Fig. 9a–c). However, topical application of a low concentration of phenylephrine (10 nM) increased TMR–dextran fluorescence in the dcLNs by 51%, even more than sodium nitroprusside (Extended Data Fig. 9a–c). These findings provide evidence that CSF outflow can be increased by pharmacological manipulation of smooth muscle cells on deep cervical lymphatics.

## Ageing-related atrophy of the NPLP

Vascular alterations are common during ageing[33], and lymphatics are no exception[34,35]. Moreover, CSF outflow to dcLNs decreases with age in mice[12,15,21,26]. We examined whether ageing-related changes occurred in the nasopharyngeal plexus or medial cervical lymphatics

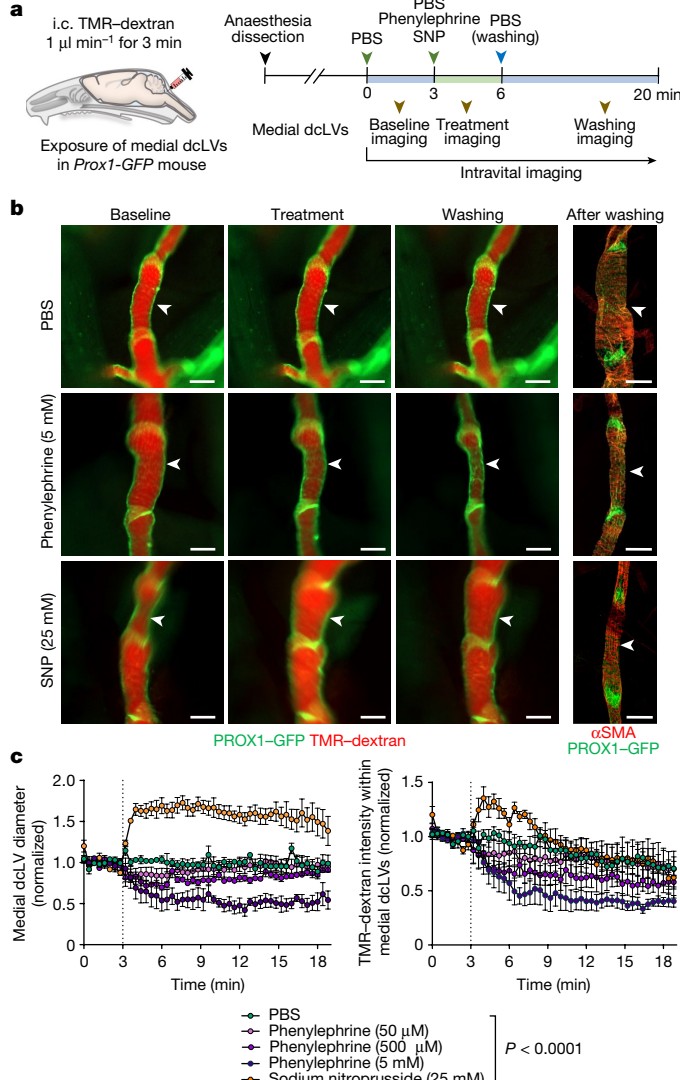

**Fig. 5 | Regulation of CSF outflow by myogenic control of medial deep cervical lymphatics. a**, Diagram of the experimental sequence for intracisternal infusion of TMR–dextran at 1.0 µl min⁻¹ for 3 min, and intravital imaging of medial deep cervical lymphatics (medial dcLVs) during pharmacological manipulation in *Prox1-GFP* mice. SNP, sodium nitroprusside. **b**, Fluorescence images showing TMR–dextran fluorescence in medial dcLVs (white arrowheads) at 2 min before treatment, during treatment and during washing. Right, immunofluorescence images of whole mounts stained for PROX1-dense lymphatic valves and αSMA⁺ circumferential smooth-muscle cells in medial dcLVs at 10 min after washing. Scale bars, 200 µm. Similar findings were obtained from *n* = 4 mice in three independent experiments. **c**, Changes in diameter and TMR–dextran fluorescence in the medial dcLVs over 17 min after the onset of five different pharmacological manipulations (vertical dotted lines). Data are mean ± s.e.m. *n* = 4 mice per group from four independent experiments. Values were normalized to the mean baseline value for each group. *P* values were calculated using two-way repeated-measures ANOVA.

by comparing mice aged 8–12 weeks (adult mice) with aged 73–102 weeks (aged mice). Analysis of the nasopharyngeal plexus in the aged mice revealed that the area of PROX1–GFP fluorescence was 14% less, the number of PROX1–GFP⁺FOXC2⁺ valves was 41% less, but LYVE1 staining was 64% greater and VEGFR3 staining was not different compared to adult mice (Fig. 6a,b and Supplementary Fig. 10).

Endothelial cells in the NPLP stained for PROX1–GFP and LYVE1 in mice aged 8–10 weeks had an oak-leaf shape, had a mixture of button junctions, zipper junctions and junctions with intermediate features, and lacked smooth-muscle coverage (Fig. 6a and Extended Data Fig. 10a). However, endothelial cells in this lymphatic plexus in aged mice had multiple abnormal features (Fig. 6a,b). Staining for phosphorylated tau and apoptosis (TUNEL) in these cells of aged mice was 2.5-fold and 7.4-fold greater than the corresponding values in adult mice (Extended Data Fig. 10b,c), raising the possibility that phosphorylated tau accumulation in the CSF contributes to the regression of the plexus in aged mice. By contrast, lymphatic valves, smooth-muscle cell coverage, and the length and diameter of the lymphangions of medial cervical lymphatics were similar in mice of both age groups (Extended Data Fig. 10d,e). These data are consistent with the contribution of ageing-related regression of the nasopharyngeal plexus to the reduction in CSF outflow to dcLNs in aged mice.

To determine the reversibility of the regression of the nasopharyngeal plexus and reduced CSF outflow to dcLNs in aged mice, we expanded the plexus by infusing AAV9-mVEGF-C-mCherry or AAV9-mCherry controls (3 × 10¹⁰ gene copies) into the cisterna magna of *Prox1-GFP* mice aged 75–78 weeks (Extended Data Fig. 11a). Three weeks later, the size of the PROX1⁺ plexus in the AAV9-mVEGF-C group was 1.3-fold greater than the size in the AAV9-mCherry controls (Extended Data Fig. 11b,c). CSF outflow to the posterior nasal and nasopharyngeal plexuses in the AAV9-mVEGF-C group was confirmed by intracisternal infusion of 0.5 µm FluoSpheres (Extended Data Fig. 11d). The AAV9-mVEGF-C group had a 2.6-fold higher amount of CSF outflow to dcLNs, assessed 30 min after intracisternal infusion of TMR–dextran, compared with the AAV9-mCherry controls (Extended Data Fig. 11e). These changes in aged mice were statistically significant but less than observed in 10-week-old mice (Extended Data Fig. 8b–d).

## Functional impairment of the NPLP with ageing

To determine whether the physiological and pharmacological properties of the medial cervical lymphatics changed during ageing, we performed ex vivo functional tests on isolated, pressurized single lymphangions (Extended Data Fig. 12a) using previously described methods[36–38]. Most lymphatics near the dcLNs did not have substantial (>5 µm) spontaneous contractions at any pressure between 0.5 and 10 cmH₂O (Extended Data Fig. 12b), as reported for other lymphatics in some regions of the mouse[36]. Segments of medial cervical lymphatics further from lymph nodes were more likely to develop spontaneous contractions (Extended Data Fig. 12c), with amplitudes of 15–20 µm and ejection fractions of 0.31–0.37 at some pressures. Similarly, most of these lymphatics developed only 5–10% spontaneous tone (Extended Data Fig. 12d), and tone did not increase with pressure, as observed in popliteal and inguinal-axillary lymphatics[36]. Most medial cervical lymphatics constricted after abluminal application of phenylephrine, and some were exquisitely sensitive (Extended Data Fig. 12e), with the threshold concentration <10 nM. In those cases, phenylephrine induced spontaneous contractions of variable amplitude (Extended Data Fig. 12e). As the lymphatics had little or no spontaneous tone, we could not assess responses to NO donors unless the vessels were preconstricted with phenylephrine. Thus, medial cervical lymphatics were exposed to increasing concentrations of phenylephrine until the constriction plateaued. The average tone achieved was around 30% of passive diameter, with a median effective concentration (EC₅₀) ≈ 50 nM for phenylephrine (Extended Data Fig. 12f). Responses to the NO donor sodium NONOate were tested at this tone. The lymphatics were quite sensitive to NONOate, with IC₅₀ ≈ 300 nM for adult mice (Extended Data Fig. 12g). The lymphatics of aged mice responded similarly but were slightly less sensitive to phenylephrine and sodium NONOate (phenylephrine EC₅₀ ≈ 200 nM and less total dilatation for NONOate), but the differences between adult and aged mice were not statistically significant (Extended Data Fig. 12f,g).

As longitudinal intravital imaging of CSF outflow in the NPLP was not feasible, we estimated the CSF outflow from measurements of

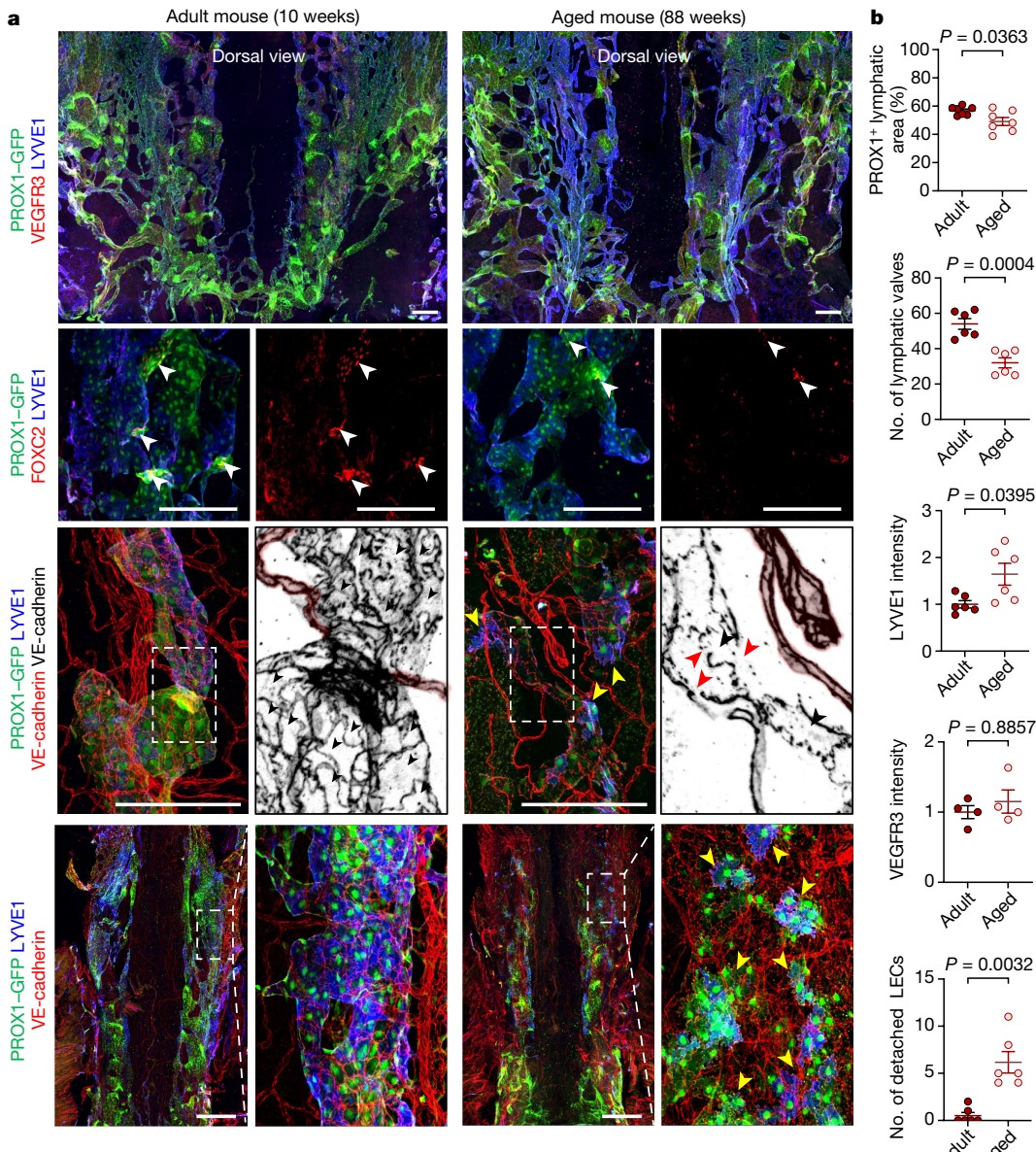

**Fig. 6 | Ageing-related alterations in the NPLP. a**, Immunofluorescence images of whole mounts showing the dorsal surface of the NPLP in adult (aged 10 weeks) and aged (aged 88 weeks) mice. Multiple abnormalities are evident in the aged mice. Row 1, the lymphatic plexus is smaller and has fewer valves. Row 2, PROX1-dense, FOXC2+ lymphatic valves (white arrowheads) are less numerous. Rows 3 and 4, the lymphatic plexus is smaller (the regions indicated by white dashed boxes are magnified in the adjacent monochrome panels); fewer LECs have an oak-leaf shape (black arrowheads); LECs have altered intercellular junctions (red arrowheads); and some cells appear to be detached from the adjacent cells (yellow arrowheads). The blood capillaries are marked by a pink overlay. Scale bars, 200 μm. Similar findings were obtained from $n = 7$ mice in three independent experiments. **b**, Comparison of the lymphatic area, valves, LYVE1 and VEGFR3 staining, and detached endothelial cells in the plexus between adult (aged 8–12 weeks) and aged (aged 73–102 weeks) mice. Each dot is the value for one mouse; $n = 7$ (PROX1+ lymphatic area), $n = 6$ (number of lymphatic valves), $n = 6$ (LYVE1 intensity), $n = 4$ (VEGFR3 intensity) and $n = 6$ (number of detached LEC) mice per group from three independent experiments. Data are mean ± s.e.m. $P$ values were calculated using two-tailed unpaired $t$-tests with Welch's correction or two-tailed Mann–Whitney $U$-tests.

TMR–dextran in the medial cervical lymphatics over a 30 min period after intracisternal infusion (Fig. 7a). TMR–dextran fluorescence was 34.1%, 45.1% and 28.8% less in the lymphatics of aged mice at 5, 10 and 15 min compared with in the lymphatics of adult mice (Fig. 7b,c). The pattern was similar at 20 and 30 min (Fig. 7b,c). Similarly, TMR–dextran fluorescence was 59% less in the dcLNs of aged mice at 30 min (Fig. 7d,e). Despite these differences, the size of dcLNs and the extent of PROX1+LYVE1+ lymphatics within them were not significantly different in adult and aged mice (Supplementary Fig. 11a,b). Together, these findings document the reduction in CSF outflow through the atrophic nasopharyngeal plexus in aged mice. They also reveal the preservation

of the structure and function of medial cervical lymphatics and dcLNs during ageing of these mice.

## Transcriptomic changes in the NPLP with ageing

To gain further insights into ageing-related changes in lymphatic endothelial cells (LECs) of the nasopharyngeal plexus, we performed single-cell RNA-sequencing (scRNA-seq) analysis of LECs isolated from the nasopharyngeal mucosa of adult (aged 10–12 weeks) and aged (aged 73–80 weeks) mice. Unsupervised clustering analysis of 842 single cells from 30 adult mice revealed five distinct clusters of LECs (Extended Data

Fig. 13a). Five subclusters were distinguished according to their differential gene expressions and were annotated by expression of known LEC subtype marker genes (Extended Data Fig. 13b). The pan-endothelial cell marker *Cdh5* was highly expressed in all of the clusters, as was the pan-LEC marker, *Prox1* (Extended Data Fig. 13c). The clusters were annotated as *Ptx3*[+] capillary LECs (*Ptx3*[+]*Stab2*[+]), *Ptx3*[−] capillary LECs (*Fndc1*[+]*Ptn*[+])[39], precollecting LECs (*Foxp2*[+]*Lyve1*[+]), upstream-valve LECs (*Cldn11*[+]*Gja4*[−]) and downstream-valve LECs (*Cldn11*[+]*Gja4*[+])[40] (Extended Data Fig. 13c). Unexpectedly, the proportion of *Ptx3*[+] capillary LECs, considered to be immune-interacting LECs[39,41], was greater than the proportion of *Ptx3*[−] capillary LECs. The greater abundance of *Ptx3*[+] LECs than *Ptx3*[−] LECs in the nasopharyngeal plexus was confirmed by immunofluorescence staining for PTX3 (Supplementary Fig. 12a,b). These transcriptomes were compared to those of 656 single LECs isolated from the nasopharyngeal mucosa of 25 aged mice. No difference was found in the five subclusters of LECs from adult and aged mice (Fig. 7f). However, heat-map and Gene Ontology (GO) analyses revealed that genes related to the apoptosis regulation and endothelial differentiation pathways in LECs were enriched more in adult mice compared with in aged mice (Fig. 7g and Extended Data Fig. 14a). By comparison, genes related to the response to type I interferon and regulation of leukocyte apoptosis and chemotaxis were enriched more in aged mice compared with in adult mice (Fig. 7g and Extended Data Fig. 14a). Moreover, expression of *Mcl1*[42] (anti-apoptotic gene) expression was lower in LECs of aged mice. Expression levels of *Il33*, Mhc genes, *Irf7*, *Ifitm2*, *Ifitm3* and *Zbp1*, which are related to inflammation and the type I IFN signalling pathway[43,44], were also higher in LECs of aged mice (Fig. 7h and Extended Data Fig. 14b). Overall, nasopharyngeal plexus LECs from aged mice had higher expression of pro-apoptotic and pro-inflammatory genes, which is a common feature of vascular ageing[33].

As the next step, we sought to determine whether upregulation of type I interferon signalling promotes regression of lymphatics in the NPLP and reduces CSF outflow to dcLNs by testing the effect of blocking interferon receptor signalling in aged mice (Extended Data Fig. 15a). However, no differences were detected in the morphology of the plexus or CSF outflow to dcLNs after an anti-mouse interferon α/β receptor subunit 1 (IFNAR-1)-blocking antibody was injected intraperitoneally (i.p.) into aged mice every 3 days for 6 weeks (Extended Data Fig. 15b–e). Despite the absence of changes in these readouts, the IFNAR-1-blocking antibody markedly reduced cyclic guanosine monophosphate-adenosine monophosphate (cGAMP)-stimulated mRNA expression of the interferon-stimulated genes *Oas2* and *Mx2* in nasopharyngeal tissues, thereby confirming the efficacy of the antibody (Extended Data Fig. 15f,g). Future experiments could address the issue of whether inhibition of type I interferon signalling for periods longer than 6 weeks can reduce plexus atrophy and increase CSF drainage in aged mice.

## Discussion

Our study identified the NPLP as a hub for CSF outflow. This lymphatic plexus was found to have unusual valves, short lymphangions and no smooth-muscle coverage. CSF from anterior and middle cranial regions of the subarachnoid space, including the pituitary gland, cavernous sinus and cribriform plate, drained through this lymphatic plexus to the dcLNs. Medial cervical lymphatics from the NPLP were found to be the largest drainage route for CSF to dcLNs. In contrast to the lymphatics in the plexus, medial cervical lymphatics had semilunar valves and smooth-muscle coverage typical of lymphangions that pumped towards lymph nodes. CSF outflow through the plexus and medial cervical lymphatic route downstream was on average 180% greater than through the lateral route from the basolateral dural lymphatics. The CSF outflow through the medial route was modulated by α-adrenergic activation and NO signalling in the lymphatic smooth muscle.

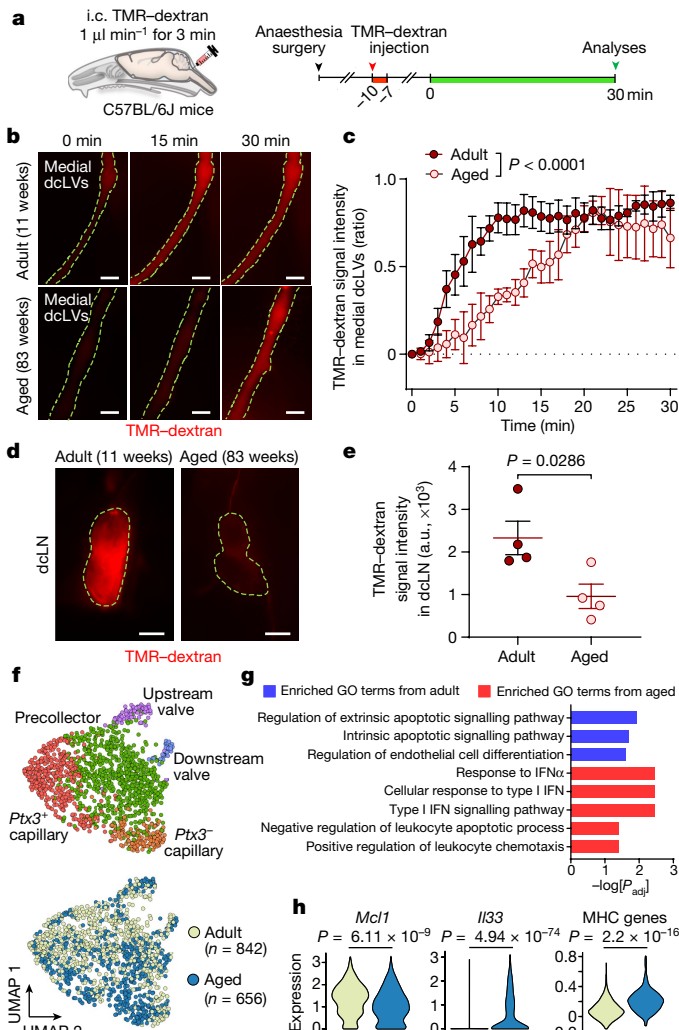

**Fig. 7 | Slowed CSF outflow through the medial deep cervical lymphatics in aged mice and ageing-related transcriptomic changes in LECs of the NPLP. a**, Diagram of the location and the experimental sequence of intracisternal infusion of TMR–dextran at 1.0 µl min[−1] for 3 min followed by measurement of TMR–dextran fluorescence in the medial deep cervical lymphatics (medial dcLVs) and dcLNs over 30 min by intravital imaging in C57BL/6J mice. **b**–**e**, Fluorescence images (**b,d**) and measurements (**c,e**) comparing TMR–dextran fluorescence in the medial dcLVs (**b,c**; outlined by yellow dashed lines) and dcLNs (**d,e**, outlined by yellow dashed lines) of adult (aged 11 weeks) and aged (aged 83 weeks) mice over 30 min after intracisternal infusion. Scale bars, 200 µm (**b**) and 500 µm (**d**). For **c**, data are mean ± s.e.m. for *n* = 4 mice per group in four independent experiments. *P* values were calculated using two-way repeated-measures ANOVA. For **e**, each dot is the value for one mouse. *n* = 4 mice per group in four independent experiments. a.u., arbitrary units. Data are mean ± s.e.m. *P* values were calculated using two-tailed Mann–Whitney *U*-tests. **f**, Uniform manifold approximation and projection (UMAP) plot visualizing five subclusters of LECs in the nasopharyngeal mucosa of adult (aged 10–12 weeks) and aged (aged 73–80 weeks) mice. The five subclusters of LECs are conserved in aged mice. The total number of LECs analysed was 1,498. **g**, GO analysis of the genes enriched in adult or aged mice. The list shows the top three GO terms significantly enriched in adult LECs (blue) and the top five GO terms significantly enriched in aged LECs (red). *P* values were calculated using the Benjamini–Hochberg correction method for multiple-hypothesis testing. **h**, Three example genes that were differentially expressed in adult and aged mice. *P* values were calculated using two-tailed model-based analysis of single cell transcriptomics (MAST) with Bonferroni post hoc test or two-tailed Wilcoxon rank-sum test.

Importantly, this feature was preserved during ageing, even when the nasopharyngeal plexus had atrophied, was functionally impaired and had LECs with pro-inflammatory changes in gene expression. Overall, the study highlights the importance of nasopharyngeal lymphatics and the transport properties of medial cervical lymphatics in CSF drainage. The study also highlights the potential for exploiting the sustained pharmacological responsiveness of medial cervical lymphatics to increase CSF outflow when clearance is impaired by ageing. These nearly inaccessible lymphatics can be functionally manipulated through drug effects on enhancing CSF transport.

Lymphatics are abundant in all regions of the upper respiratory tract, including the nose, nasopharynx and trachea[45,46]. They are also abundant in the hard and soft palates and other parts of the oral cavity. Despite this widespread presence of lymphatics, our study revealed that the lymphatics in the nasopharynx preferentially function as a route for CSF outflow to dcLNs. The nasopharynx is close to the cribriform plate and sphenoid bone, where cranial nerves I to VI exit the skull foramina. CSF outflow is known to drain through lymphatics in cranial nerve sheaths en route to lymph nodes in the neck[2,5,17,20,24,25,47,48]. Although we identified three upstream regions of lymphatics that connect to the nasopharyngeal plexus, additional lymphatics in cranial nerve sheaths contribute to CSF drainage[24]. Overall, these findings are consistent with a recent report[16].

Cervical lymphatics are thought to be responsible for around 50% of CSF outflow to cervical lymph nodes[3,12,17]. The remainder of CSF outflow drains from the spinal cord to mediastinal, iliac and sacral lymph nodes[14,24,49], or through perivascular spaces[3], although lymphatics can contribute to the latter. Here we revealed that the nasopharyngeal/medial cervical lymphatic route carries more CSF than the basolateral dura/lateral cervical lymphatic route. For this reason, we consider the NPLP to be an important route for CSF outflow, particularly from the base of the skull. Our findings also documented the importance of the lymphangion-pumping action of medial cervical lymphatics in the regulation of CSF outflow from the skull anteriorly, whereas the lateral route has a similar role for CSF outflow from the dural lymphatics posteriorly.

We also found that CSF outflow is actively regulated by adrenergic and NO signalling in lymphatic smooth muscle. Although most medial cervical lymphatics examined ex vivo did not have appreciable tone or spontaneous contractions, as is typical of lymphatics from other regions ex vivo[50], they were highly sensitive to phenylephrine. As a consequence, noradrenaline from sympathetic innervation and circulating catecholamines could increase tone and reactivity of these lymphatics. Indeed, the results indicate that the lymphatics have around 50% basal tone in vivo, which would enable NO donors to promote dilatation in the absence of exogenous phenylephrine. Importantly, our findings demonstrate that CSF drainage to dcLNs can be increased by topical application of a low concentration of phenylephrine or sodium nitroprusside to deep cervical lymphatics. However, pharmacological manipulation of deep cervical lymphatics in clinical applications would require less invasive approaches, such as innovative drug targeting or delivery methods. Methodological advancements could also enable measurement of the dynamics of deep cervical lymphatics in vivo. Nonetheless, the findings provide proof of concept for increasing CSF outflow by pharmacological manipulation of deep cervical lymphatics.

Among the limitations of our study, deep anaesthesia and removal of neck musculature were required to expose the nasopharynx and medial cervical lymphatics, both of which could alter the physiological dynamics of CSF drainage. Another potential limiting factor is the effect of surgical interventions on CSF drainage through lymphatics to the nasopharyngeal plexus, because cerebral blood flow and vascular pulsation contribute to CSF circulation, which in turn influences CSF outflow[51]. Although the imaging methods used were very informative, more advanced intravital imaging methods such as synchrotron X-ray

imaging with CSF tracers could reveal additional features of the dynamics of CSF drainage through the nasopharyngeal plexus and medial cervical lymphatics under physiological conditions. The nasopharyngeal plexus expanded in response to VEGF-C overexpression and regressed naturally during ageing but could not be selectively ablated in mice in loss-of-function studies performed to determine the adaptability of other routes for CSF clearance.

The decrease in CSF outflow with ageing is reported to result from reduced CSF production, changes in intracranial circulation and impaired lymphatic efflux[20,21,26,52–54]. Although multiple factors can reduce lymphatic efflux, our findings indicate that ageing-related atrophy and molecular alterations in LECs of the nasopharyngeal plexus contribute to the reduction in CSF outflow. The most prominent alterations in aged mice were apoptosis and regression of LECs in the dorsal side of the plexus, which is the first destination of CSF outflow. The plexus also had fewer valves in aged mice. Although the mechanism of the ageing-related changes in lymphatics is unclear, contributing factors could be continuous exposure to metabolites and substances (for example, phosphorylated tau) from the brain. Further studies are warranted to identify the mechanisms underlying the alterations and to develop methods for their prevention and reversal. As the downstream deep cervical lymphatics were unchanged in aged mice, they are potential targets for pharmacological manipulation to increase CSF outflow under pathological conditions.

Despite technical challenges, sufficient LECs were isolated from the nasopharyngeal mucosa of adult and aged mice for meaningful scRNA-seq analysis. Typical of LECs of other organs[39,55], nasopharyngeal lymphatics were found to have five distinct subclusters. The subcluster of *Ptx3*[+] capillary LECs in the plexus was larger than in other organs[39], raising the possibility of greater changes in inflammation or immune surveillance[39,41]. Genes related to inflammation and type I interferon signalling[43,44] were enriched in aged LECs from the plexus, which is considered to be a hallmark of vascular ageing[33] and is consistent with findings of a previous study[21] in aged dural lymphatics.

Dysregulation of CSF outflow through dural lymphatics is reported to exaggerate the phenotype in models of Alzheimer's disease[21,23] and to delay recovery after experimental stroke[56] or traumatic brain injury[57]. Despite the clinical relevance, the contribution of dural lymphatics to the severity of experimental autoimmune encephalomyelitis, an animal model of multiple sclerosis, is controversial. Some evidence[22] indicates that the severity can be alleviated by reducing dural lymphatics, but more recent studies[58,59] report little or no beneficial effect. Expansion of the dural lymphatic network with VEGF-C can increase resistance to glioblastoma progression by promoting immunosurveillance against the tumour[60]. These diverse activities illustrate that lymphatics not only contribute to CSF clearance but also to central nervous system immune surveillance, immune cell turnover, and other normal and pathological processes.

Overall, our findings emphasize the importance of the NPLP as a hub for CSF outflow and highlight the potential for increasing CSF outflow under pathological conditions by pharmacological activation of enhancing CSF transport through medial cervical lymphatics.

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

## Methods

### Study approval

All animal care and experimental procedures were approved by Institutional Animal Care and Use Committees of the Korea Advanced Institute of Science and Technology (KAIST) (KA2023-014-v1) and the University of Missouri (9797) for the mice and the Korea Research Institute of Bioscience & Biotechnology (KRIBB-AEC-22237) for the primates.

### Animals

C57BL/6J mice (aged 8 to 12 weeks) were purchased from DBL or from JAX. Aged C57BL/6J mice (aged 73 to 102 weeks) were purchased from the Animal Center of Ageing Science of Korea Basic Science Institute or from JAX. *Prox1-GFP* mice[18] (aged 8 to 12 weeks (adult) and 73 to 102 weeks (aged)) were bred and maintained under specific-pathogen-free conditions at KAIST. All of the mice fed with ad libitum access to a standard diet and water were bred under a 12 h–12 h light–dark cycle at 23–24 °C and 40–60% humidity. Mice of both sexes were used for all of the experiments. Mice were anaesthetized by i.p. injection of a mixture of anaesthetics (80 mg per kg ketamine and 8 mg per kg xylazine) or urethane (1.5 mg per kg) before or during being the procedures. The heart and respiratory rates of the mice were measured using a physiological monitoring system (75-1500, Harvard Apparatus). The body temperature was maintained at 36.5–37.5 °C during the entire surgical and imaging procedures. All experiments were performed during the light period. The head and neck portions of the primate (*M. fascicularis*, aged 6–13 years) were collected during autopsy at the National Primates Center of KRIBB.

### Intravital imaging of CSF outflow from the nasopharynx to deep cervical lymphatics

To acquire intravital images of CSF outflow in the nasopharynx, and deep cervical lymphatics and lymph nodes, 3 µl of PBS containing TMR–dextran (10 kDa, 50 mg ml⁻¹ or 70 kDa, 25 mg ml⁻¹; Invitrogen, D1816) was infused at 1.0 µl min⁻¹ for 3 min into the intracranial cavity through the cisterna magna of *Prox1-GFP* or C57BL/6J mice. To begin this procedure, an anaesthetized mouse was placed into the prone position on a stereotaxic frame under a surgical microscope. The head was adjusted to a 90° angle to the body axis with the help of a mouthpiece to facilitate access to the cisterna magna. After a skin incision in the midline of the posterior neck, the muscle layers were carefully separated with microscissors. The atlanto-occipital membrane overlying the cisterna magna was superficially penetrated using a 33-gauge NanoFil needle (World Precision Instruments) and then 3 µl of PBS containing TMR–dextran was infused into the subarachnoid space at 1 µl min⁻¹ for 3 min using a micro-syringe (88000, Hamilton) and a micro-infusion machine (Fusion 100, Chemyx). The needle was left in the position for 10 min and slowly removed from the mouse to prevent a CSF leakage. The muscle layers and neck skin were then sutured with 6-0 black silk (Ailee, SK617). Alternatively, an i.h. infusion was made. To set up this procedure, a small hole was drilled at the medial–lateral axis 1 mm and anterior–posterior axis −1.5 mm relative to the bregma after exposure of the skull on a stereotaxic frame. A custom-made glass pipet (diameter, 20 µm) connected to a PE-20 catheter was inserted to a depth of 2 mm. PBS (300 nl) containing TMR–dextran was infused into the hippocampus at a rate of 100 nl min⁻¹ for 3 min using a micro-syringe and a micro-infusion machine. After the infusion, the glass pipet was left in place for 5 min to prevent backflow and then slowly removed. The hole was sealed with a mixture of resin and superglue. After the infusion, the abdominal aorta was cut to remove the blood, and the sternocleidomastoid and omohyoid muscles were dissected and retracted under a surgical microscope (SZX16, Olympus) after a midline incision of the neck skin was made. In this step, exsanguination was required for precise imaging to prevent blood from the massive dissection of neck muscles from obscuring the image field. The dcLNs on the longus collis muscle and lateral cervical lymphatics on the scalene muscle were then carefully exposed. By dissection of the space between the pharyngeal muscle and the digastric muscle, the medial cervical lymphatics adjacent to the hypoglossal nerve was exposed. Further dissections in the cephalic direction were made to obtain vital imaging from the nasopharynx to medial cervical lymphatics and from jugular foramen to lateral cervical lymphatics. To ensure proper placement, a 24-gauge polyethylene catheter (Angiocath Plus, BD, 382412) was inserted into the jugular foramen. To directly access the nasopharynx, the lower mandible was removed and the soft palate was peeled off. TMR–dextran outflow through the NPLP was then captured from the ventral and dorsal sides of the nasopharynx. Intravital images of the indicated region were captured using a fluorescence stereo zoom microscope (AxioZoom V16, Carl Zeiss) with a Plan-Neofluar Z ×1.0 objective lens with a HE-GFP or Cy3 filter (Carl Zeiss). The entire procedure of this perimortem imaging was performed within 5 min after cutting the abdominal aorta.

### Intracisternal delivery of FluoSphere microbeads, AAV-VEGF-C delivery, ligation of deep cervical lymphatics and pharmacological treatments

A total of 3 µl of FluoSphere microbead solution (diameter, 0.5 µm; polystyrene, carboxylate-modified surface, red fluorescent (580/605), 2% solids 98% dry weight, Thermo Fisher Scientific, F8887) or 3 µl of Texas-Red-conjugated ovalbumin (5 mg ml⁻¹, O23021, Thermo Fisher Scientific) was infused at 1.0 µl min⁻¹ for 3 min into the subarachnoid space of *Prox1-GFP* mice at the cisterna magna. Subsequently, the head was collected for the histological analysis as described in the 'Tissue preparation for histological analysis' section below. A total of 3 µl of AAV9-VEGF-C-mCherry (AAV9-275994-mCherry, Vector Biolabs) or AAV9-mCherry (7107, Vector Biolabs), with a concentration of $1 \times 10^{13}$ gene copies per ml in PBS was infused into the subarachnoid space of *Prox1-GFP* mice at the cisterna magna at 1 µl min⁻¹ over 3 min. At 3 weeks after infusion, TMR–dextran was similarly infused and its fluorescence was subsequently measured in the dcLNs. Thereafter, the nasopharynx, diaphragm and dura were removed for histological analysis. To determine which side of deep cervical lymphatics was more responsible for CSF outflow, either the medial or lateral cervical lymphatics was ligated with 10-0 polypropylene suture (W2794, Ethicon) after neck muscle dissection in 10-week-old *Prox1-GFP* mice. The same operation without the ligation was performed for the sham control group. Then, 24 h later, intravital imaging was performed after i.h. infusion of TMR–dextran. To examine the effects of pharmacological agents on CSF outflow through the deep cervical lymphatics or to the dcLNs, the medial cervical lymphatics were exposed and immersed with 100 µl of PBS after intracisternal infusion of TMR–dextran. After baseline imaging for 3 min, phenylephrine (10 nM, 1 µM, 50 µM, 500 µM or 5 mM), sodium nitroprusside (3 µM, 30 µM or 25 mM) or nothing in 100 µl of PBS was topically applied for 3 min, followed by washing with PBS. The diameter and TMR–dextran fluorescence of deep cervical lymphatics were then measured for 20–30 min. TMR–dextran fluorescence was also measured in the dcLNs at 30 min after the intracisternal infusion of TMR–dextran. All values were normalized to the mean baseline value.

### Long-term blocking of interferon type I signalling

Aged *Prox1-GFP* mice (aged 70–88 weeks) received i.p. injection of 200 µg of anti-interferon type I signalling blocking antibody (anti-IFNAR-1 antibodies; anti-mouse interferon α/β receptor subunit 1 antibodies, BE0241, BioXcell) or 200 µg mouse IgG isotype control (BE0083, BioXcell) every 72 h for 6 weeks. The blocking effects of the antibody were validated by measuring mRNA expression of the interferon-stimulated genes *Oas2* and *Mx2* in the nasopharyngeal tissue of adult (aged 10–14 weeks) C57BL/6J mice. The mice were pretreated

with 200 µg of anti-IFNAR-1 antibodies or IgG, 1 h before i.p. administration of 2′3′-cGAMP (300 µg, TLRL-NACGA23-5, InvivoGen) or PBS, and the tissues were collected 4 h later for analysis.

## Tissue preparation for histological analysis

To acquire the sagittal sectioned image (for Fig. 1b), at 60 min after intra-cisternal infusion of TMR–dextran, the head and neck of a *Prox1-GFP* mouse were cut at the C2 vertebrae level and sampled immediately after cutting the abdominal aorta to remove blood. The sample was then cut along the sagittal plane in half with a blade and images were captured without fixation using a fluorescence stereo zoom microscope (AxioZoom V16, Carl Zeiss) with a Plan-Neofluar Z ×1.0 objective lens with a HE-GFP or Cy3 filter (Carl Zeiss). For immunofluorescence staining analyses, after right atrium puncture, ice-cold PBS was perfused into the left ventricle to remove blood. Then, 4% paraformaldehyde (PFA) solution was injected through the left ventricle to fix the tissues. For whole-mount preparations, the dcLNs and the attached afferent lymphatics were sampled with the surrounding muscles. The naso-pharyngeal mucosa was isolated by removing the surrounding skull, nerves and soft tissues using a fine forceps and surgical microscissors under a surgical microscope. The collected tissues were post-fixed with 2% PFA solution for 2 h at 4 °C. For the cryo-section of the mouse head, the head was submerged into 2% PFA solution for 12 h at 4 °C for post-fixation. Subsequently, the samples were immersed in 0.5 M EDTA solution for 48 h at 4 °C for decalcification, dehydrated by submerging in 30% sucrose solution for 48 h at 4 °C, embedded and frozen in a frozen section medium (Leica), and cut into a 30 µm sections using a Cryocut Microtome (Leica). For tissue clearing, heads perfused with 4% PFA were immersed further in 4% PFA overnight at 4 °C for post-fixation, washed with PBS and incubated in CUBIC-L solution (TCI, T3740) with daily change for 7 days at 37 °C. After tissue clearing and PBS washing, the samples were subjected to decalcification, immunofluorescence staining and imaging. For preparation of the primates, the animals were perfused with ice-cold saline and then decapitated. The head samples were fixed with 4% PFA for 2 h, and 2% PFA for 12 h at 4 °C. Subsequently, the samples were decalcified with 0.5 M EDTA, pH 8.0 for 3 weeks at 4 °C. The samples were placed into a fresh EDTA solution every 4 days. The decalcified heads were trimmed along the following tissue boundaries: the anterior (choana), the posterior (occipital bone), the dorsal (optic nerve) and the ventral (uvula). Trimmed samples were cut in half along the sagittal plane and the brain was removed from the skull. For whole-mount preparations, the nasopharyngeal mucosa was carefully separated from the skull base and soft palate.

## Immunostaining

The samples were incubated in 5% normal donkey serum (017-000-121, Jackson ImmunoResearch) for 1 h at room temperature. Next, the samples were incubated with primary antibodies (1:400) dissolved in 5% normal donkey serum at 4 °C for 12 h. After washing in PBS, the samples were incubated with secondary antibodies (1:1,000) dissolved in 5% normal donkey serum at 4 °C for 12 h. The samples that had been processed for clearing and decalcification were incubated with donkey serum for 24 h at room temperature; then with primary antibodies at 1:200 dilution at room temperature for 10 days; and finally with secondary antibodies at 1:100 dilution at room temperature for 3 days. After PBS washing, the samples were covered with DAPI-containing mounting medium (H1200, Vector) or refractive index matching solution (D-PROTOSS)[61]. The primary antibodies used were as follows: anti-LYVE1 (rabbit polyclonal, 11-034, Angio-bio), anti-CD31 (hamster monoclonal, 2H8, MAB1398Z, Merck), anti-VE-cadherin (goat polyclonal, AF1002, R&D), anti-VEGFR3 (goat polyclonal, AF743, R&D), anti-αSMA-Cy3 (mouse monoclonal, 1A4, C6198, Sigma-Aldrich), anti-β3 tubulin (mouse monoclonal, 2G10, ab78078, Abcam), anti-FOXC2 (sheep polyclonal, AF6989, R&D), anti-LYVE1 (rabbit polyclonal, DP3500, OriGene), anti-collagen type IV (goat polyclonal, AB769, Merck), anti-laminin α5 (rabbit polyclonal, EWL004, Kerafast), anti-tyrosine hydroxylase (rabbit polyclonal, AB152, Merck), anti-vesicular acetylcholine transporter (goat poly-clonal, ABN100, Merck), anti-phospho-tau (mouse monoclonal, AT8, MN1020, Thermo Fisher Scientific), anti-mannose receptor (CD206, rabbit polyclonal antibody, ab64693, Abcam), anti-PTX3 (rabbit polyclonal antibody, ALX-210-365-C050, Enzo Life Sciences). The following secondary antibodies were used: Alexa Fluor 488-, 594- and 647- conjugated anti-rabbit (711-545-152, 711-585-152, 711-605-152), anti-goat (705-585-147), anti-sheep (713-585-147) and anti-hamster (127-605-160) secondary antibodies (Jackson ImmunoResearch) in blocking buffer overnight at 4 °C. All of the antibodies used in this study were validated for the species and applications by the indicated manufacturers.

## TUNEL assay

To detect apoptotic cells in the nasopharynx, the terminal deoxynu-cleotidyl transferase biotin-dUTP nick end labelling (TUNEL) assay was performed according to the manufacturer's instructions (12156792910, Merck).

## Imaging and morphometric analyses

Immunofluorescence images were acquired using the LSM800 or LSM880 confocal microscope (Carl Zeiss). ZEN (v.2.3) software (Carl Zeiss) was used for the acquisition and processing of images. Confocal images of whole mounts or sections of tissues were maximum-intensity projections of tiled or single-plane *z*-stack images through the entire thickness of tissues. All of the images had a resolution of 512 × 512 or 1,024 × 1,024 pixels and were obtained with the following objectives: air objectives Plan-Apochromat ×10/0.45 numerical aperture (NA) M27 and Plan-Apochromat ×20/0.8 NA M27; LD C-Apochromat ×40/1.1 NA water-immersion Corr M27 (LSM 880) with multichannel scanning in the frame. The samples that underwent tissue clearing and decalcifica-tion were imaged using a light-sheet fluorescence microscope (LSFM, Carl Zeiss) with an EC Plan-Neofluar ×5/0.16 lens. Morphometric meas-urements were performed using ImageJ software (NIH) or Zen soft-ware (Carl Zeiss) on maximum-intensity-projection confocal images. The PROX1+ lymphatic area and the number of lymphatic valves and detached LECs were measured on the dorsal side of the nasopharynx at the following boundaries: anterior (most posterior part of poste-rior nasal lymphatic plexus), posterior (Eustachian tube), and lateral (a perpendicular line from Eustachian tube to nasopharyngeal lymphatics). The number of lymphatic valves and detached LECs were manually counted. Signal intensities of VEGFR3 and LYVE1 were meas-ured in the dorsal side of the nasopharynx in the region (1.5 mm×3 mm) defined by the aforementioned boundaries. The PROX1+ lymphatic area in the diaphragm was analysed in four 500 µm × 500 µm random fields per sample. The PROX1+ lymphatic area around the superior sagittal sinus was analysed in four 400 µm × 800 µm fields located near the confluence of sinus per sample. VE-cadherin+ junctional patterns of endothelial cells of the nasopharyngeal lymphatics were analysed in 200 µm × 200 µm as previously described[15]. Button-like junctions were defined as discontinuous, dot-like intercellular junc-tions, whereas zipper-like junctions were defined as continuous inter-cellular junctions. Junctions that did not match either pattern were categorized as mixed type. The number of lymphatic valves and the length of lymphangions in the deep cervical lymphatics were manu-ally measured. αSMA+ smooth muscle coverage per lymphangion was measured in three lymphangions of each deep cervical lymphatics using a Weka trainable segmentation of ImageJ plugin[62]. Phospho-rylated tau was measured in 1 mm ×1 mm regions of the dorsal side of nasopharyngeal lymphatics. TUNEL-positive cells were counted and expressed as the percentage of total LECs in two randomly selected 150 µm × 150 µm fields on the dorsal side of nasopharyngeal lymphatics.

## Ex vivo studies of pressurized deep cervical lymphatics

Mice were anaesthetized by i.p. injection of ketamine–xylazine and placed face up onto a heated tissue dissection/isolation pad. A proximal-to-distal incision was made in the skin from the neck to the sternum. While trimming loose facia, the submandibular gland and thymus on one side of the mouse were retracted with small clamps to expose the trachea and muscles overlying dcLNs. A 0.5–1.5 mm long segment of medial cervical lymphatic vessel was removed using fine forceps and microscissors and transferred to a dish containing Krebs solution + 0.5% BSA. The procedure was repeated on the other side of the animal. Both deep cervical lymphatics were then pinned with short segments of 40 μm stainless steel wire onto a Sylgard-coated dissection chamber filled with Krebs-BSA buffer at room temperature. The surrounding adipose and connective tissues were removed by microdissection. An isolated dcLV was then transferred to a 3 ml observation chamber on the stage of a Zeiss inverted microscope, cannulated, pressurized to 1 cmH$_2$O using two glass micropipettes (50–60 μm outer diameter). With the vessel pressurized, the segment was cleared of the remaining connective and adipose tissue. Polyethylene tubing was attached to the back of each glass micropipette and connected to a computerized pressure controller, with independent control of inflow and outflow pressures. To minimize diameter-tracking artifacts associated with longitudinal bowing at higher intraluminal pressures, input and output pressures were briefly raised to 10 cmH$_2$O at the beginning of each experiment, and the vessel segment was stretched axially to remove any longitudinal slack. After this procedure, each dcLV was allowed to equilibrate at 37 °C with pressure set to 1 cmH$_2$O. Constant exchange of Krebs buffer was maintained using a peristaltic pump at a rate of 0.5 ml min$^{-1}$. Within 30 min after the temperature stabilized, some vessels began to exhibit spontaneous contractions. Custom LabVIEW programs (National Instruments) acquired real-time analogue data and digital video through an A-D interface (National Instruments) and detected the inner diameter of the vessel[63]. Videos of the contractile activity of lymphatics were recorded for further analyses under bright-field illumination at 30 fps using a firewire camera (Basler, Graftek Imaging).

## Assessment of responses to pressure, phenylephrine and NONOate

To assess physiological responses to pressure, intraluminal pressure of deep cervical lymphatic segments was lowered from 1 to 0.5 cmH$_2$O, then raised to 1, 2, 3, 5, 8 and 10 cmH$_2$O, while recording internal diameter for 1–2 min at each pressure. Both the input and output pressures were maintained at equal levels so that there was no imposed pressure gradient for forward flow. After pressure was returned to 1 cmH$_2$O for 5 min, phenylephrine was applied to the bath in cumulative concentrations, while recording diameter for 1–2 min at each concentration. Once a maximum level of tone had been reached (typically 40–50% of the passive diameter), diethylamine NONOate sodium salt hydrate (sodium NONOate, Merck) was applied in cumulative concentrations, while measuring diameter at each concentration. At the end of each experiment, the vessel was equilibrated by perfusion with calcium-free Krebs buffer containing 3 mM EGTA for 30 min, and passive diameters were obtained at each level of intraluminal pressure.

## Plate-based single-cell sequencing of nasopharyngeal LECs

Nasopharyngeal tissue was used to isolate LECs from both sexes of adult ($n$ = 30) and aged ($n$ = 25) mice. After anaesthesia, the mice were perfused with ice-cold PBS, and the nasopharyngeal mucosa was removed and pooled in DMEM/F12 medium (Gibco). The tissue was cut into small pieces and incubated in dissociation buffer containing 1 mg ml$^{-1}$ of collagenase IV (Roche), 1 mg ml$^{-1}$ of dispase (Gibco) and 0.1 mg ml$^{-1}$ DNase I (Gibco) at 37 °C for 30 min with gentle inverting every 10 min. Digested samples were filtered through a 70 μm strainer and 2% FBS

was added to stop digestion. The cells were centrifuged for 8 min at 500$g$ and resuspended with PBS for washing. To exclude dead cells, 1:1,000 of Ghost dye (TONBO bioscience) was added to the resuspended cells for 15 min at 4 °C. PBS was then added for washing followed by staining with phycoerythrin/Cy7 anti-mouse CD326 (Ep-CAM, G8.8, 118216, BioLegend) antibodies, APC anti-mouse podoplanin antibodies (8.1.1, 127410, BioLegend) and phycoerythrin-labelled anti-mouse CD31 antibodies (MEC13.3, 102508, BioLegend). CD31$^+$PDPN$^+$ cells were considered to be LECs and were sorted using the FACS Aria Fusion (Beckton Dickinson) system. Sorted LECs were directly placed into each well in a 96-well plate containing a lysis buffer. The plates were snap-frozen with liquid nitrogen and stored at −80 °C. Following the Smart-Seq3 protocol[64], plate-based single-cell libraries were generated. In brief, mRNAs from lysed cells were reverse transcribed. cDNAs were amplified and purified using Ampure XP beads (Beckman Coulter). Purified cDNAs were diluted (100 pg μl$^{-1}$) and tagmented using the Tn5 transposase included in the Nextra XT DNA library preparation kit (FC-131-1024, Illumina). Using custom index primers, tagmented products were amplified and then pooled into a single tube. After final cleanup using Ampure XP beads, the libraries were analysed using the TapeStation for quality control. Libraries passing the quality control checks were sequenced on the Illumina High-X platform.

## Pre-processing of single-cell sequencing data

Sequenced libraries were demultiplexed and aligned to mouse reference genome (mm10) by STAR (v.2.7.9.a). The featureCount (v.2.0.1) function from Subread package was used to merge the aligned files and to build raw read count matrices. For cell quality control, cells detected with less than 2,000 genes and cells with more than 10% of total reads mapped to mitochondrial genes were considered to be low-quality/dead cells and were discarded. At the gene level, genes expressed in less than three cells were removed from the expression matrix.

## Clustering analysis

For clustering and visualization of single cells, the R package Seurat was used (v.4.1.0). In brief, log$_2$ normalization was applied after dividing each count for a gene in a cell by the total number of counts in a given cell, with multiplication of $1 \times 10^4$ and addition of 1 pseudocount. Consequently, the resulting expression matrixes were transformed to have values similar to log-transformed counts per million. Then, the top 2,000 genes with the highest variability in each dataset were selected using the FindVariableFeatures function with the following options: selection.method = "vst". Those highly variable genes were scaled and centred while regressing out confounding variables such as number of total counts and the percentage of reads mapped to mitochondrial genes. Moreover, module scores for dissociation-induced genes and ribosomal genes were calculated using the AddModuleScore function and regressed.

For visualization in two-dimensional space, principal component analysis was performed, and the top 15 principal components were used as the input for UMAP analysis. For neighbourhood identification and cluster assignment, the shared nearest neighbourhood graph was built by using the top 15 principal components and the Louvain algorithm was applied. For identifying differentially expressed genes between cells, we used the FindMarkers function in Seurat with the following options: test.use = "MAST", logfc.threshold=0.3, min.pct=0.3. While performing differential expression testing, we excluded dissociation-induced genes, mitochondrial and ribosomal genes. In merging adult and aged mouse datasets, no batch correction method was used, as no evident batch effect was observed for clustering.

## Statistical analysis

Sample sizes were chosen on the basis of standard power calculations (with $\alpha$ = 0.05 and power of 0.8) and no statistical methods

were used to predetermine sample size. The experiments were randomized, and investigators were blinded to allocation during experiments and outcome assessment. Data were tested for normality using Shapiro–Wilk and Kolmogorov–Smirnov one-sample tests. Depending on the data distribution, parametric or nonparametric statistics were used. The statistical significance of differences was determined using two-tailed Student's *t*-test, two-tailed Welch's *t*-tests, Brown–Forsythe ANOVA, two-way ANOVA test or two-tailed Mann–Whitney *U*-tests. Two-way repeated-measures ANOVA was used when comparing the time-series data between the two groups. Statistical analysis was performed using Prism 10 (GraphPad Software, v.10.1.0). All data are presented as mean ± s.e.m. Statistical significance was set at $P < 0.05$.

## Reporting summary

Further information on research design is available in the Nature Portfolio Reporting Summary linked to this article.

## Data availability

The scRNA-seq data of this study are available at the NCBI Gene Expression Omnibus under accession codes GSE227311 (adult mice) and GSE227324 (aged mice). All other data supporting the findings in this study are available within the Article and its Supplementary Information. Source data are provided in the Supplementary Information.

## Code availability

The codes used for scRNA-seq are available at Zenodo (https://zenodo.org/records/10115336). The LabVIEW program used for pressure and diameter data collection of isolated lymphatic vessels is available at Zenodo (https://doi.org/10.5281/zenodo.8286107). The LabVIEW program used for diameter tracking of isolated lymphatic vessels is available at Zenodo (https://doi.org/10.5281/zenodo.8286119).

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

**Acknowledgements** We thank Y.-K. Hong for the *Prox1-GFP* mice. H.J. was supported by the MD-PhD/Medical Scientist Training Program from the Ministry of Science & ICT of Korea. This study was supported by the Institute for Basic Science and funded by the Ministry of Science and ICT, Republic of Korea (IBS-R025-D1-2015 to G.Y.K.) and R01 HL-122578 from the National Institutes of Health to M.J.D.

**Author contributions** J.-H.Y., H.J., H.J.K., D.M.M., M.J.D. and G.Y.K. conceived and designed the study. J.-H.Y., H.J., M.J.D. and H.J.K. performed most of the experiments and generated the figures. J.-H.Y., H.J., H.J.K., D.M.M., M.J.D. and G.Y.K. wrote the manuscript with contributions from S.P.H., M.J.Y., J.H.A. and Y.-C.K. under the supervision of D.M.M., M.J.D. and G.Y.K.; J.-H.Y., H.J., H.J.K., S.P.H., M.J.Y., D.M.M., M.J.D. and G.Y.K. analysed and interpreted the data and generated figures. M.J.Y. performed scRNA-seq with contributions from J.-H.Y. and H.J.; M.J.Y., J.-H.Y. and H.J. analysed scRNA-seq datasets. H.J.K. and M.J.D. performed ex vivo isolated afferent lymphatic vessel experiments and data analysis. J.-H.Y., H.J., H.J.K., S.P.H., M.J.Y., Y.-C.K. and J.H.A. actively discussed the experiments and the results under the supervision of D.M.M., M.J.D. and G.Y.K.; J.S. and Y.L. provided the primate tissue samples. M.J.D. and G.Y.K. supervised and directed the project.

**Competing interests** The authors declare no competing interests.

**Additional information**
**Correspondence and requests for materials** should be addressed to Michael J. Davis or Gou Young Koh.

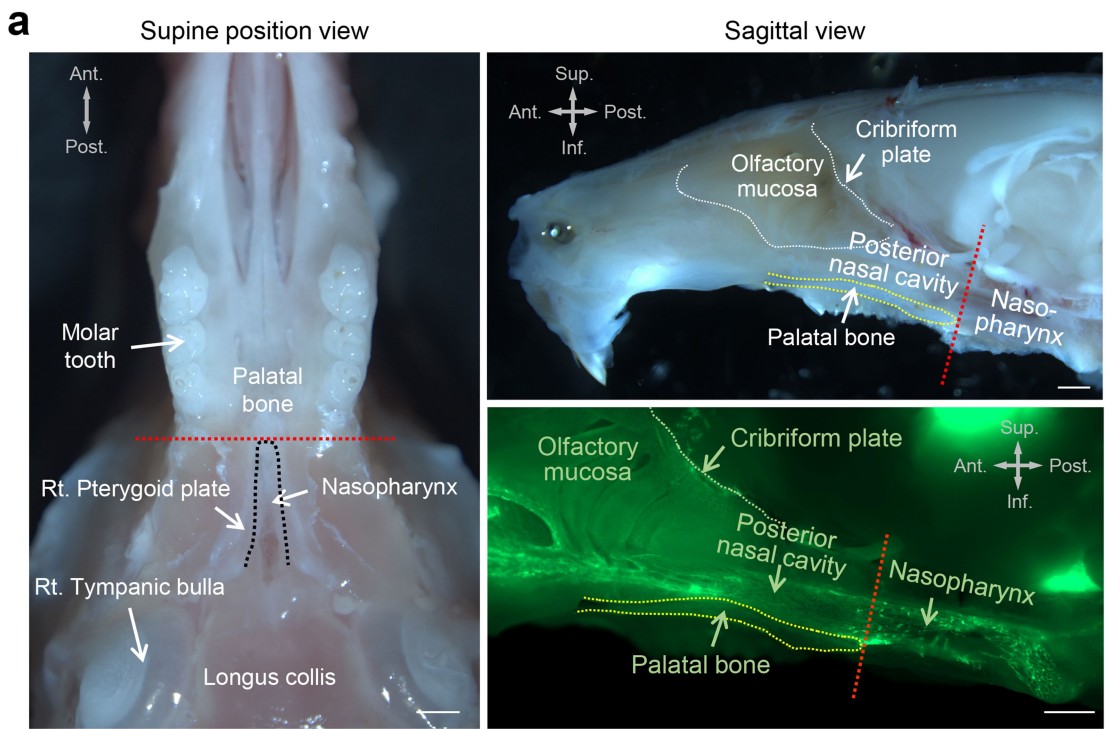

**a**

Supine position view

Sagittal view

Ant. / Post.

Molar tooth

Palatal bone

Rt. Pterygoid plate

Nasopharynx

Rt. Tympanic bulla

Longus collis

Sup. / Ant. ⟷ Post. / Inf.

Cribriform plate

Olfactory mucosa

Posterior nasal cavity

Palatal bone

Naso-pharynx

Olfactory mucosa

Cribriform plate

Posterior nasal cavity

Nasopharynx

Palatal bone

Sup. / Ant. ⟷ Post. / Inf.

Prox1-GFP

**b**

Posterior nasal lymphatic plexus

Nasopharyngeal lymphatic plexus

Skull base

Skull base

Prox1-GFP DAPI

**Extended Data Fig. 1 | Posterior margin of the palatal bone separates the nasopharyngeal lymphatic plexus from the posterior nasal lymphatic plexus. a**, Bright-field and fluorescence images of upper jaw and nasopharynx viewed from below (left) and laterally (upper right) and in a sagittal section through the palatal bone (lower right) of Prox1-GFP mice. The palatal bone is outlined (yellow dashed lines), and the posterior margin is marked with a red dashed line. The nasopharynx is outlined by a black dashed line. Anatomical positions are indicated in the upper left or right corner. Scale bars, 1 mm.

Similar findings were obtained from n = 4 mice in two independent experiments. Ant., anterior; Post., posterior; Sup., superior; Inf., inferior anatomical position. **b**, Fluorescence images of coronal sections of the nasopharynx at the level of the posterior nasal lymphatic plexus (left, white arrowhead) and the level of the nasopharyngeal lymphatic plexus (right, green arrowhead) in Prox1-GFP mice. White dashed line box regions are enlarged in the lower panels. Scale bars, 200 μm. Similar findings were obtained from n = 4 mice in two independent experiments.

**a**

Pituitary gland

Cranial nerve V

Clivus

Choana

Nasopharynx

Retropharyngeal space (dissected)

Soft palate

Uvula

Retropharyngeal lymph node (removed)

Longus collis

Dorsal
Ant. — Post.
Ventral

**b**

LYVE1 Collagen IV

**Extended Data Fig. 2 | Location of the nasopharyngeal lymphatic plexus in the mucosa of the nasopharynx of the primate, *Macaca fascicularis*. a**, Mid-sagittal section of the head and neck of *Macaca fascicularis* with relevant landmarks labelled. Anatomical positions are indicated in the lower left corner. Scale bar, 3 mm. Ant., anterior; Post., posterior anatomical position. **b**, Immunofluorescence images of thick sections of nasopharyngeal mucosa stained for LYVE1+ (green) of lymphatics and collagen IV+ (red) on vessels. Green dashed-line box in upper panel is enlarged in the lower panel. White arrowheads mark lymphatic valves. Scale bar, 1 mm. Similar findings were obtained from n = 5 *Macaca fascicularis* in two independent experiments.

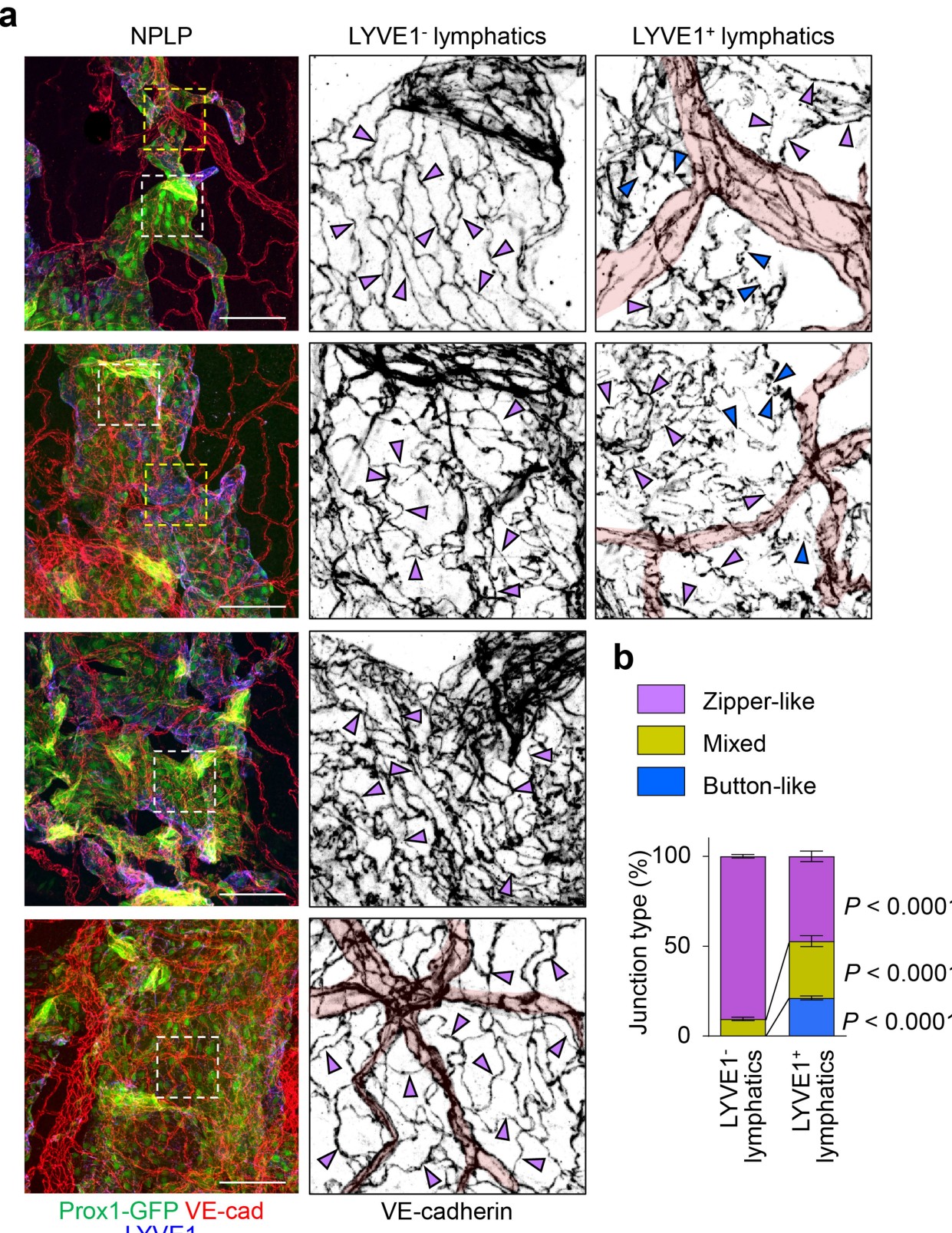

**a**

NPLP | LYVE1⁻ lymphatics | LYVE1⁺ lymphatics

Prox1-GFP  VE-cad
LYVE1

VE-cadherin

**b**

Zipper-like
Mixed
Button-like

Junction type (%)

$P < 0.0001$
$P < 0.0001$
$P < 0.0001$

LYVE1⁻ lymphatics
LYVE1⁺ lymphatics

**Extended Data Fig. 3 | Intercellular junctions of lymphatic endothelial cells of the nasopharyngeal lymphatic plexus. a,b,** Immunofluorescence images of whole mounts of the dorsal and ventral portions of nasopharyngeal lymphatic plexus (NPLP) stained for VE-cadherin (VE-cad) and LYVE1 in adult (8–10 weeks old) Prox1-GFP mice. White and yellow dashed-line box areas are enlarged in the centre and right panels. Blood capillaries are highlighted in pink. Zipper-like intercellular junction (magenta arrowheads) predominated (95%) in lymphatics with little or no LYVE1 staining, whereas button-like intercellular junctions (blue arrowheads) were more numerous (21%) in LYVE1⁺ lymphatics. Scale bars, 100 μm. Similar findings were obtained from n = 4 mice in two independent experiments. Bars indicate mean ± s.e.m. *P* values for junctions in LYVE1⁻ and LYVE1⁺ lymphatics were calculated by two-way ANOVA test followed by two-tailed Holm-Sidak's multiple comparison *post-hoc* test.

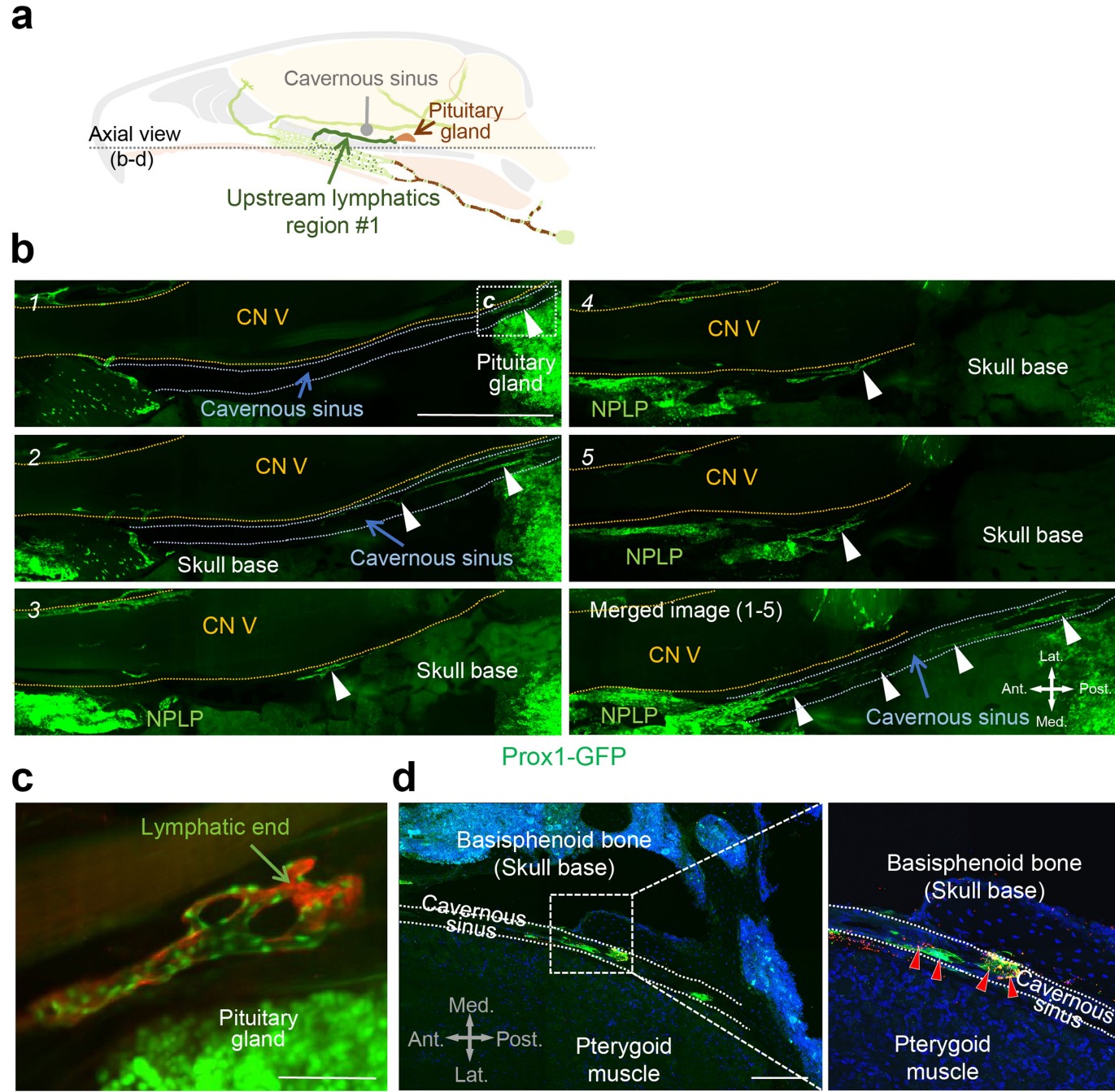

**Prox1-GFP** (panel b)

**Prox1-GFP LYVE1** (panel c)

**Prox1-GFP FluoSphere DAPI** (panel d)

**Extended Data Fig. 4 | Prox1+/LYVE1+ lymphatics near the pituitary gland extend to the nasopharyngeal lymphatic plexus along cranial nerve V and cavernous sinus of Prox1-GFP mice. a**, Diagram of a sagittal view of dural lymphatics designated upstream lymphatics region #1. **b**, Light-sheet fluorescence microscopic images showing serial optical sections (numbered) of upstream lymphatics region #1 (white arrowheads) originating near the Prox1+ pituitary gland. These lymphatics course along the cavernous sinus (outlined with blue-dotted lines) and beneath cranial nerve (CN) V (outlined with yellow-dotted lines) en route to the nasopharyngeal lymphatic plexus (NPLP). Anatomical positions are indicated in the lower right corner. Scale bar, 1 mm. Similar findings were obtained from n = 5 mice in three independent experiments. Ant., anterior; Post., posterior; Med., medial; Lat., lateral

anatomical position. **c**, Light-sheet fluorescence microscopic image showing LYVE1-stained (red), blunt-ended Prox1+/LYVE1+ dural lymphatics (green nuclei) in an enlargement of the region in **b** section 1 marked by white-dotted line box (**c**) near the Prox1+ pituitary gland (bright green). Scale bar, 100 μm. **d**, Fluorescence microscopic images of section showing Prox1+ upstream lymphatics region #1 containing FluoSpheres (red arrowheads) along the cavernous sinus. The region of the white dashed-lined box is enlarged in the right panel. Anatomical positions are indicated in the lower left corner. Scale bar, 200 μm. Similar findings were obtained from n = 3 mice in two independent experiments. Ant., anterior; Post., posterior; Med., medial; Lat., lateral anatomical position.

**a**

PPA

Axial view
(b-c)

Upstream lymphatics
region #2

**b**

*1*

Posterior nasal lymphatic plexus
(connected to NPLP)

Med.

Ant. ← → Post.

Lat.

*

PPA

*2*

Posterior nasal lymphatic plexus
(connected to NPLP)

Lymphatic
valve

*

Lymphatic
tuft

Upstream lymphatics region #2

PPA

Prox1-GFP LYVE-1

**c**

Tuft cells in olfactory epithelium

Med.

Ant. ← → Post.

Lat.

Lymphatic
tuft

*

Upstream lymphatics
region #2 along PPA

Posterior nasal
lymphatic plexus
(connected to NPLP)

Prox1-GFP FluoSphere

**Extended Data Fig. 5** | See next page for caption.

**Extended Data Fig. 5 | Lymphatics in upstream lymphatics region #2 along the pterygopalatine artery reach the nasopharyngeal lymphatic plexus through the posterior nasal plexus. a**, Diagram of sagittal view of upstream lymphatics region #2. **b**, Light-sheet fluorescence microscopic images showing serial optical sections (numbered) of lymphatics (outlined with green-dotted lines marked by white arrowheads) along the pterygopalatine artery (PPA, outlined with red-dotted lines) en route to the posterior nasal (green arrows) and nasopharyngeal lymphatic plexuses. A valve (green arrowhead) is located where lymphatics along the PPA join the posterior nasal lymphatic plexus. The white asterisk marks a segment of lymphatic with unknown connections.

Anatomical positions are indicated in the upper right corner. Scale bars, 200 μm. Similar findings were obtained from n = 5 mice in three independent experiments. Ant., anterior; Post., posterior; Med., medial; Lat., lateral anatomical position. **c**, Fluorescence images of sections showing Prox1⁺ lymphatics in upstream lymphatics region #2 containing FluoSpheres (red arrowheads) along the PPA. Tuft cells of the olfactory epithelium are Prox1⁺. Anatomical positions are indicated in the upper right corner. Scale bar, 200 μm. Similar findings were obtained from n = 3 mice in two independent experiments. Ant., anterior; Post., posterior; Med., medial; Lat., lateral anatomical position.

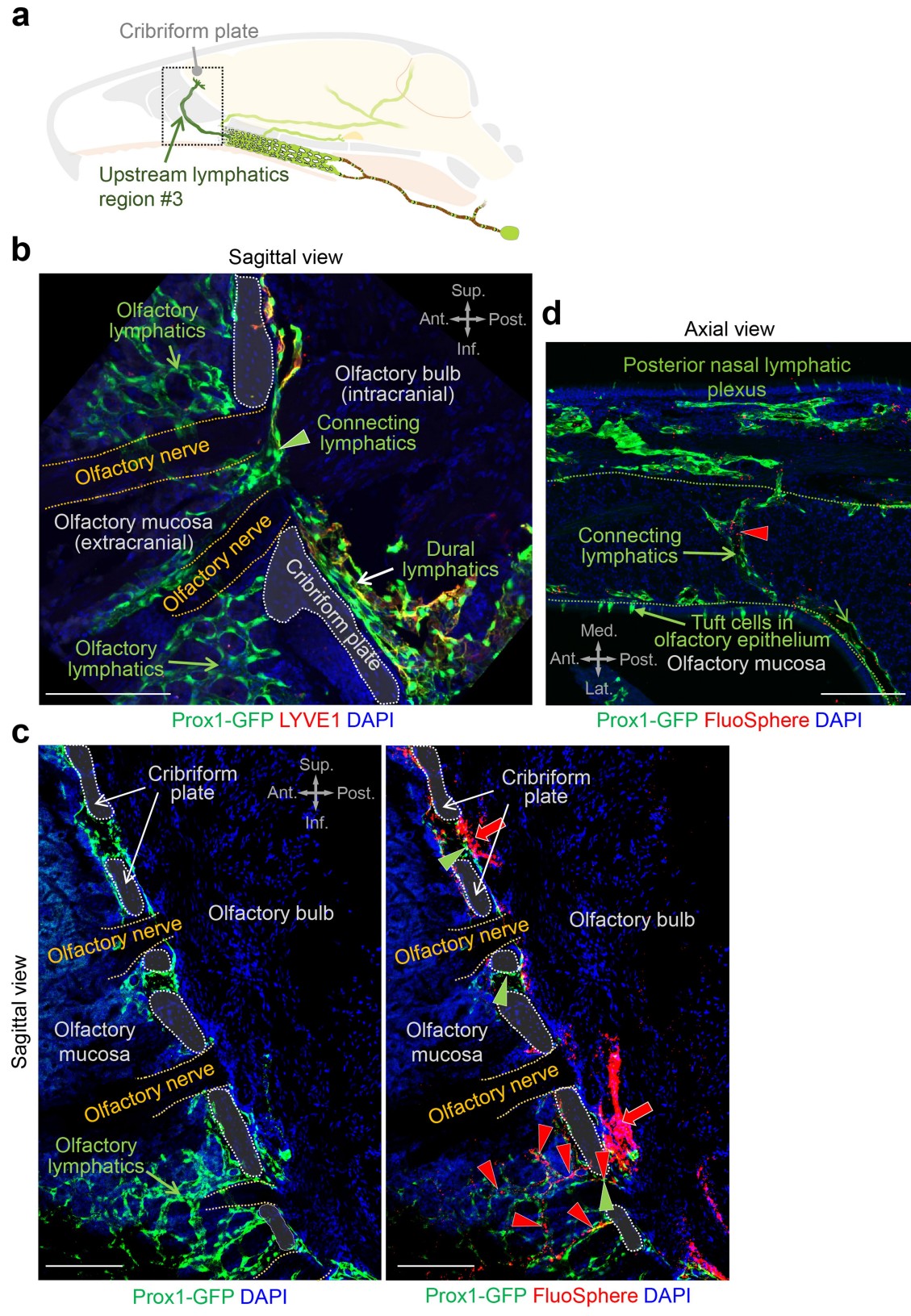

**a**

Cribriform plate

Upstream lymphatics region #3

**b**

Sagittal view

Olfactory lymphatics

Sup.
Ant. — Post.
Inf.

Olfactory bulb (intracranial)

Connecting lymphatics

Olfactory nerve

Olfactory mucosa (extracranial)

Olfactory nerve

Cribriform plate

Dural lymphatics

Olfactory lymphatics

Prox1-GFP LYVE1 DAPI

**d**

Axial view

Posterior nasal lymphatic plexus

Connecting lymphatics

Med.
Ant. — Post.
Lat.

Tuft cells in olfactory epithelium
Olfactory mucosa

Prox1-GFP FluoSphere DAPI

**c**

Sagittal view

Cribriform plate

Sup.
Ant. — Post.
Inf.

Olfactory nerve

Olfactory bulb

Olfactory mucosa

Olfactory nerve

Olfactory lymphatics

Prox1-GFP DAPI

Cribriform plate

Olfactory nerve

Olfactory bulb

Olfactory mucosa

Olfactory nerve

Prox1-GFP FluoSphere DAPI

**Extended Data Fig. 6** | See next page for caption.

**Extended Data Fig. 6 | Lymphatics near the olfactory bulb cross the cribriform plate with olfactory nerves, traverse the olfactory mucosa, and connect to the posterior nasal lymphatic plexus. a**, Drawing of a mid-sagittal view of upstream lymphatics region #3. **b**, Immunofluorescence image of section showing a sagittal view of Prox1$^+$ lymphatics near the cribriform plate (white dashed lines). Dural lymphatics near the olfactory bulb are LYVE1$^+$ (white arrow) but connecting lymphatics (green arrowhead) and lymphatics in the olfactory mucosa (green arrow) are LYVE1$^-$. Olfactory nerves are outlined by yellow dashed lines. Anatomical positions are indicated in the upper right corner. Scale bar, 100 μm. Similar findings were obtained from n = 3 mice in two independent experiments. Ant., anterior; Post., posterior; Sup., superior; Inf., inferior anatomical position. **c**, Immunofluorescence images of sagittal section through the olfactory bulb and cribriform plate (white dashed lines) showing adjacent Prox1$^+$ lymphatics (left panel, image optimized for Prox1) that contain FluoSpheres (right panel, red arrow, image optimized for microspheres). FluoSpheres are also present in perineural lymphatics (green arrowheads) within the plate and in the olfactory mucosa (red arrowheads). Olfactory nerves are outlined by yellow dashed lines. Anatomical positions are indicated in the upper right corner. Scale bars, 100 μm. Similar findings were obtained from n = 3 mice in two independent experiments. Ant., anterior; Post., posterior; Sup., superior; Inf., inferior anatomical position. **d**, Immunofluorescence image of section showing an axial view of Prox1$^+$ connecting lymphatics containing FluoSpheres (red arrowhead) located between the olfactory mucosa and the posterior nasal lymphatic plexus. Anatomical positions are indicated in the lower left corner. Scale bar, 200 μm. Similar findings were obtained from n = 3 mice in two independent experiments. Ant., anterior; Post., posterior; Med., medial; Lat., lateral anatomical position.

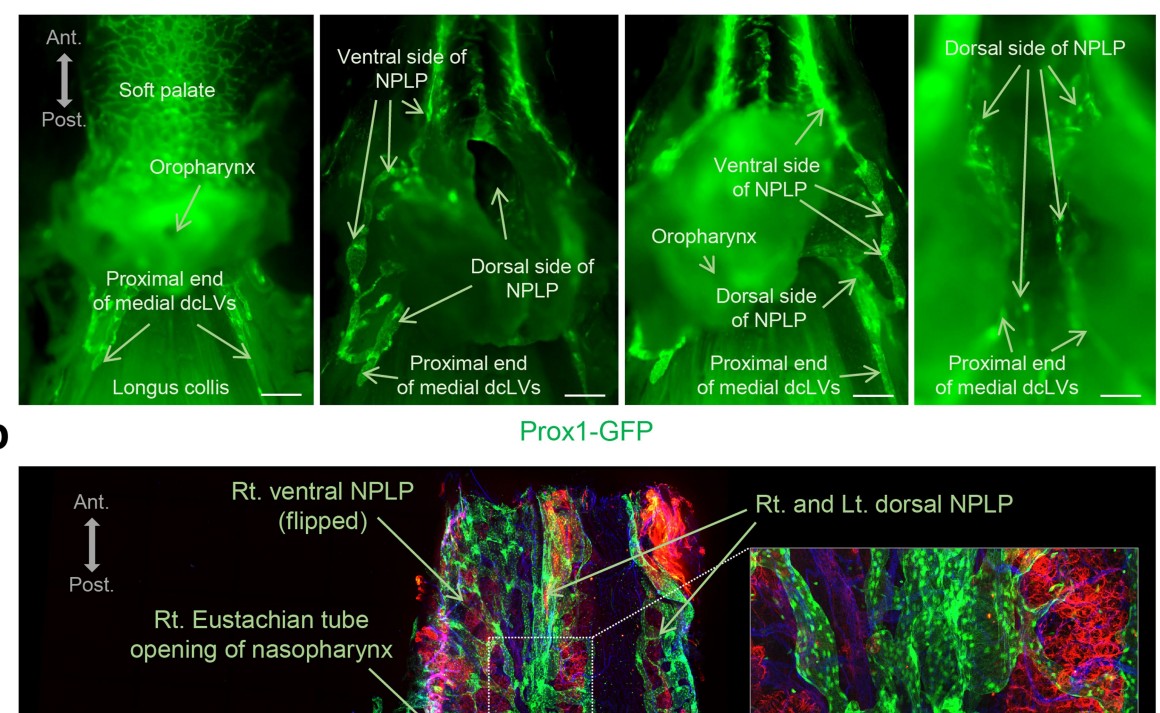

**a**

Prox1-GFP

**b**

Prox1-GFP αSMA CD31

**Extended Data Fig. 7** | See next page for caption.

**Extended Data Fig. 7 | Downstream lymphatics from the nasopharyngeal lymphatic plexus merge and connect to medial deep cervical lymphatics.** **a**, Fluorescence images of whole mounts of nasopharyngeal region showing four views of direct connections between lymphatics downstream from the nasopharyngeal lymphatic plexus (NPLP) and the proximal end of medial deep cervical lymphatics (medial dcLVs) of Prox1-GFP mice. Anatomical positions are indicated in the upper left corner. Scale bars, 1 mm. Similar findings were obtained from n = 4 mice in three independent experiments. Ant., anterior; Post., posterior anatomical position. **b**, Immunofluorescence images of nasopharyngeal whole mount stained for α-smooth muscle actin (αSMA, red) and CD31 (blue) in Prox1-GFP mice showing direct connections of downstream lymphatics from the NPLP to the proximal end of medial dcLVs. White dashed-line boxes mark three regions that are enlarged to show detail. Lymphatics from ventral and dorsal sides of the NPLP merge and become ventral and dorsal branches of the medial dcLVs, and then those branches merge and become the main medial dcLVs. The NPLP is not covered by αSMA⁺ smooth muscle cells (SMCs), but medial dcLVs are covered by SMCs from the proximal end to distal end. Anatomical positions are indicated in the upper left corner. Scale bar, 1 mm. Similar findings were obtained from n = 4 mice in three independent experiments. Ant., anterior; Post., posterior anatomical position.

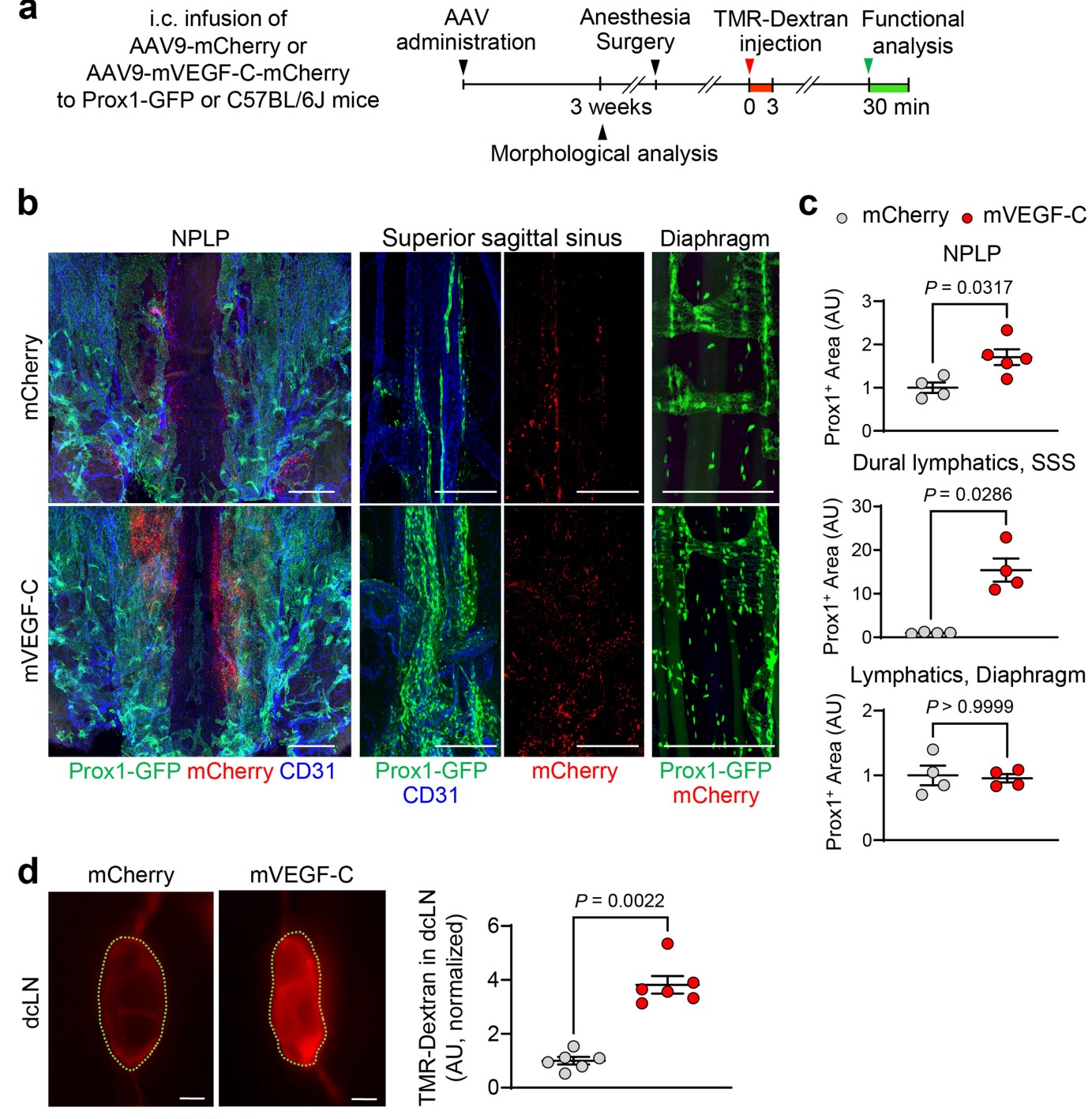

**Extended Data Fig. 8 | Intracisternal AAV-mVEGF-C delivery expands nasopharyngeal lymphatic plexus and increases CSF outflow to deep cervical lymph nodes. a**, Diagram of the experimental sequence of intracisternal (i.c.) infusion of $3 \times 10^{10}$ genome copies of AAV9-mCherry (mCherry) or AAV9-mVEGF-C-mCherry (mVEGF-C) in 3 µl of PBS, followed by i.c. infusion of TMR-Dextran at 1.0 µl/min for 3 min at 3 weeks later and measurement of TMR-Dextran fluorescence in deep cervical lymph nodes (dcLN) 30 min later in Prox1-GFP or C57BL/6J mice. **b,c**, Immunofluorescence images of tissue whole mounts stained for mCherry (red) and CD31 (blue) and area measurements comparing Prox1+ lymphatics in the NPLP, dura along the superior sagittal sinus (SSS), and diaphragm after delivery of mCherry or mVEGF-C. The similarity of mCherry expression in the NPLP and SSS of both groups is evidence that the AAV9-vectors were similarly transduced to these tissues. Scale bars, 500 µm

(NPLP) and 200 µm (SSS, diaphragm). Each dot is the value for one mouse; Prox1+ area of the NPLP (mCherry, n = 4; mVEGF-C, n = 5), Prox1+ area of the dural lymphatics of SSS (n = 4 mice/group), Prox1+ area of diaphragm (n = 4 mice/group), TMR-Dextran intensity in dcLN (n = 6 mice/group) in three independent experiments. Bars indicate mean ± s.e.m. AU (arbitrary unit) normalized to control mean. $P$ values were calculated by two-tailed Mann-Whitney test. **d**, Fluorescence images and measurements comparing TMR-Dextran fluorescence in deep cervical lymph nodes (dcLN, outlined by green dashed lines) in mice treated with mCherry or mVEGF-C. Scale bars, 200 µm. Each dot is the value for one mouse; n = 6 mice/group in three independent experiments. Bars present mean ± s.e.m. AU, arbitrary unit, normalized to mCherry mean. $P$ values were calculated by two-tailed Mann-Whitney test.

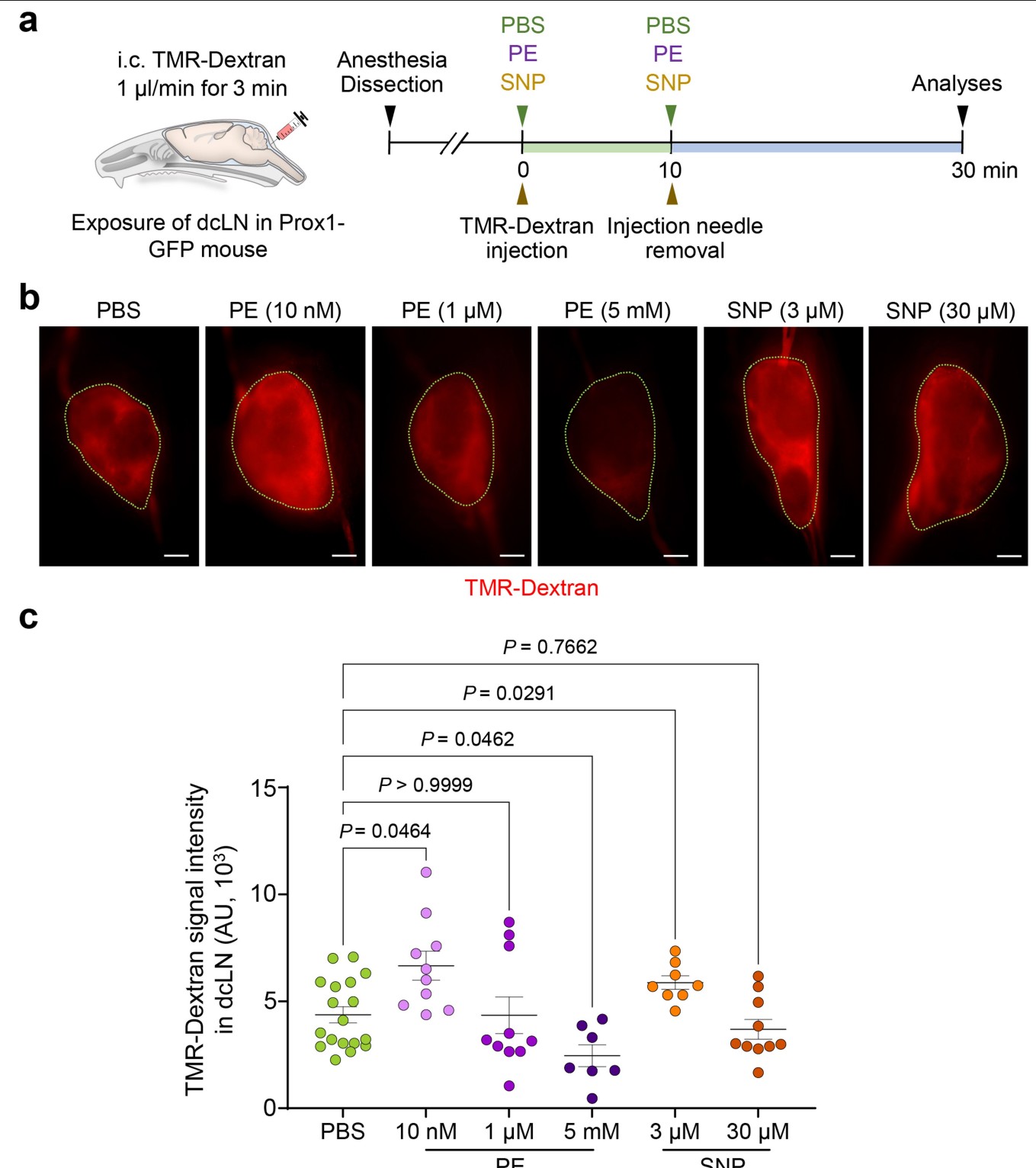

**Extended Data Fig. 9 | CSF drainage into deep cervical lymph nodes increased by topical application of low concentration of phenylephrine or sodium nitroprusside to deep cervical lymphatics. a**, Diagram of experimental sequence of intracisternal (i.c.) infusion of TMR-Dextran at 1.0 μl/min for 3 min, removal of injection needle, topical application of phenylephrine (PE) or sodium nitroprusside (SNP) to deep cervical lymphatics, and measurement of TMR-Dextran fluorescence in deep cervical lymph nodes (dcLN) of Prox1-GFP mice.

**b,c**, Fluorescence images and measurements of TMR-Dextran fluorescence in dcLNs (outlined by green dashed lines) at 30 min after application of multiple concentrations of PE or SNP. Scale bars, 200 μm. Each dot is the value for one mouse; PBS (n = 18), 10 nM PE (n = 10), 1 μM PE (n = 10), 5 mM PE (n = 7), 3 μM SNP (n = 8), 30 μM SNP (n = 10) in five independent experiments. Error bars indicate mean ± s.e.m. AU, arbitrary unit. *P* values were calculated by two-way ANOVA test followed by two-tailed Dunnett's T3 multiple comparison *post-hoc* test.

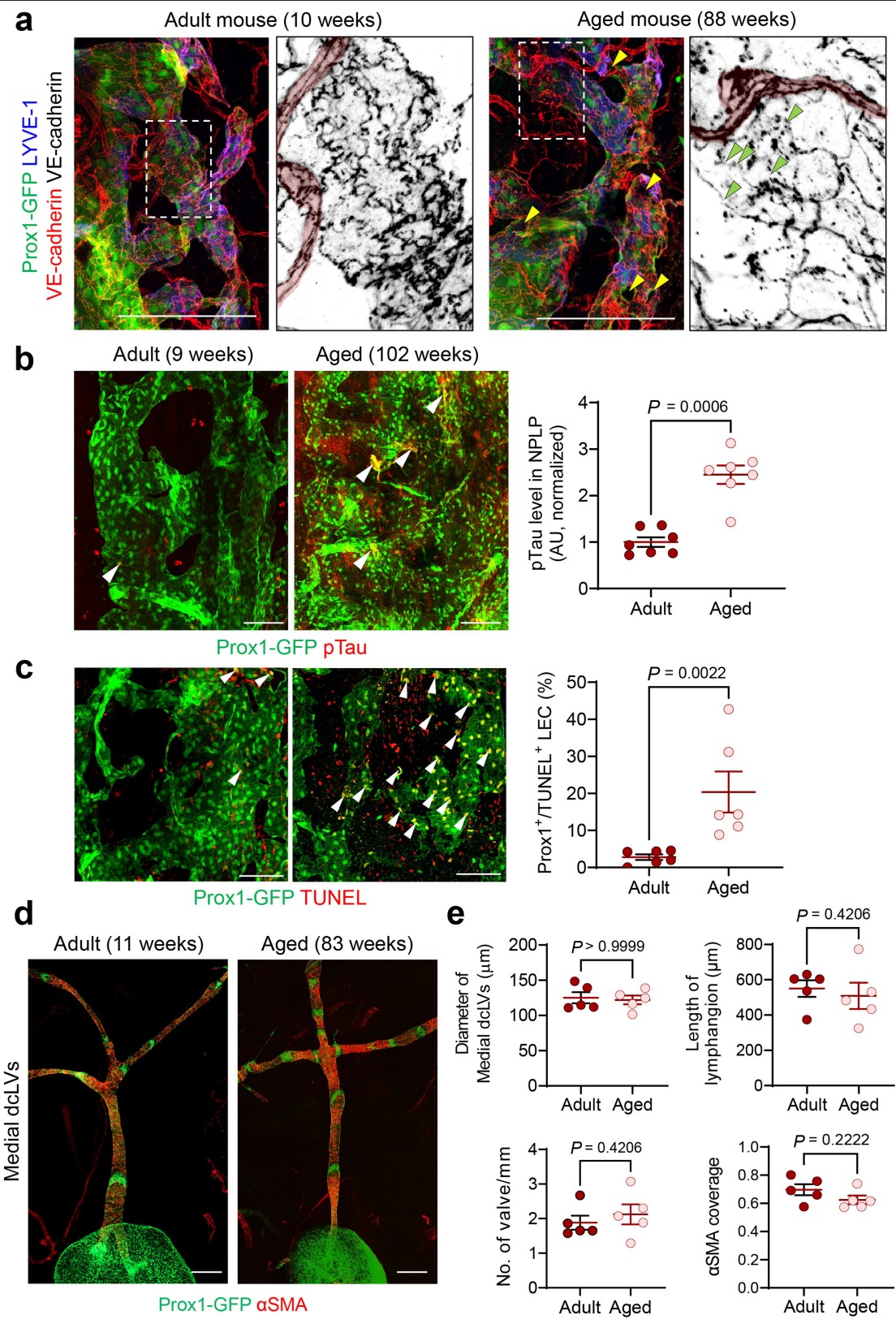

**Extended Data Fig. 10** | See next page for caption.

**Extended Data Fig. 10 | Ageing-related alterations in nasopharyngeal lymphatic plexus but no apparent ageing-related structural alterations of medial deep cervical lymphatics. a**, Immunofluorescence images of whole mounts comparing the dorsal region of the nasopharyngeal lymphatic plexus in adult (10 weeks old) and aged (88 weeks old) mice. White dashed-line boxed regions are enlarged in the monochrome panels. Blood capillaries are marked by pink highlighting. In aged mice, Prox1+ lymphatic plexus have disintegrated intercellular junctions (green arrowheads), and some of which appear detached (yellow arrowheads). Scale bars, 200 μm. Similar findings were obtained from n = 7 mice in three independent experiments. **b,c**, Immunofluorescence images of whole mounts comparing phosphorylated (pTau) and apoptosis (TUNEL+) in nasopharyngeal lymphatic plexus (NPLP) of young (9 weeks old) and aged (95-102 weeks old) mice. pTau+ and TUNEL+ nuclei (red) are significantly more numerous in lymphatic endothelial cells (white arrowheads) in the NPLP of aged mice. Scale bars, 100 μm. Each dot is the value for one mouse; pTau level in NPLP (n = 7 mice/group) and Prox1+/TUNEL+ LEC (n = 6 mice/group) in three independent experiments. Bars indicate mean ± s.e.m. AU, arbitrary unit. P values were calculated by two-tailed Mann-Whitney test. **d,e**, Immunofluorescence images of whole mounts comparing the diameter, length, valves (Prox1-GFP+, green), and smooth muscle coverage (α-smooth muscle actin, αSMA, red) of medial deep cervical lymphatics (medial dcLVs) in adult (10-11 weeks old) and aged (83-88 weeks old) mice. Scale bars, 200 μm. Each dot is the value for one mouse; n = 5 mice/group in three independent experiments. Bars indicate mean ± s.e.m. P values were calculated by two-tailed Mann-Whitney test.

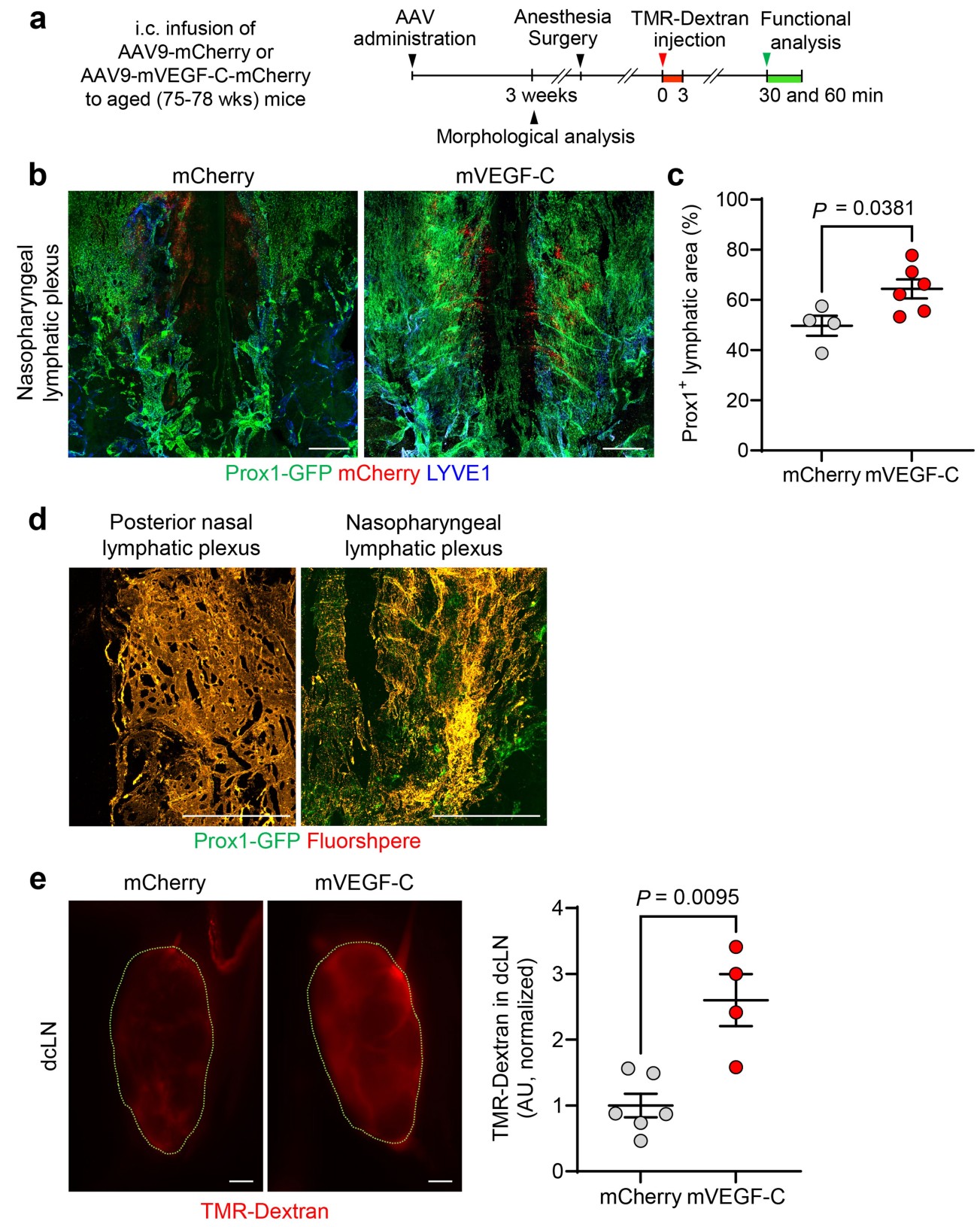

**Extended Data Fig. 11** | See next page for caption.

**Extended Data Fig. 11 | Expansion of nasopharyngeal lymphatic plexus and CSF outflow to deep cervical lymph nodes in aged mice by intracisternal AAV-mVEGF-C. a**, Diagram of the sequence of intracisternal (i.c.) infusion of $3 \times 10^{10}$ genome copies of AAV9-mCherry (mCherry) or AAV9-mVEGF-C-mCherry (mVEGF-C) in 3 µl of PBS followed 3 weeks later by i.c. infusion of TMR-Dextran at 1.0 µl/min for 3 min (or 0.5-µm FluoSpheres) and measurement of TMR-Dextran fluorescence in deep cervical lymph nodes (dcLN) 30 min later in aged (75–78 weeks old) Prox1-GFP or C57BL/6J mice. **b,c**, Immunofluorescence images comparing whole mounts of the nasopharyngeal lymphatic plexus after AAV delivery of mCherry or mVEGF-C. mCherry expression (red) is present in the NPLP of both groups, indicating that the AAV9-vectors were similarly transduced in these lymphatics. Scale bars, 500 µm. Each dot is the value for one mouse; Prox1⁺ lymphatic area (mCherry, n = 4; mVEGF-C, n = 6) in two

independent experiments. Bars indicate mean ± s.e.m. AU, arbitrary unit. *P* value was calculated by two-tailed Mann-Whitney test. **d**, Fluorescence images showing the distribution of FluoSpheres in lymphatics of whole mounts of the posterior nasal and nasopharyngeal plexuses at 1 h after the i.c. infusion to mice that received AAV9-mVEGF-C-mCherry. Scale bar, 500 µm. Similar findings were obtained from n = 3 mice in two independent experiments. **e**, Fluorescence images and measurements of TMR-Dextran fluorescence in deep cervical lymph nodes (dcLN, outlined by green dashed lines) in aged mice after AAV delivery of mCherry or mVEGF-C. Scale bars, 200 µm. Each dot is the value for one mouse; TMR-Dextran intensity in dcLN (mCherry, n = 6; mVEGF-C, n = 4) in three independent experiments. Bars present mean ± s.e.m. AU, arbitrary unit. *P* value was calculated by two-tailed Mann-Whitney test.

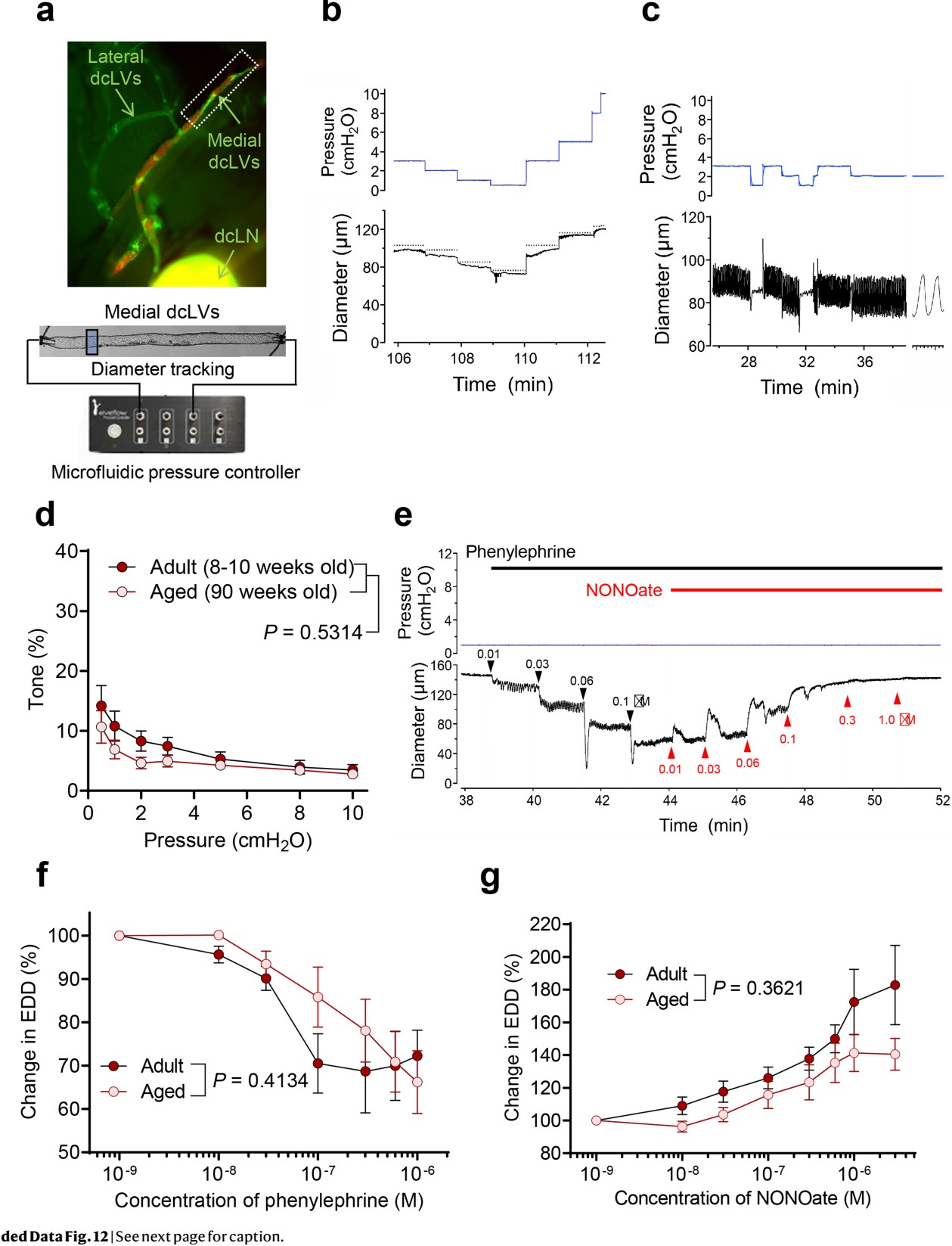

**Extended Data Fig. 12** | See next page for caption.

**Extended Data Fig. 12 | Active responses of ex vivo medial deep cervical lymphatics to pressure, phenylephrine, and NONOate. a**, Images showing medial deep cervical lymphatics (medial dcLVs) in situ and after cannulation ex vivo for measuring diameter and intraluminal pressure. The experimental setup enabled diameter measurements of isolated, cannulated lymphatics with intraluminal pressure governed by a microfluidic pressure controller. White box marks the segment of medial dcLVs typically dissected for ex vivo studies. **b**, Response of typical medial dcLV to pressure steps over the physiological range 0.5 to 10 cmH$_2$O. Only small spontaneous amplitude fluctuations are evident at lower pressures. Dotted lines indicate the passive diameter at each pressure, as determined in Ca$^{2+}$-free Krebs at the end of the experiment. **c**, Example of medial dcLV with large-amplitude spontaneous diameter changes (contractions) at pressure settings of 1 or 2 cmH$_2$O. Diameter changes ceased at 0.5 cmH$_2$O. Contraction frequency was 13-14 per min. Expanded trace on far right shows two individual contractions, each with a duration of 2.4 sec. **d**, Plot of active tone as a function of intraluminal pressure in medial dcLVs from adult (8-10 weeks old; n = 10) and aged (90 weeks old; n = 12) mice. Tone was calculated as the difference in active and passive end-diastolic diameter at each pressure and expressed as a percentage of the passive diameter at that pressure. Bars present mean ± s.e.m. $P$ value was calculated by two-way repeated measures ANOVA. **e**, Recordings of pressure and diameter that illustrate the dosing protocol for inducing tone and assessing concentration-dependent responses to phenylephrine, followed by assessment of concentration-dependent responses to sodium NONOate, at a constant pressure of 1 cmH$_2$O. Small spontaneous contractions follow the first exposure to phenylephrine (10 nM). **f**, Lack of age-related difference in concentration-dependent constrictions to phenylephrine in medial dcLVs from adult (8-10 weeks old; n = 11) and aged (90 weeks old; n = 12) mice. EDD, end diastolic diameter. Bars present mean ± s.e.m. $P$ value was calculated by a mixed effect analysis. **g**, Lack of age-related difference in concentration-dependent dilatations to sodium NONOate (in the continued presence of phenylephrine) in adult (8-10 weeks old; n = 11) and aged (90 weeks old; n = 12) mice. EDD, end diastolic diameter. Bars present mean ± s.e.m. $P$ value was calculated by two-way repeated measures ANOVA.

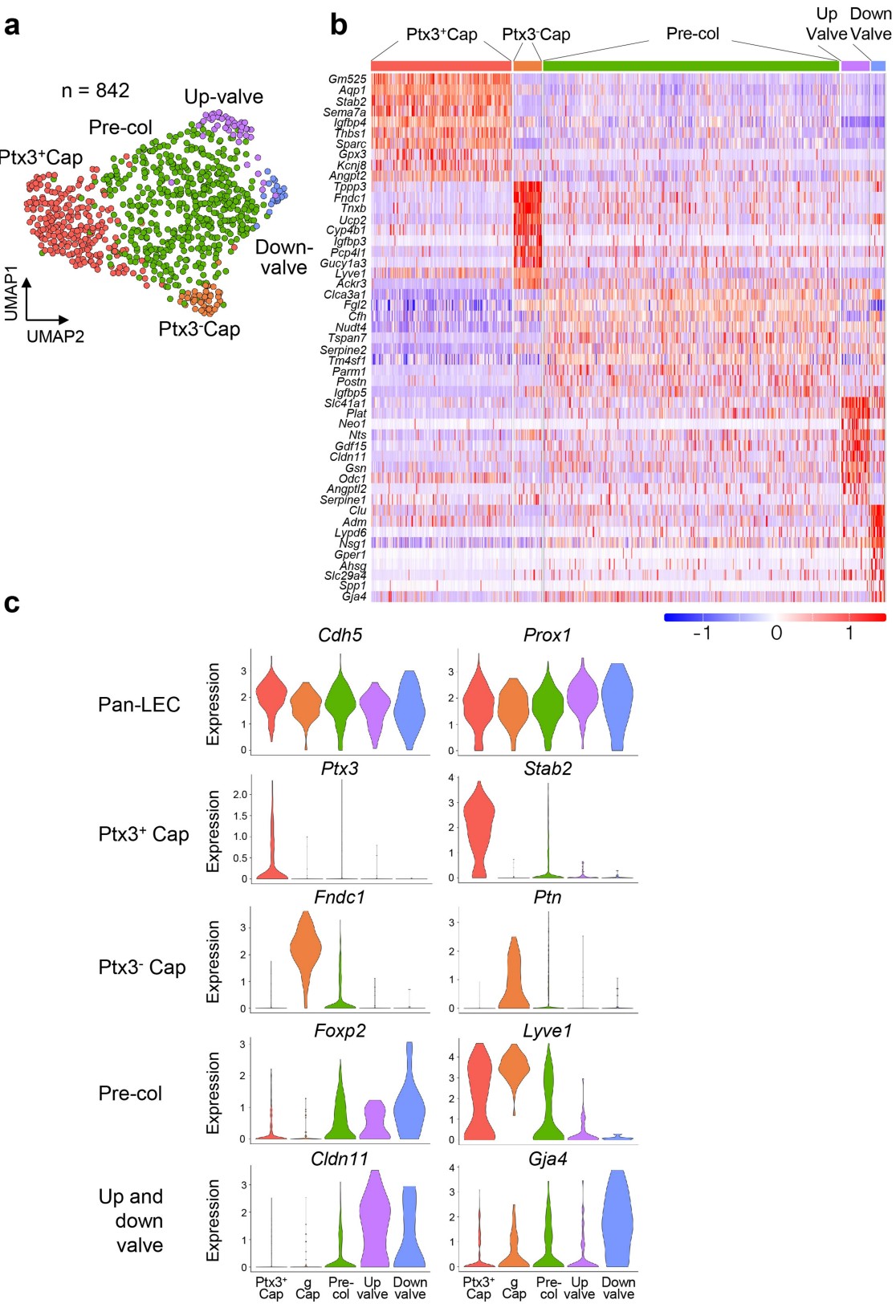

**Extended Data Fig. 13 | Single-cell transcriptomics of LECs of the nasopharynx in adult mice. a**, UMAP plot visualizing five subclusters of lymphatic endothelial cells (LECs) in the nasopharyngeal mucosa of adult (10-12 weeks old) mice. Pentraxin-3 (Ptx3), Ptx3⁻ capillary LECs (Ptx3⁻ Cap), Ptx3⁺ capillary LECs (Ptx3⁺ Cap), pre-collecting LECs (Pre-col), upstream valve LECs (up-valve), downstream valve LECs (down-valve). Total number of LECs analysed = 842. **b**, Heatmap showing differential expression of genes for the five subclusters of LECs. **c**, Violin plots comparing the expression of ten representative genes in the five sub-clusters. Little difference was found in *Cdh5* or *Prox1* among the subclusters, but *Ptx3*, *Stab2*, *Foxp2*, *LYVE1*, *Cldn11*, and *Gja4* had prominent cluster-related differences.

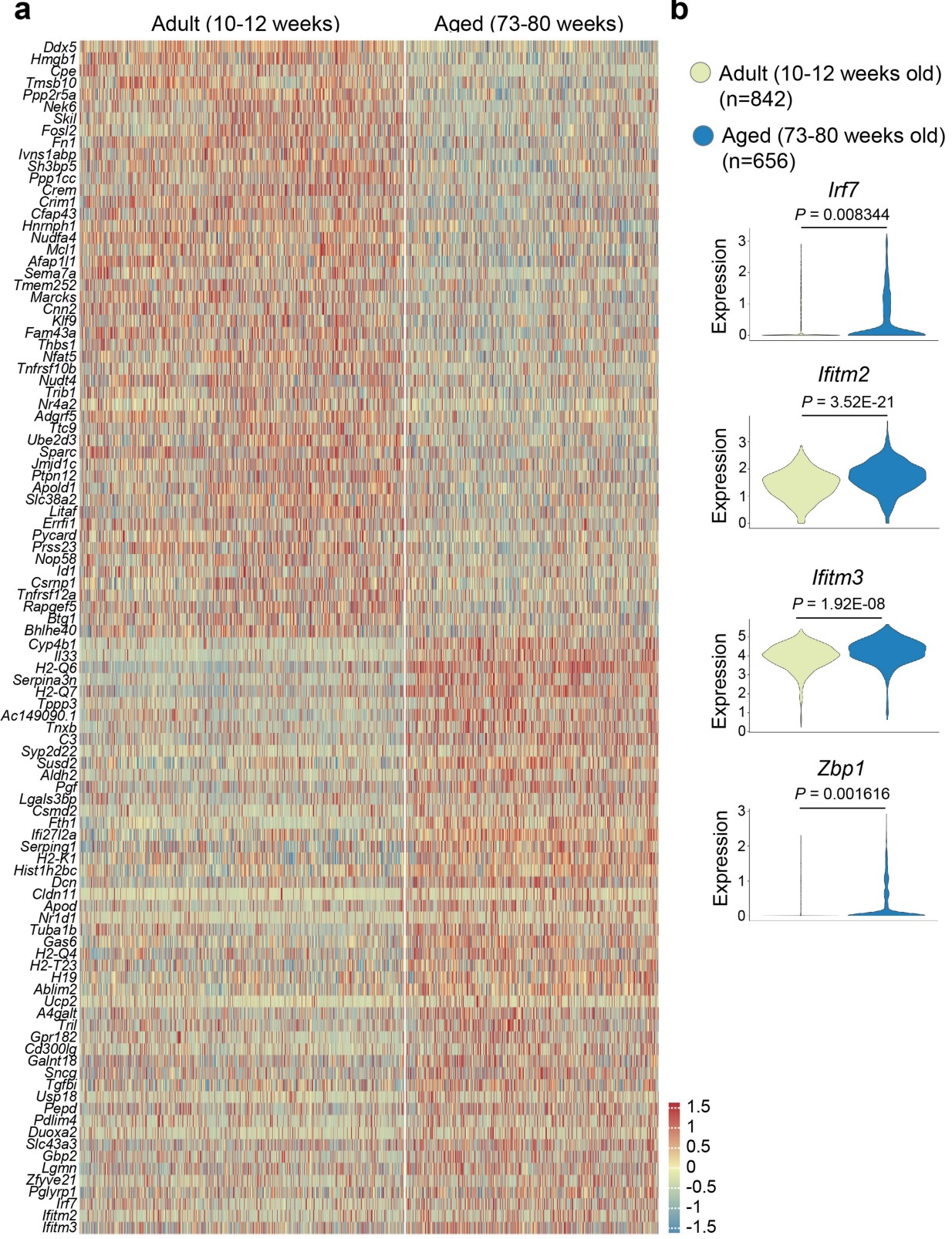

**Extended Data Fig. 14 | Ageing-related transcriptomic changes in lymphatic endothelial cells of nasopharyngeal lymphatic plexus. a**, Heatmap showing 50 differentially expressed genes in LECs of the nasopharyngeal mucosa of adult (10–12 weeks old) and aged (73–80 weeks old) mice. The total number of LECs analysed is 1,498. **b**, Violin plots showing as examples seven genes that were differentially expressed in adult and aged mice. Aged mice has higher expression of genes *Irf7, Ifitm2, Ifitm3*, and *Zbp1* involved in the type I interferon response. *P* values were calculated by two-tailed MAST with Bonferroni *post hoc* test or two-tailed Wilcoxon rank test.

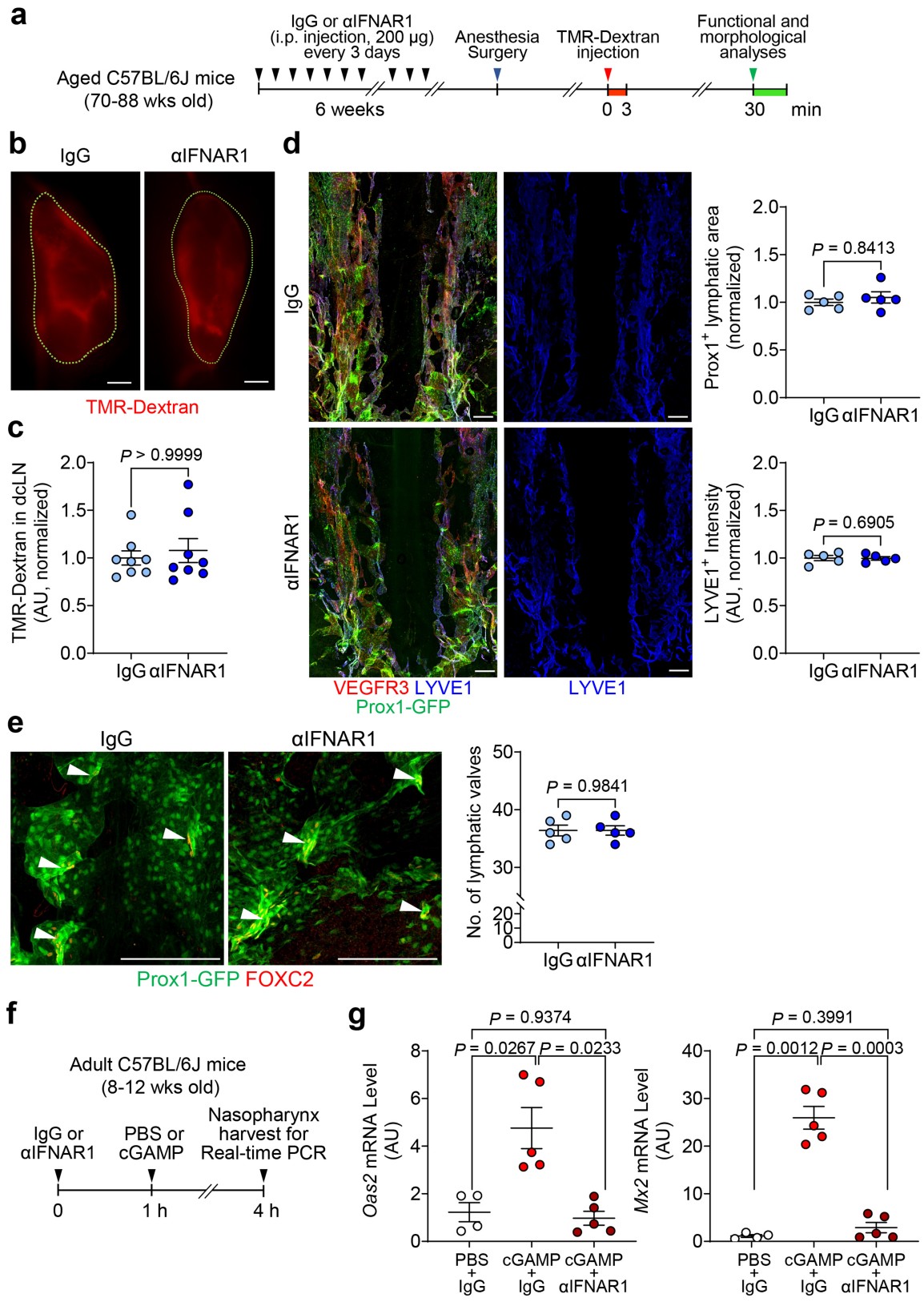

**Extended Data Fig. 15** | See next page for caption.

**Extended Data Fig. 15 | Lack of reversal of ageing-related regression of nasopharyngeal lymphatic plexus (NPLP) and reduced CSF drainage by inhibition of interferon type I signalling for 6 weeks. a**, Diagram of the experimental sequence for intraperitoneal (i.p.) injection of 200 µg of IgG or αIFNAR1 into aged (70-88 weeks old) C57BL/6J or Prox1-GFP mice every 72 hr for 6 weeks followed by intracisternal (i.c.) infusion of TMR-Dextran at 1.0 µl/min for 3 min and measurement of TMR-Dextran fluorescence in deep cervical lymph nodes (dcLN) 30 min later. **b,c**, Fluorescence microscopic images comparing amount of TMR-Dextran fluorescence in dcLNs (outlined by green dashed lines) after IgG or αIFNAR1. Scale bars, 200 µm. Each dot is the value for one mouse; n = 8 mice/group in two independent experiments. Error bars indicate mean ± s.e.m. AU, arbitrary unit. AU (arbitrary unit) normalized to mean IgG value. *P* values were calculated by two-tailed Mann-Whitney test. **d,e**, Immunofluorescence images of nasopharyngeal plexus whole mounts stained for Prox1-GFP, VEGFR3, and LYVE1 after IgG or αIFNAR1 administered to aged mice for 6 weeks. Scale bars, 200 µm. No differences between the groups are evident in lymphatic area, LYVE1 intensity, or lymphatic valves. Each dot is the value for one mouse; n = 5 mice/group from two independent experiments. Bars indicate mean ± s.e.m. AU, arbitrary unit. AU (arbitrary unit) normalized to mean control value = 1.0. *P* values were calculated by two-tailed Mann-Whitney test. **f,g**, Diagram of the experimental sequence for validating inhibition of interferon type I signalling by i.p. injection of 200 µg of αIFNAR1 antibody into adult (8-12 weeks old) C57BL/6J mice. One hour after injection of IgG or αIFNAR1 antibody, 300 µg of cGAMP was injected i.p. to stimulate type I interferon signalling. At 4 hr, nasopharyngeal tissue was removed for real-time PCR analysis of interferon-stimulating genes, *Oas2* and *Mx2*. The blocking antibody reduced *Oas2* and *Mx2* expression to the PBS baseline. Each dot is the value for one mouse; *Oas2* mRNA level and *Mx2* mRNA level (PBS+IgG, n = 4; cGAMP+IgG, n = 5; cGAMP+ αIFNAR1, n = 5) in two independent experiments. Error bars indicate mean ± s.e.m. AU, arbitrary unit. *P* values were calculated by Brown-Forsythe ANOVA and two-tailed Dunnett's T3 multiple comparison test.

# Reporting Summary

## Statistics

For all statistical analyses, confirm that the following items are present in the figure legend, table legend, main text, or Methods section.

| n/a | Confirmed | |
|---|---|---|
| ☐ | ☒ | The exact sample size (*n*) for each experimental group/condition, given as a discrete number and unit of measurement |
| ☐ | ☒ | A statement on whether measurements were taken from distinct samples or whether the same sample was measured repeatedly |
| ☐ | ☒ | The statistical test(s) used AND whether they are one- or two-sided<br>*Only common tests should be described solely by name; describe more complex techniques in the Methods section.* |
| ☐ | ☒ | A description of all covariates tested |
| ☐ | ☒ | A description of any assumptions or corrections, such as tests of normality and adjustment for multiple comparisons |
| ☐ | ☒ | A full description of the statistical parameters including central tendency (e.g. means) or other basic estimates (e.g. regression coefficient) AND variation (e.g. standard deviation) or associated estimates of uncertainty (e.g. confidence intervals) |
| ☐ | ☒ | For null hypothesis testing, the test statistic (e.g. *F*, *t*, *r*) with confidence intervals, effect sizes, degrees of freedom and *P* value noted<br>*Give P values as exact values whenever suitable.* |
| ☒ | ☐ | For Bayesian analysis, information on the choice of priors and Markov chain Monte Carlo settings |
| ☒ | ☐ | For hierarchical and complex designs, identification of the appropriate level for tests and full reporting of outcomes |
| ☒ | ☐ | Estimates of effect sizes (e.g. Cohen's *d*, Pearson's *r*), indicating how they were calculated |

*Our web collection on statistics for biologists contains articles on many of the points above.*

## Software and code

Policy information about availability of computer code

| Data collection | The following softwares were used for data collection:<br>LSM image software (Carl Zeiss)<br>Zen 2.3 software (Carl Zeiss)<br>Labview (National Instruments) |
|---|---|
| Data analysis | The following softwares were used for data analysis:<br>Zen 2.3 software (Carl Zeiss)<br>ImageJ software (NIH)<br>Graphpad Prism10 (GraphPad Software, version 10.1.0)<br>STAR (version 2.7.9a)<br>featureCounts (version 2.0.1)<br>R: The R Project for Statistical Computing<br>R: package 'Seurat' (version 4.1.0)<br><br>Code availability: The codes used for scRNA-seq are available at https://zenodo.org/records/10115336. The LabVIEW program used for pressure and diameter data collection of isolated lymphatic vessels is available at https://doi.org/10.5281/zenodo.8286107. The LabVIEW program used for diameter tracking of isolated lymphatic vessels is available online at https://doi.org/10.5281/zenodo.8286119. |

For manuscripts utilizing custom algorithms or software that are central to the research but not yet described in published literature, software must be made available to editors and reviewers. We strongly encourage code deposition in a community repository (e.g. GitHub). See the Nature Portfolio guidelines for submitting code & software for further information.

## Data

Policy information about availability of data

All manuscripts must include a data availability statement. This statement should provide the following information, where applicable:
- Accession codes, unique identifiers, or web links for publicly available datasets
- A description of any restrictions on data availability
- For clinical datasets or third party data, please ensure that the statement adheres to our policy

The scRNA-seq data of this study are available in the NCBI's Gene Expression Omnibus under accession codes GSE227311 (adult mice) and GSE227324 (aged mice). All other data supporting the findings in this study are available within the paper and its Supplementary Information. Source data are provided with this paper.

## Research involving human participants, their data, or biological material

Policy information about studies with human participants or human data. See also policy information about sex, gender (identity/presentation), and sexual orientation and race, ethnicity and racism.

| | |
|---|---|
| Reporting on sex and gender | Not applicable |
| Reporting on race, ethnicity, or other socially relevant groupings | Not applicable |
| Population characteristics | Not applicable |
| Recruitment | Not applicable |
| Ethics oversight | Not applicable |

Note that full information on the approval of the study protocol must also be provided in the manuscript.

# Field-specific reporting

Please select the one below that is the best fit for your research. If you are not sure, read the appropriate sections before making your selection.

☒ Life sciences ☐ Behavioural & social sciences ☐ Ecological, evolutionary & environmental sciences

For a reference copy of the document with all sections, see nature.com/documents/nr-reporting-summary-flat.pdf

# Life sciences study design

All studies must disclose on these points even when the disclosure is negative.

| | |
|---|---|
| Sample size | Sample size were chosen on the basis of standard power (with α = 0.05 and power of 0.8) performed for similar experiments and no statistical methods were used to predetermine sample size as previously published. |
| Data exclusions | No samples were excluded from the analysis. |
| Replication | Experiments were replicated at least once for all analyses and number of reproductions of each experimental findings is described in each figure legend. All attempts at experimental replication were successful. |
| Randomization | Animal from different cages, but within the same experimental group, were selected to assure randomization. |
| Blinding | The investigators were blinded during the experiments and quantifications. |

# Reporting for specific materials, systems and methods

We require information from authors about some types of materials, experimental systems and methods used in many studies. Here, indicate whether each material, system or method listed is relevant to your study. If you are not sure if a list item applies to your research, read the appropriate section before selecting a response.

## Materials & experimental systems

| n/a | Involved in the study |
|---|---|
| ☐ | ☒ Antibodies |
| ☒ | ☐ Eukaryotic cell lines |
| ☒ | ☐ Palaeontology and archaeology |
| ☐ | ☒ Animals and other organisms |
| ☒ | ☐ Clinical data |
| ☒ | ☐ Dual use research of concern |
| ☒ | ☐ Plants |

## Methods

| n/a | Involved in the study |
|---|---|
| ☒ | ☐ ChIP-seq |
| ☒ | ☐ Flow cytometry |
| ☒ | ☐ MRI-based neuroimaging |

## Antibodies

| | |
|---|---|
| Antibodies used | Primary antibodies used were: anti-LYVE1 (rabbit polyclonal, 11-034, Angiobio); anti-CD31 (hamster monoclonal, clone 2H8, MAB1398Z, Merck); anti-VE-cadherin (goat polyclonal, AF1002, R&D); anti-VEGFR3 (goat polyclonal, AF743, R&D); anti-αSMA-Cy3 (mouse monoclonal, clone 1A4, C6198, Sigma); anti-β3 tubulin (mouse monoclonal, clone 2G10, ab78078, abcam); anti-Foxc2 (sheep polyclonal, AF6989, R&D); anti-LYVE1 (rabbit polyclonal, DP3500, OriGene); anti-collagen type IV (goat polyclonal, AB769, Merck); anti-Laminin α5 (rabbit polyclonal, EWL004, kerafast); anti-tyrosine hydroxylase (rabbit polyclonal, AB152, Merck), anti-vesicular acetylcholine transporter (goat polyclonal, ABN100, Merck); anti-phospho-tau (mouse monoclonal, clone AT8, MN1020, Thermo); anti-mannose receptor (CD206, rabbit polyclonal antibody, ab64693, abcam); anti-Ptx3 (rabbit polyclonal antibody, ALX-210-365-C050, Enzo Life Sciences), phycoerythrin/Cy7 anti-mouse CD326 (rat monoclonal, Ep-CAM, clone G8.8, 118216, BioLegend) antibody, APC anti-mouse podoplanin antibody (syrian hamster monoclonal, clone 8.1.1, 127410, BioLegend), and phycoerythrin-labeled anti-mouse CD31 antibody (rat monoclonal, clone MEC13.3, 102508, BioLegend). The following secondary antibodies were used: Alexa Fluor™ 488-, 594- and 647- conjugated anti-rabbit (711-545-152, 711-585-152, 711-605-152), anti-goat (705-585-147), anti-sheep (713-585-147), anti-hamster (127-605-160) secondary antibodies (Jackson ImmunoResearch). Nuclei were stained with DPAI (H-1200, Vector). Primary antibodies were diluted at 1:200 or 1:400 and secondary antibodies were diluted at 1:100 or 1:1000 for all the immunostainings or lymphatic endothelial cell sorting. |
| Validation | All the antibodies were validated for the species (mouse or monkey) and applications (immunohistochemistry or lymphatic endothelial cell sorting) by the correspondent manufacturer, which is described in the manufacturer's website. Our usage was described in the Methods section of the manuscript as below.<br><br>Samples were incubated in 5% normal donkey serum (017-000-121, Jackson ImmunoResearch) for 1 h at RT. Then, they were incubated with primary antibodies (1:400) dissolved in 5% normal donkey serum at 4°C for 12 h. After washing in PBS, they were incubated with secondary antibodies (1:1000) dissolved in 5% normal donkey serum at 4°C for 12 h. The samples that had been the clearing and decalcification were incubated with donkey serum for 24 h at RT, primary antibodies at 1:200 dilution at RT for 10 days, and secondary antibodies at 1:100 dilution at RT for 3 days. After PBS washing, the samples were covered with DAPI-containing mounting medium (H1200, Vector) or refractive index matching solution (D-PROTOSS). |

## Animals and other research organisms

Policy information about studies involving animals; ARRIVE guidelines recommended for reporting animal research, and Sex and Gender in Research

| | |
|---|---|
| Laboratory animals | Prox1-GFP mice (8 weeks to 12 weeks old for adult, 73 weeks to 102 weeks old for aged, Choi et al., Blood, 2011; provided by Dr. Young-Kwon Hong, University of Southern California) were transferred, established, and bred in SPF animal facilities at KAIST under a 12 h-12 h light-dark cycle at 23-24 °C and 40-60 % humidity. C57BL/6J mice (8 weeks to 12 weeks old) were purchased from DBL (Soul, Korea). Aged C57BL/6J mice (73 weeks to 102 weeks old) were purchased from Animal Center of Ageing Science of Korea Basic Science Institute (Gwangju, Korea) or from JAX (USA). The head and neck portions of the primate (Macaca fascicularis, 6-13 years old) were obtained from the National Primates Center of KRIBB. |
| Wild animals | The study did not involve wild animals. |
| Reporting on sex | In this study, we randomized both genders were selected and used because gender was not a major criteria serve the purpose of this study. |
| Field-collected samples | The study did not involve samples collected from the field. |
| Ethics oversight | All animal care and experimental procedures were approved by Institutional Animal Care and Use Committees of the Korea Advanced Institute of Science and Technology (KAIST) (KA2023-014-v1) and the University of Missouri (9797) for the mice and the Korea Research Institute of Bioscience and Biotechnology (KRIBB-AEC-22237) for the primates. |

Note that full information on the approval of the study protocol must also be provided in the manuscript.

