## [Peer Review File · Nature]

Manuscript Title: Nasopharyngeal lymphatic plexus is a hub for cerebrospinal fluid drainage

Reviewer Comments & Author Rebuttals

Reviewer Reports on the Initial Version:

Referees' comments:

Referee #1 (Remarks to the Author):

In the manuscript submitted by Yoon et al, the authors have utilized a combination of intravital imaging, whole-mount stainings of tissues and ex vivo analysis of lymphatic contractility to characterize what they have termed the nasopharyngeal lymphatic plexus (NPLP) and its connections to the deep cervical lymphatic vessels and nodes. The authors, with support from previously published studies, have convincingly demonstrated that this network is a key hub for cerebrospinal fluid drainage. This study, thus, adds another piece of the puzzle to the complicated and somewhat controversial topic of CSF clearance routes to the lymphatic system.

The authors have presented beautiful images and videos of truly impressive whole-mounted samples demonstrating the plexus of lymphatic vessels at this location. The authors have also mapped three potential upstream pathways for CSF drainage that converge on this network and have shown that the downstream vessels are bona fide lymphatic collectors draining into the medial afferent lymphatics of the deep cervical lymph nodes. This is the strongest part of the manuscript and has given an unprecedented view of a lymphatic vessel network that has been difficult to access and visualize using previous techniques. The authors then performed technically-challenging studies involving ligation of medial vs. lateral cervical lymphatic vessels to provide evidence that the vessels draining from the medial network appear to drain more CSF tracer to the deep cervical lymph nodes than the vessels draining from the lateral aspect.

The authors then demonstrate that this network appears to exhibit signs of regression, while the collecting vessel network remains intact, in aged mice. The authors also show that both young and aged collecting vessels are responsive to known pharmacological modulators of lymphatic tone. They suggest based upon these findings that the collecting vessels draining this lymphatic plexus may represent a target for enhancing CSF efflux in aging or neurodegenerative conditions.

Specific concerns:

1. The authors propose to regulate the pumping or contractility of the collecting lymphatic vessels near the deep cervical lymph nodes to assist in the clearance of CSF during aging or neurodegenerative conditions. This would indeed represent a novel target, and setting aside the obvious difficulty one might encounter in specifically targeting this part of the lymphatic system in the clinic, it is unclear exactly why the authors have focused on this. No differences were seen in the morphology (or function) of the lymphatic vasculature of this location in aged mice, unlike the differences that were seen at the NPLP. Why haven't the authors utilized the AAV-mVEGFC approach

to attempt to reestablish the function of a network that had regressed during the aging process? Furthermore, there is still no convincing evidence in the literature or within this study that the decline in CSF outflow that has been observed in several studies in aged mice is truly an issue in impaired lymphatic efflux, rather than a general aging-related decline in CSF production or circulation.

2. The authors appear to have utilized the diameter or intensity changes of the medial cervical lymphatics as a surrogate measure for CSF outflow. For example, in the discussion (pg 13, lines 355-356) the authors write: "CSF outflow through the medial route was reduced by alpha-adrenergic activation and increased by NO signaling in lymphatic smooth muscle." There are issues with such an approach. As the authors are surely aware, lymphatic transport in collecting vessels is complex and involves a combination of tonic and phasic regulation of a chain of lymphangions in the collecting lymphatics to transport lymph fluid against increasing pressure back to the systemic circulation. A simple increase in diameter may allow more flow if pressure gradients become favorable but it may also reflect a lymph stasis. The authors would either need dynamic tracking of microparticles within the vessel or an indicator of accumulation of the tracer (e.g. in the downstream deep cervical lymph nodes as utilized in Figure 8) to show that the CSF tracer has indeed been transported to a greater or lesser degree by the treatments.

3. On a related note, the authors repeatedly refer to activation of lymphatic pumping in statements such as "pharmacologic activation of deep cervical lymphatic pumping toward lymph nodes can still increase CSF outflow" (summary, line 50-51) and "CSF outflow could still be increased by pharmacological activation of pumping of the deep cervical lymphatics." (background, line 100-101). This is contradictory to the above claim that NO induced dilation would lead to more CSF transport. Without a more direct measurement of CSF outflow, it is difficult to estimate whether an approach stimulating lymphatic contractions (such as with alpha-adrenergic agonist) or lymphatic dilation (such as with an NO donor) would be more favorable for an overall increase in CSF efflux. It might be context-dependent depending on the specific pathology of interest.

4. The authors claim that the lateral network of deep cervical lymphatic vessels is originating from the jugular foramen and thus represents drainage of the basolateral dural lymphatic vessels. Could the authors provide images that support this claim? The scheme shown in Extended Data Figure 8 appears to contradict this. Here the basolateral network along the medial meningeal artery and petrosquamous sinus appears to be connected to the lymphatics that exit at the "U Lymphatic-2" site.

5. It is not clear how the RNA sequencing data fits into the story other than providing some insights on a few inflammatory factors that might be altered in aged lymphatic vessels. The authors have not incorporated this data to generate specific diagnostic or therapeutic targets.

Minor comments:

The authors used decalcified tissue, cleared tissue and whole mounted tissue, but have not made it clear which images are taken from each type of processed tissue.

The authors have used the term "intravital imaging" (page 4, line 92) to define what is for the most part results obtained from post-mortem imaging, this is misleading.

How do the authors differentiate between the “lymphatic layer” and the NPLP? Is this based on anatomical location or morphology? The layer appears for the most part like a sheet-like lymphatic network, however, for example in Extended Data Fig. 8, the NPLP appears more sheet-like.

The quantifications of the aged tissue shown in Figure 6 are somewhat confusing. The authors state that the LYVE1 staining is 164% greater in aged lymphatics (I believe this should read 64% greater based in the numbers shown). What is the significance of this increased LYVE-1 staining? The zoomed in areas appear to show what appears to be a loss of LYVE1 staining in the vessels rather than an overall increase as apparent in the overview.

The authors stated that “FluoSpheres remained in place during tissue processing”, however, this doesn’t appear to be entirely true. In Extended Data 5-7 the Fluospheres appear to be present in many unexpected locations including within bone of the skull or bone marrow. Was this an artifact of the tissue sectioning?

The authors claim (page 3, line 86-87) that their previous MRI study revealed CSF-containing contrast agent in dural lymphatics at the base of the skull. Due to resolution limitations, it would be impossible to visualize this with MRI. Please rephrase.

Please specify at which level the coronal section view in Figure 1 is taken from. Would this represent an area containing the lymphatic layer or the NPLP?

The authors mention in the limitations section of the discussion that the NPLP “could not be selectively ablated in mice in loss-of-function studies performed to determine the adaptability of other routes for CSF clearance.” Did the authors attempt these studies? It is unclear from this sentence.

The authors have not mentioned the superficial cervical lymph nodes and the associated lymphatic network which has also been shown in several previous studies to drain CSF. Does this network receive any drainage from the NPLP?

Referee #2 (Remarks to the Author):

“The nasopharyngeal lymphatic plexus is a hub for cerebrospinal fluid drainage and is regressed with ageing”

By Yoon et al

Recent studies have highlighted the importance of lymphatic cerebral spinal fluid (CSF) drainage in health and disease states. It is known that tracers injected into the CSF are found in deep cervical lymph nodes, indicating that CSF drains via lymphatic routes. However, the details of these drainage routes between the base of the brain and lymph nodes remains unclear. Here the authors describe the anatomical structure of nasopharyngeal lymphatic vasculature and its connections to collecting points around the base of the brain in adult (8-12 week old) mice. The authors note unusually

shaped valves in some places in the nasopharyngeal lymphatic vasculature, and lack of smooth muscle marker SMA around these lymphatic vessels where they are closest to the brain. They confirm that tracers injected into cisterna magna or hippocampus appear in the nasopharyngeal lymphatic vasculature and deep cervical lymph nodes at various time points after injection and that medial lymphatic vessels contribute more to clearance of the observed tracers to the lymph nodes. Topical application of phenylephrine or NO donor sodium NONOate resulted in constriction or dilation of the nasopharyngeal lymphatic vessels, respectively, indicating functional control of CSF drainage. AAV delivery of VEGFC to the lymphatic vasculature resulted in increases in lymphatic density and increased drainage of CSF tracers. In aged mice, nasopharyngeal lymphatic vasculature was less dense, or “atrophied” as the authors called it, but was otherwise functional when compared to younger adult mice. Finally, scRNAseq transcriptomic analysis indicated that nasopharyngeal lymphatics in aged mice exhibited increased inflammatory and apoptotic markers compared to the adult mice.

Overall, through a set of technically challenging experiments and observations, this manuscript provides a careful and detailed description of nasopharyngeal lymphatics in the mouse CNS, including a description of changes with aging, and includes a useful transcriptomic dataset. Specific comments are below.

Comments:

1. Since the authors use a method that significantly alters blood flow during the surgery to expose the nasopharyngeal lymphatics and lymph nodes, could this affect the CSF and tracer flow through the lymphatic vessels? Blood flow is also thought to contribute to CSF movements in the brain.
2. Could the observed differences in medial vs. lateral lymphatic CSF tracer drainage be due in some part to differences in CSF tracer concentrations where these vessels are most proximal to the brain, where their collection points would presumably be? See for example the tracer distribution in figure 2b.
3. The authors need to better explain the advantage and/or reasoning of intrahippocampal tracer injections vs. inter cisterna magna injections for the experiments performed.
4. The authors should comment on the possible causes/mechanism for detachment of lymphatic endothelial cells observed in the aged mice.

Minor:

5. In general, it is difficult to distinguish which datapoints are significantly different from each other. P-value information are given in the figure captions, but this does not indicate what conditions and data sets/points this reflects. Please include more information in the figures.
6. Figure 1a: what is the significance of the dashed green lines in the lateral view?
7. Figure 6a: Contrast adjustments are needed to better reflect the results. For instance, the LYVE-1 staining in the adult mouse top panel is very dim compared to the aged mice, making comparison difficult. Similarly, VEGFR3 staining is difficult to see in the top panels. In the lower panels, the

magnification is too low to get a clear view of the features pointed out by the arrows. Higher magnification images would be helpful, as well as images that better illustrate the detached lymphatic endothelial cells and valve structures. As presented, it is not obvious what the arrowheads are pointing out.

8. Figure captions: Why is “ $p < 0.05$ ” listed in the same sentences that indicate no significant differences?

9. Please provide statistical information for figure 9d.

10. Extended data figure 7c, magnified view: It is not clear that the fluorospheres are inside the lymphatic vessels. It looks like the spheres are non-specifically distributed in the region shown.

11. Extended data Figure 9: A diagram with orientation information for the views presented (particularly in panel b) would be helpful.

12. Extended data figure 10b: The images presented for the SSS could be improved. There is little mCherry labeling in both mVEGFC and control conditions, and it looks like more in the mCherry controls.

Referee #3 (Remarks to the Author):

Yoon et al. have produced a spectacular piece of work exploring the nasopharyngeal lymphatic plexus, a still concealed part of the lymphatic system associated with the central nervous system (CNS). The nasopharyngeal lymphatic plexus lays between the skull base and the palatine bone, collecting the cerebrospinal fluid (CSF) outflow from the skull and draining it towards the cervical lymph nodes which are the main brain-draining lymph nodes.

Here, the authors show that the nasopharyngeal lymphatic plexus is the main extra-cranial route for CSF outflow. The nasopharyngeal lymphatic plexus collects the lymph from multiple meningeal lymphatic circuits and connects with the medial lymphatic vessels that provide a main entry to CSF into the cervical lymph nodes. Considering that the clearance and immune surveillance of CNS tissues require an efficient drainage of macromolecules and immune cells transported by the CSF and that both processes are altered in many neurological disorders, new knowledge about the anatomy and functional properties of the CNS lymphatic circuit has strong potential regarding the treatment of neurological diseases and cognitive aging. The present work aligns well this objective with two main contributions. First, it demonstrates that CSF drainage can be modulated by targeting the extra-cranial lymphatic system which is impaired but still functional in aged mice. Second, it shows that adrenergic/nitric oxide (NO) signaling drugs regulate the contraction/dilatation of extra-cranial lymphatic collectors, respectively, and can be used to stimulate CSF lymphatic drainage, as an alternative strategy to the so far unique approach by VEGF-C (vascular endothelium growth factor - C) delivery.

The authors performed a technical tour de force by expert dissections allowing them to access and

provide high resolution imaging of this hardly accessible lymphatic bed. They extensively characterized and beautifully illustrated the nasopharyngeal lymphatic plexus and its downstream cervical collectors by combining IHC techniques in adult and aged mice as well as in non-human primates (*Macaca*) with cervical lymphatic ligation experiments and fluorescent tracer injection into the CSF/CNS. Ex vivo studies on isolated cervical lymphatics brought additional important information on the elastic properties of these collectors. A difficult experiment of peri-mortem intravital imaging after intra-cisternal injection of TMR-Dextran allowed the authors to further demonstrate that CSF outflow via medial cervical lymphatics was reduced by alpha-adrenergic activation and increased by NO signaling in the lymphatic smooth muscle that surround medial lymphatics. A final single cell analysis of nasopharyngeal lymphatics in adult and aged is completing the study, showing that the adult plexus includes more Ptx3+ capillaries with immune interplay than other organs and that pro-apoptotic gene expression as well as inflammation and type I interferon signaling globally increase with age in plexus endothelial cells.

Altogether the authors have generated a spectacular and convincing set of outstanding interest data. However, several questions should be further investigated.

I. First, we would like to know whether deep cervical lymphatics can be directly targeted in living mice and if their pharmacological manipulation can delay glymphatic drainage and behavioral alterations during aging. This investigation requires to overcome technical challenges, as to access the nasopharyngeal plexus or cervical lymphatics in living animals, for example in Alzheimer's disease models such as 5XFAD mice but would indisputably raise the impact of the study by providing evidence that triggering extra-cranial lymphatics allows to delay cognitive aging.

II. Another investigation that I would suggest relates to the inflammatory phenotype detected by scRNA-Seq in medial lymphatics in aging mice and would test if the adult phenotype could be rescued by antagonistic drugs against anti-type I IFN signaling (sifalimumab or anifrolumab) and if this treatment may eventually improve CSFG drainage.

III. From the 2D images shown in extended data Fig 7c, it is hard to be convinced that meningeal and olfactory epithelium lymphatics form a continuous network. I suggest the authors to perform additional LYVE1 and Ptx3 labeling on cribriform plate sections (coronal, sagittal and horizontal) from Prox1-GFP mice and reconstruct the 3D-lymphatic network in this region. Why are there no FluoSpheres detected in the olfactory epithelium on the left panel? On the right panel, most of FluoSpheres are located outside Prox1 vessels and appear to rather follow nerve branches through cribriform pores. From personal experience, I learned how much misleading can be 2D-scans through the irregular surfaces of the cribriform plate and would suggest the author to be cautious on any statement regarding the connection between these circuits.

The quality of their manuscript may also be improved by addressing some minor points outlined below.

Minor points:

1. Fig 1a. The anterior lymphatic layer is hard to visualize, as well as the LYVE1 pattern on the ventral view and the VEGFR3 pattern. There is enough space available to show each marker in a separated panel. There is also no legend for the dotted green line indicating the nasopharyngeal cavities.
2. Fig1b. A three-dimensional representation of the inverted saddle model should replace the present diagram.

3. Fig 3g. The present image of beta3tubulin labeling reiterates the message of Fig 3f and could be replaced by pictures of Tyrosine hydroxylase (TH) and choline acetyltransferase (ChAT) labeling that would inform on the adrenergic/cholinergic innervation of cervical lymphatics.
4. Figs 2, 3, 4 and 5. In all experiments, the authors intentionally injected 10kDa Dextran to monitor CSF drainage. Would they use higher molecular weight dextran (60-70kDa) and later time points for imaging, they could also track tracer-carrier phagocytes within extra-cranial lymphatics. This could be performed and associated with anti-CD45 or CD11b/c labeling of myeloid carrier cells, without quantification, after intra-CNS injection to demonstrate the traffic of brain tracer-carrier phagocytes within extra-cranial lymphatics close to cervical lymph nodes and how much time it takes to be detected.
5. Fig 7d, f, g. Indicate statistical values with either numbers or stars on the figures. Also simplify the corresponding legend.
6. Fig 9. Ptx3 expression by nasopharyngeal lymphatics should be assessed by IHC.
7. Extended data Fig 1. A 3D-reconstruction of the nasopharyngeal plexus could be generated from the set of dorsal-ventral-lateral views collected by the authors. The quality of generated data is calling for such an integrated representation.
8. Extended data Fig 8 and corresponding text (p 7) deserve discussion with respect to previous studies describing basal and ventral connections between meningeal and extra-cranial lymphatic circuits (PMID: 35776089) as well as on the main importance of nasopharyngeal route for CSF lymphatic drainage (PMID: 18972423, PMID: 34905509, PMID: 35776089 Fig. 5 F and Fig. S3, E and F).
9. Text lines 376-378. Please quote previous studies on spinal lymphatic drainage of CSF (PMID: 31455602, PMID: 31597914).
10. In the M&M section, p.20 line 592 the authors mention the use of light sheet fluorescence microscopy (LSFM) but there is no corresponding images or data. Please delete or provide LSFM data.

Referee #4 (Remarks to the Author):

This is a meticulous and elegant study filling an important knowledge gap in our understanding of the pathways of cerebrospinal fluid drainage to the deep cervical lymph nodes. Defining these anatomical routes is currently a matter of lively and controversial debate in the field. The present study from the laboratories of Dr. Davis and Dr. Koh significantly advances the field by identifying the nasopharyngeal lymphatic plexus as a major “hub” for CSF outflow from lymphatics from the cribriform plate but also additional intracranial regions towards the deep cervical lymph nodes. The study provides an in depth analysis of the cellular and molecular makeup of the lymphatics of the nasopharyngeal lymphatic plexus and how they change with age. Furthermore, the authors explored if the regressing nasopharyngeal lymphatic plexus could be pharmacologically targeted for potential therapeutic enhancement of CSF drainage to deep cervical lymph nodes in neurological disorders.

This is an original study with exceptional relevance to our understanding of afferent pathways of CNS immunity. The authors have employed state of the art methodology that is illustrated in each figure where necessary in detail and have referred to the shortcomings of studying CSF drainage pathways in anesthetized mice. I have not identified major issues and have only some minor

comments the authors may wish to consider.

Figure 8 d and e compare the tracer accumulation in deep cervical lymph nodes of young and aged mice. It would be informative to understand if the entire architecture of these aged lymph nodes has changed such that tracers would fail to accumulate.

CSF drainage has in addition been observed to occur towards the superficial cervical or mandibular lymph nodes. It would be instructive if the authors did observe tracer drainage only to the deep cervical lymph nodes.

The authors refer to the lateral afferent dc lymph node lymphatics to those connecting to the basolateral lymphatics through the jugular foramen. How do they envisage CSF drainage to dc lymph nodes from the dorsal dural lymphatics?

In the discussion the authors highlight that modification of dural lymphatics may by enhancing immune cell migration from the CNS to the deep cervical lymph nodes ameliorate experimental autoimmune encephalomyelitis (EAE) citing a study from the Kipnis laboratory. However recent studies by the laboratories of Kari Alitalo (doi: 10.1172/JCI130308) and Odoardi/Flügel (doi: 10.1038/s41593-022-01108-3) have questioned these findings. As EAE generally serves as a model for multiple sclerosis the authors may wish to rephrase this statement.

For the interested readers it would further be instructive if the authors included discussion of the recent findings by the Proulx laboratory on CSF outflow pathways at the cribriform plate (doi: 10.1016/j.ebiom.2023.104558) in the context of their findings.

In several figures arrowheads referred to are missing, and scale bars should be double checked. Figure 1 a – lateral view includes dashed lines not mentioned etc.

Author Rebuttals to Initial Comments:

Referees' comments:

Referee #1 (Remarks to the Author)

In the manuscript submitted by Yoon et al, the authors have utilized a combination of intravital imaging, whole-mount stainings of tissues and ex vivo analysis of lymphatic contractility to characterize what they have termed the nasopharyngeal lymphatic plexus (NPLP) and its connections to the deep cervical lymphatic vessels and nodes. The authors, with support from previously published studies, have convincingly demonstrated that this network is a key hub for cerebrospinal fluid drainage. This study, thus, adds another piece of the puzzle to the complicated and somewhat controversial topic of CSF clearance routes to the lymphatic system.

The authors have presented beautiful images and videos of truly impressive whole-mounted samples demonstrating the plexus of lymphatic vessels at this location. The authors have also mapped three potential upstream pathways for CSF drainage that converge on this network and have shown that the downstream vessels are bona fide lymphatic collectors draining into the medial afferent lymphatics of the deep cervical lymph nodes. This is the strongest part of the manuscript and has given an unprecedented view of a lymphatic vessel network that has been difficult to access and visualize using previous techniques. The authors then performed technically-challenging studies involving ligation of medial vs. lateral cervical lymphatic vessels to provide evidence that the vessels draining from the medial network appear to drain more CSF tracer to the deep cervical lymph nodes than the vessels draining from the lateral aspect.

The authors then demonstrate that this network appears to exhibit signs of regression, while the collecting vessel network remains intact, in aged mice. The authors also show that both young and aged collecting vessels are responsive to known pharmacological modulators of lymphatic tone. They suggest based upon these findings that the collecting vessels draining this lymphatic plexus may represent a target for enhancing CSF efflux in aging or neurodegenerative conditions.

Response: We thank the reviewer for the favorable comments and for the very careful, thorough, and insightful review.

Specific concerns:

1. The authors propose to regulate the pumping or contractility of the collecting lymphatic vessels near the deep cervical lymph nodes to assist in the clearance of CSF during aging or neurodegenerative conditions. This would indeed represent a novel target, and setting aside the obvious difficulty one might encounter in specifically targeting this part of the lymphatic system in the clinic, it is unclear exactly why the authors have focused on this.

Response: We now explain more clearly in the Introduction (page 4, lines 101-106) and Results (page 10, lines 273-274; page 10-11, lines 292-299, and Extended Data Fig. 19a-c; page 11, lines 319-322, and Extended Data Fig. 21b,c) the rationale for focusing on deep cervical lymphatics and address this issue in the Discussion (page 15, lines 435-444; page 16, lines 468-482) in the context of the findings and the reviewer's comments.

Deep cervical lymphatics, which carry CSF from the nasopharyngeal lymphatic plexus to the deep cervical lymph nodes, are surrounded by circular smooth muscle and remain unchanged with ageing. Importantly, our new findings show that topical application of a low concentration of phenylephrine or sodium nitroprusside to deep cervical lymphatics increases CSF drainage to deep cervical lymph nodes. These proof-of-concept studies now set the scene for potential clinical applications through innovative drug targeting, delivery, or other methods for promoting CSF transport through deep cervical lymphatics, without resorting to intracranial approaches for increasing CSF drainage.

No differences were seen in the morphology (or function) of the lymphatic vasculature of this location in aged mice, unlike the differences that were seen at the NPLP. Why haven't the authors utilized the AAV-mVEGFC approach to attempt to reestablish the function of a network that had regressed during the aging process?

Response: We performed a new experiment with intracisternal delivery of AAV9-mVEGF-C-mCherry and AAV9-mCherry to test the reversibility of age-related changes in nasopharyngeal lymphatics and CSF outflow to deep cervical lymph nodes. The findings are described in the Results (page 11-12, lines 326-337, and Extended Data Fig. 23a-e).

These new data show that three weeks after viral delivery, CSF outflow to lymphatics in the nasopharyngeal plexus and deep cervical lymph nodes was increased in the AAV9-mVEGF-C group over the AAV9-mCherry control group. However, it is difficult to delineate the bottlenecks along the CSF drainage route that would have to be corrected to increase CSF outflow by the intracisternal delivery of AAV9-mVEGF-C.

Furthermore, there is still no convincing evidence in the literature or within this study that the decline in CSF outflow that has been observed in several studies in aged mice is truly an issue in impaired lymphatic efflux, rather than a general aging-related decline in CSF production or circulation.

Response: We agree and acknowledge in the Introduction (page 3, lines 63-66) that reduced CSF production during ageing could contribute to the decline in CSF outflow. We reinforce this point with additional references in the Discussion (page 17, lines 494-500). Our new data now add to these previous factors the contribution of impaired lymphatic efflux of CSF through extracranial lymphatics to lymph nodes.

2. The authors appear to have utilized the diameter or intensity changes of the medial cervical lymphatics as a surrogate measure for CSF outflow. For example, in the discussion (pg 13, lines 355-356) the authors write: “CSF outflow through the medial route was reduced by alpha-adrenergic activation and increased by NO signaling in lymphatic smooth muscle.” There are issues with such an approach. As the authors are surely aware, lymphatic transport in collecting vessels is complex and involves a combination of tonic and phasic regulation of a chain of lymphangions in the collecting lymphatics to transport lymph fluid against increasing pressure back to the systemic circulation. A simple increase in diameter may allow more flow if pressure gradients become favorable but it may also reflect a lymph stasis. The authors would either need dynamic tracking of microparticles within the vessel or an indicator of accumulation of the tracer (e.g. in the downstream deep cervical lymph nodes as utilized in Figure 8) to show that the CSF tracer has indeed been transported to a greater or lesser degree by the treatments.

Response: Our original intention was to provide evidence that contraction and dilation of the deep cervical lymphatics can be pharmacologically controlled by topical application of an adrenergic agonist and NO donor. This goal led to our initial use of relatively high concentrations of phenylephrine (PE) or sodium nitroprusside (SNP).

Based on the reviewer’s insightful suggestion, we performed new experiments that revealed that a low concentration of PE or SNP significantly increased TMR-Dextran accumulation in deep cervical lymph nodes. These new data are reported in the Results (page 10-11, lines 292-299 and Extended Data Fig. 19a-c), and the limitations and future directions are also described (page 16, lines 475-482).

Here, we would like to mention that the low concentration of PE or SNP did not alter the diameter and TMR-Dextran fluorescence of deep cervical lymphatics visible by fluorescence stereomicroscopy; however, it increased rhythmic contraction or dilation of deep cervical lymphatics (in **Fig 7e**), which could facilitate CSF outflow to deep cervical lymph nodes. The findings reported in the revised manuscript can now be extended by two-photon microscopic imaging and measurement and other analyses in future studies.

3. On a related note, the authors repeatedly refer to activation of lymphatic pumping in statements such as “pharmacologic activation of deep cervical lymphatic pumping toward lymph nodes can still increase CSF outflow” (summary, line 50-51) and “CSF outflow could still be increased by pharmacological activation of pumping of the deep cervical lymphatics.” (background, line 100-101). This is contradictory to the above claim that NO induced dilation would lead to more CSF transport. Without a more direct measurement of CSF outflow, it is difficult to estimate whether an approach stimulating lymphatic contractions (such as with alpha-adrenergic agonist) or lymphatic dilation (such as with an NO donor) would be more favorable for an overall increase in CSF efflux. It might be context-dependent depending on the specific pathology of interest.

Response: This important issue was addressed by replacing the term “pumping” with “transport” throughout the manuscript. It was also addressed in our response to Comment 2. We are delighted to report that we found a decrease or increase of CSF outflow through deep cervical lymphatics after topical application of a narrow concentration range of phenylephrine or sodium nitroprusside. This finding provides proof-of-concept that justifies development of new approaches for enhancing CSF transport through extracranial lymphatics in the treatment of CSF-related diseases.

4. The authors claim that the lateral network of deep cervical lymphatic vessels is originating from the jugular foramen and thus represents drainage of the basolateral dural lymphatic vessels. Could the authors provide images that support this claim? The scheme shown in Extended Data Figure 8 appears to contradict this. Here the basolateral network along the medial meningeal artery and petrosquamous sinus appears to be connected to the lymphatics that exit at the “U Lymphatic-2” site.

Response: We provide new images to support this claim in the Results (page 9, lines 240-242, and Extended Data Fig. 16a-f). The problem was corrected by revising the drawing in Extended Data Fig. 11 to show the CSF flow routes from meningeal lymphatics to lateral deep cervical lymphatics. Abbreviations in Extended Data Fig. 11 and legend were removed to make the drawing easier to understand.

5. It is not clear how the RNA sequencing data fits into the story other than providing some insights on a few inflammatory factors that might be altered in aged lymphatic vessels. The authors have not incorporated this data to generate specific diagnostic or therapeutic targets.

Response: This issue was addressed by explaining the rationale for obtaining scRNA-seq data in the Results (page 11, lines 316-319; page 14, lines 407-419) and considering the relevance of these data to ageing in the Discussion (page 17, lines 508-515).

Minor comments:

1. The authors used decalcified tissue, cleared tissue and whole mounted tissue, but have not made it clear which images are taken from each type of processed tissue.

Response: We solved this problem by describing in the revised Figure legends which tissues were decalcified and cleared for light-sheet fluorescence microscopy (LSFM). The methods used for capturing LSFM images after decalcification and clearing are described in the Methods section.

The Figure legends also describe the processing of tissue whole mounts and sections for images obtained by confocal fluorescence microscopy.

2. The authors have used the term “intravital imaging” (page 4, line 92) to define what is for the most part results obtained from post-mortem imaging, this is misleading.

Response: We agree and addressed the misleading aspect of this term by changing the sentence (page 4, lines 96-98) to “we performed fluorescence imaging of CSF outflow in anesthetized Prox1-GFP mice after surgical exposure of nasopharyngeal and cervical lymphatics. These mice had fluorescent lymphatics from transgenic expression of Prospero-related homeobox 1 (Prox1)-green fluorescence protein (GFP)²⁰.”

3. How do the authors differentiate between the “lymphatic layer” and the NPLP? Is this based on anatomical location or morphology? The layer appears for the most part like a sheet-like lymphatic network, however, for example in Extended Data Fig. 8, the NPLP appears more sheet-like.

Response: We agree that the distinction between “lymphatic layer” and NPLP was not sufficiently clear. New findings revealed that the structure originally called “lymphatic layer”, which is located in the posterior region of the nasal cavity, is a flattened, condensed lymphatic plexus. Therefore, we renamed it “posterior nasal lymphatic plexus” (page 5, lines 114-115) and described its distinctive features in the revised manuscript (page 5, lines 114-117).

4. The quantifications of the aged tissue shown in Figure 6 are somewhat confusing. The authors state that the LYVE1 staining is 164% greater in aged lymphatics (I believe this should read 64% greater based in the numbers shown). What is the significance of this increased LYVE-1 staining? The zoomed in areas appear to show what appears to be a loss of LYVE1 staining in the vessels rather than an overall increase as apparent in the overview.

Response: We appreciate the reviewer finding this error and have corrected the value to 64% greater in the Results (page 11, line 309).

As for the significance, the change in LYVE1 could promote immune cell trafficking to the lymphatics (Jackson DG, Matrix Biology, 2019); however, a more complete understanding of the significance awaits future studies. Although the zoomed area showed disintegrated LECs that lacked LYVE1, overall LYVE1 was higher in aged NPLP (page 11, lines 309-310, and Fig. 6a,b and Extended Data Fig. 20).

5. The authors stated that “FluoSpheres remained in place during tissue processing”, however, this doesn’t appear to be entirely true. In Extended Data 5-7 the Fluospheres appear to be present in many unexpected locations including within bone of the skull or bone marrow. Was this an artifact of the tissue sectioning?

Response: The reviewer correctly points out that some FluoSpheres in images were artifactually scattered during tissue sectioning. This problem was solved by improved

sectioning methods and providing more convincing images and descriptions (page 7, lines 189-195, and Extended Data Fig. 10a-d).

6. The authors claim (page 3, line 86-87) that their previous MRI study revealed CSF-containing contrast agent in dural lymphatics at the base of the skull. Due to resolution limitations, it would be impossible to visualize this with MRI. Please rephrase.

Response: We agree and have corrected this sentence in the Introduction (page 4, lines 89-91) to acknowledge that the identification of these structures in our previous study was limited by the resolution of MRI.

7. Please specify at which level the coronal section view in Figure 1 is taken from. Would this represent an area containing the lymphatic layer or the NPLP?

Response: The coronal plane of section in Figure 1a is now specified in Extended Data Fig. 1b and Extended Data Fig. 2.

8. The authors mention in the limitations section of the discussion that the NPLP “could not be selectively ablated in mice in loss-of-function studies performed to determine the adaptability of other routes for CSF clearance.” Did the authors attempt these studies? It is unclear from this sentence.

Response: Yes, we now explain in this part of the Discussion (page 17, lines 490-491) that our comment was based on the vascular damage occurring on Prox1-GFP mice after intracisternal infusion of verteporfin (2 mM, SML0534, Merck) packed with home-made liposomes and activated with intra-nasopharyngeal optic fiber emitting 690 nm laser light (MLL-III-690, CNI laser). In addition, our comment also based on failure of local ablation of the NPLP in the nasopharynx by intranasal administration of tamoxifen to Prox1-Cre^{ERT2}-VEGFR3 flox/flox mice.

9. The authors have not mentioned the superficial cervical lymph nodes and the associated lymphatic network which has also been shown in several previous studies to drain CSF. Does this network receive any drainage from the NPLP?

Response: At 30 min after infusion into the cisterna magna, TMR-Dextran was detected in both superficial and deep cervical lymph nodes. However, we found no evidence that superficial cervical lymph nodes receive TMR-Dextran drainage from the NPLP. TMR-Dextran in CSF outflow through the NPLP-to deep cervical lymphatics drained exclusively to deep cervical lymph nodes. This feature is now mentioned in the Results (page 6, line 147-149, and Extended Data Fig. 5a-c).

Referee #2 (Remarks to the Author):

“The nasopharyngeal lymphatic plexus is a hub for cerebrospinal fluid drainage and is regressed with ageing” By Yoon et al

Recent studies have highlighted the importance of lymphatic cerebral spinal fluid (CSF) drainage in health and disease states. It is known that tracers injected into the CSF are found in deep cervical lymph nodes, indicating that CSF drains via lymphatic routes. However, the details of these drainage routes between the base of the brain and lymph nodes remains unclear. Here the authors describe the anatomical structure of nasopharyngeal lymphatic vasculature and its connections to collecting points around the base of the brain in adult (8-12 week old) mice. The authors note unusually shaped valves in some places in the nasopharyngeal lymphatic vasculature, and lack of smooth muscle marker SMA around these lymphatic vessels where they are closest to the brain. They confirm that tracers injected into cisterna magna or hippocampus appear in the nasopharyngeal lymphatic vasculature and deep cervical lymph nodes at various time points after injection and that medial lymphatic vessels contribute more to clearance of the observed tracers to the lymph nodes. Topical application of phenylephrine or NO donor sodium NONOate resulted in constriction or dilation of the nasopharyngeal lymphatic vessels, respectively, indicating functional control of CSF drainage. AAV delivery of VEGFC to the lymphatic vasculature resulted in increases in lymphatic density and increased drainage of CSF tracers. In aged mice, nasopharyngeal lymphatic vasculature was less dense, or “atrophied” as the authors called it, but was otherwise functional when compared to younger adult mice. Finally, scRNAseq transcriptomic analysis indicated that nasopharyngeal lymphatics in aged mice exhibited increased inflammatory and apoptotic markers compared to the adult mice.

Overall, through a set of technically challenging experiments and observations, this manuscript provides a careful and detailed description of nasopharyngeal lymphatics in the mouse CNS, including a description of changes with aging, and includes a useful transcriptomic dataset. Specific comments are below.

Response: We appreciate the reviewer’s positive overall comments and excellent suggestions for improving the manuscript.

Comments:

1. Since the authors use a method that significantly alters blood flow during the surgery to expose the nasopharyngeal lymphatics and lymph nodes, could this affect the CSF and tracer flow through the lymphatic vessels? Blood flow is also thought to contribute to CSF movements in the brain.

Response: This important question is addressed by describing the limitations (page 16-17, lines 484-489) in the Discussion section.

2. Could the observed differences in medial vs. lateral lymphatic CSF tracer drainage be due in some part to differences in CSF tracer concentrations where these vessels are most proximal to the brain, where their collection points would presumably be? See for example the tracer distribution in figure 2b.

Response: As the reviewer points out, the CSF tracer could have preferential access to medial deep cervical lymphatics over lateral deep cervical lymphatics. However, new studies, described in the revised Results (page 9, lines 255-261, and Extended Data Fig. 17), revealed that after infusion into CSF the amount of TMR-Dextran was 2.6-fold higher in the anterior and middle regions that drain through medial cervical lymphatics than in the posterior portion that drains through lateral cervical lymphatics in the skull base. Thus, CSF accumulated in the anterior and middle portions of the skull base; it then drained through perineural lymphatics connected to the nasopharyngeal lymphatic plexus, medial cervical lymphatics, and deep cervical lymph nodes, before entering the systemic blood circulation.

3. The authors need to better explain the advantage and/or reasoning of intrahippocampal tracer injections vs. inter cisterna magna injections for the experiments performed.

Response: As shown in original Extended Data Fig. 12 (now ED Fig. 18), rapid clearance of TMR-Dextran to deep cervical lymph nodes (dcLNs) after intracisternal infusion led to the age comparison experiment. However, the variability of values after intracisternal infusion prevented detection of small age differences. As we now describe in the Results (page 9, line 264-268), our experiments revealed that the longer observation period and slow, stable clearance of TMR-Dextran to dcLNs after intrahippocampal infusion provided a more consistent baseline. This approach was therefore used for the ageing studies.

4. The authors should comment on the possible causes/mechanism for detachment of lymphatic endothelial cells observed in the aged mice.

Response: We appreciate the reviewer raising this issue that deserves greater attention in the original manuscript. We addressed this issue by using quantitative immunofluorescence methods to determine whether β -amyloid, phosphorylated tau (pTau) and TUNEL+ apoptotic lymphatic endothelial cells (LECs) were more abundant in the NPLP of aged mice. The findings of greater accumulation of pTau and more apoptotic LECs in the NPLP of aged mice, without significant detection of β -amyloid, are now described in the revised Results (page 11, lines 316-319, and Extended Data 22a,b).

Minor:

5. In general, it is difficult to distinguish which datapoints are significantly different from each other. P-value information are given in the figure captions, but this does not indicate what conditions and data sets/points this reflects. Please include more information in the figures.

Response: The requested statistical information has been added to all figures.

6. Figure 1a: what is the significance of the dashed green lines in the lateral view?

Response: The dashed green lines were removed from Fig. 1a to prevent confusion and added to new Extended Data Fig.1, where the significance is described in the figure legend.

7. Figure 6a: Contrast adjustments are needed to better reflect the results. For instance, the LYVE-1 staining in the adult mouse top panel is very dim compared to the aged mice, making comparison difficult. Similarly, VEGFR3 staining is difficult to see in the top panels.

Response: We appreciate these comments on Fig.6a and now show LYVE-1 and VEGFR3 separately in Extended Data Fig. 20 for greater clarity.

In the lower panels, the magnification is too low to get a clear view of the features pointed out by the arrows. Higher magnification images would be helpful, as well as images that better illustrate the detached lymphatic endothelial cells and valve structures. As presented, it is not obvious what the arrowheads are pointing out.

Response: We agree that the panels of Fig.6a identified by the reviewer are too small and the abundant arrowheads in the second row of figures confused the interpretation. We solved this problem by enlarging these images and reducing the arrowheads to 2-3 per image (Extended Data Fig. 20a,b). We also orient all arrowheads in the same direction to simplify the interpretation.

8. Figure captions: Why is “ $p < 0.05$ ” listed in the same sentences that indicate no significant differences?

Response: As described under Minor point 5, we checked that statistical information was included in all figure legends and added any missing information, corrected errors, and eliminated inconsistencies.

9. Please provide statistical information for figure 9d.

Response: The missing statistical information was added to Fig.9d.

10. Extended data figure 7c, magnified view: It is not clear that the fluorospheres are inside the lymphatic vessels. It looks like the spheres are non-specifically distributed in the region shown.

Response: We are pleased to provide a more convincing image and legend for Extended Data Fig. 7 (now Extended Data Fig. 10). After conducting new experiments in Prox1-GFP mice, we identified convincing lymphatic connections in tissues sectioned in the sagittal plane in conjunction with LYVE-1 labeling and intracisternal injection of FluoSpheres.

Although some movement of the tracer during processing was unavoidable, we minimized the scattering of FluoSpheres during the preparation of sagittal sections and now provide these new findings and descriptions in the Results (page 7, lines 189-195, and Extended Data Fig. 10).

11. Extended data Figure 9: A diagram with orientation information for the views presented (particularly in panel b) would be helpful.

Response: We agree that the complex anatomy shown in Fig.9 could be difficult to interpret and have added the requested diagram (now Extended Data Fig. 11) to orient the reader to the major structures shown.

12. Extended data figure 10b: The images presented for the SSS could be improved. There is little mCherry labeling in both mVEGFC and control conditions, and it looks like more in the mCherry controls.

Response: Yes, little mCherry labeling was expected around the superior sagittal sinus (SSS) in either group (new Extended Data Fig. 13b) because most of the AAV was cleared by 3 weeks after administration. Previous images were replaced by new ones (Extended Data Fig. 13b) that show low but identifiable and comparable mCherry expressions from the AAV9 constructs.

Referee #3 (Remarks to the Author):

Yoon et al. have produced a spectacular piece of work exploring the nasopharyngeal lymphatic plexus, a still concealed part of the lymphatic system associated with the central nervous system (CNS). The nasopharyngeal lymphatic plexus lays between the skull base and the palatine bone, collecting the cerebrospinal fluid (CSF) outflow from the skull and draining it towards the cervical lymph nodes which are the main brain-draining lymph nodes.

Here, the authors show that the nasopharyngeal lymphatic plexus is the main extra-cranial route for CSF outflow. The nasopharyngeal lymphatic plexus collects the lymph from multiple meningeal lymphatic circuits and connects with the medial lymphatic vessels that provide a main entry to CSF into the cervical lymph nodes. Considering that the clearance and immune surveillance of CNS tissues require an efficient drainage of macromolecules and immune cells transported by the CSF and that both processes are altered in many neurological disorders, new knowledge about the anatomy and functional properties of the CNS lymphatic circuit has strong potential regarding the treatment of neurological diseases and cognitive aging. The present work aligns well this objective with two main contributions. First, it demonstrates that CSF drainage can be modulated by targeting the extra-cranial lymphatic system which is impaired but still functional in aged mice. Second, it shows that adrenergic/nitric oxide (NO) signaling drugs regulate the contraction/dilatation of extra-cranial lymphatic collectors, respectively, and can be used to stimulate CSF lymphatic drainage, as an alternative strategy to the so far unique approach by VEGF-C (vascular endothelium growth factor -C) delivery.

The authors performed a technical tour de force by expert dissections allowing them to access and provide high resolution imaging of this hardly accessible lymphatic bed. They extensively characterized and beautifully illustrated the nasopharyngeal lymphatic plexus and its downstream cervical collectors by combining IHC techniques in adult and aged mice as well as in non-human primates (*Macaca*) with cervical lymphatic ligation experiments and fluorescent tracer injection into the CSF/CNS. Ex vivo studies on isolated cervical lymphatics brought additional important information on the elastic properties of these collectors. A difficult experiment of peri-mortem intra-vital imaging after intra-cisternal injection of TMR-Dextran allowed the authors to further demonstrate that CSF outflow via medial cervical lymphatics was reduced by alpha-adrenergic activation and increased by NO signaling in the lymphatic smooth muscle that surround medial lymphatics. A final single cell analysis of nasopharyngeal lymphatics in adult and aged is completing the study, showing that the adult plexus includes more Pt3+ capillaries with immune interplay than other organs and that pro-apoptotic gene expression as well as inflammation and type I interferon signaling globally increase with age in plexus endothelial cells.

Altogether the authors have generated a spectacular and convincing set of outstanding interest data. However, several questions should be further investigated.

Response: We appreciate the reviewer's favorable comments and thorough, insightful, and careful review. We are pleased the reviewer found the data supported by images interesting and convincing.

1. First, we would like to know whether deep cervical lymphatics can be directly targeted in living mice and if their pharmacological manipulation can delay glymphatic drainage and behavioral alterations during aging. This investigation requires to overcome technical challenges, as to access the nasopharyngeal plexus or cervical lymphatics in living animals, for example in Alzheimer's disease models such as 5XFAD mice but would indisputably raise the impact of the study by providing evidence that triggering extra-cranial lymphatics allows to delay cognitive aging.

Response: The reviewer raises the important issue of in vivo pharmacologic manipulation of CSF drainage that is highly relevant to the translational impact of our findings. We addressed this question in new experiments described in the Methods (page 21, lines 619-628) with findings reported in the Results (page 10, lines 292-299 and Extended Data Fig. 19a-c) and considered in the Discussion (page 16, lines 475-482).

We are delighted to report that we found an increase of the CSF outflow after stimulation of deep cervical lymphatics after topical application of a narrow concentration range of phenylephrine or sodium nitroprusside. We believe that this finding provides proof-of-concept for future studies in mouse models of Alzheimer's disease that take advantage of innovative pharmacological approaches to determine whether augmented pumping of deep cervical lymphatics can delay the decline in cognitive function.

2. Another investigation that I would suggest relates to the inflammatory phenotype detected by scRNA-Seq in medial lymphatics in aging mice and would test if the adult phenotype could be rescued by antagonistic drugs against anti-type I IFN signaling (sifalimumab or anifrolumab) and if this treatment may eventually improve CSF drainage.

Response: We appreciate this suggestion and have performed new experiments using anti-type I IFN signaling antibody in aged mice. These studies are described in the Methods (page 21-22, lines 630-639), and the findings are reported in the Results (page 14, lines 407-419, and Extended Data Fig. 27) and considered in the Discussion. Although we found no significant improvement of CSF drainage after 6-week blockade of type I IFN signaling, we acknowledge that further work is needed to determine whether different treatment regimens have beneficial effects on CSF drainage.

3. From the 2D images shown in extended data Fig 7c, it is hard to be convinced that meningeal and olfactory epithelium lymphatics form a continuous network. I suggest the authors to perform additional LYVE1 and Ptx3 labeling on cribriform plate sections (coronal, sagittal and horizontal) from Prox1-GFP mice and reconstruct the 3D-lymphatic network in this region. Why are there no FluoSpheres detected in the olfactory epithelium on the left panel? On the right panel, most of FluoSpheres are located outside Prox1 vessels and

appear to rather follow nerve branches through cribriform pores. From personal experience, I learned how much misleading can be 2D-scans through the irregular surfaces of the cribriform plate and would suggest the author to be cautious on any statement regarding the connection between these circuits.

Response: Yes, we agree that connections between the subarachnoid space and lymphatics in the cribriform plate and more distally are challenging to identify. We have followed the reviewer's recommendation in a new experiment to obtain more detailed information about these connections. After several trials with Prox1-GFP mice, we could only identify convincing lymphatic connections in 2D images of sagittal tissue sections in conjunction with LYVE-1 labeling and intracisternal injection of FluoSpheres. Although some movement during sectioning was unavoidable, we minimized the scattering of FluoSpheres during the preparation of sagittal sections. We now provide these new findings and descriptions in the Results (page 7, lines 189-195, Extended Data Fig. 10 and Supplementary Video 6).

The quality of their manuscript may also be improved by addressing some minor points outlined below.

Minor points:

1. Fig 1a. The anterior lymphatic layer is hard to visualize, as well as the LYVE1 pattern on the ventral view and the VEGFR3 pattern. There is enough space available to show each marker in a separated panel. There is also no legend for the dotted green line indicating the nasopharyngeal cavities.

Response: Thank you for this suggestion. We now present the multicolor images without green lines in Fig. 1a and show the same images with dotted green lines outlining the nasopharyngeal plexus along with separate single-color images showing LYVE1 and VEGFR3 in Extended Data Fig. 1a. The dotted green lines are described in the legend for Extended Data Fig. 1a.

2. Fig1b. A three-dimensional representation of the inverted saddle model should replace the present diagram.

Response: We are pleased to add as revised Fig. 1b a new 3D drawing of the inverted saddle shape of the nasopharyngeal lymphatic plexus (NPLP) and posterior nasal plexus.

3. Fig 3g. The present image of beta3tubulin labeling reiterates the message of Fig 3f and could be replaced by pictures of Tyrosine hydroxylase (TH) and choline acetyltransferase (ChAT) labeling that would inform on the adrenergic/cholinergic innervation of cervical lymphatics.

Response: This helpful comment led to new immunofluorescence studies which revealed that nearly all β 3-tubulin⁺ nerve fibers on these lymphatics are tyrosine hydroxylase⁺

adrenergic axons. No nerve fibers stained for vesicular acetylcholine transporter (VACHT) in cholinergic axons. These new data were added to the Results and figures (page 9, lines 234-238 and Extended Data Fig. 15a,b).

4. Figs 2, 3, 4 and 5. In all experiments, the authors intentionally injected 10kDa Dextran to monitor CSF drainage. Would they use higher molecular weight dextran (60-70kDa) and later time points for imaging, they could also track tracer-carrier phagocytes within extra-cranial lymphatics. This could be performed and associated with anti-CD45 or CD11b/c labeling of myeloid carrier cells, without quantification, after intra-CNS injection to demonstrate the traffic of brain tracer-carrier phagocytes within extra-cranial lymphatics close to cervical lymph nodes and how much time it takes to be detected.

Response: These are excellent recommendations. In new experiments, we injected higher molecular weight TMR-Dextran (70kDa), monitored CSF drainage, and obtained the same results as with 10kDa TMR-Dextran. Alternatively, we injected fluorescence-labeled ovalbumin and stained CD206+ macrophages in the nasopharyngeal lymphatic plexus. Again, we obtained the same finding as with 10kDa TMR-Dextran. All these findings are presented in the revised Results (page 6, lines 153-156 and Extended Data Fig. 6a-d). It is important to track tracer-carrier phagocytes that are derived from the CSF space and transport through extracranial lymphatics to cervical lymph nodes as a future, separate study. However, this would require careful consideration (e.g. phagocytic activity of the perivascular macrophages, migration properties of the phagocytes into the lymphatics, and so on).

5. Fig 7d, f, g. Indicate statistical values with either numbers or stars on the figures. Also simplify the corresponding legend.

Response: We added the statistical P values and simplified the legends in the revised manuscript.

6. Fig 9. Ptx3 expression by nasopharyngeal lymphatics should be assessed by IHC.

Response: We stained nasopharyngeal lymphatics for Ptx3 immunoreactivity according to methods of Taija Makinen's lab (Petkova *et al.*, (JEM, 2023) and present the findings in the revised Results (page 13, lines 391-393, and Extended Data Fig. 26a,b).

7. Extended data Fig 1. A 3D-reconstruction of the nasopharyngeal plexus could be generated from the set of dorsal-ventral-lateral views collected by the authors. The quality of generated data is calling for such an integrated representation.

Response: Good idea! We are pleased to provide Supplementary Video 1 as a 3D-reconstruction of dorsal-ventral-lateral views of the nasopharyngeal lymphatic plexus.

8. Extended data Fig 8 and corresponding text (p 7) deserve discussion with respect to previous studies describing basal and ventral connections between meningeal and extra-cranial lymphatic circuits (PMID: 35776089) as well as on the main importance of nasopharyngeal route for CSF lymphatic drainage (PMID: 18972423, PMID: 34905509, PMID: 35776089 Fig. 5 F and Fig. S3, E and F).

Response: We agree this background is important and now discuss the complementary findings of PMID: 35776089 (Jacob et al., JEM 2022) in the Discussion (page 16, line 455) and refer to the studies of PMID: 34905509 (Decker et al, JCI Insight, 2022) in the Results (page 11, line 304) and Discussion (page 17, line 495). Although the studies of PMID: 18972423 (Pan *et al*, Head and Neck, 2009) provided a detailed description of nasal lymphatics in human cadaver samples, it is uncertain which of these lymphatics are associated with CSF outflow. Therefore, we concluded that reference to Pan *et al*. was not warranted in this context.

9. Text lines 376-378. Please quote previous studies on spinal lymphatic drainage of CSF (PMID: 31455602, PMID: 31597914).

Response: Thank you for these references. PMID 31455602 (Ma et al. JEM 2019) is now cited in the revised Discussion (page 16, line 459) and PMID 31597914 (Jacob et al. Nature Commun 2019) is also cited (page 16, lines 459).

10. In the M&M section, p.20 line 592 the authors mention the use of light sheet fluorescence microscopy (LSFM) but there is no corresponding images or data. Please delete or provide LSFM data.

Response: We now specify the images obtained by LSFM in the respective Figure legends.

Referee #4 (Remarks to the Author):

This is a meticulous and elegant study filling an important knowledge gap in our understanding of the pathways of cerebrospinal fluid drainage to the deep cervical lymph nodes. Defining these anatomical routes is currently a matter of lively and controversial debate in the field. The present study from the laboratories of Dr. Davis and Dr. Koh significantly advances the field by identifying the nasopharyngeal lymphatic plexus as a major “hub” for CSF outflow from lymphatics from the cribriform plate but also additional intracranial regions towards the deep cervical lymph nodes. The study provides an in-depth analysis of the cellular and molecular makeup of the lymphatics of the nasopharyngeal lymphatic plexus and how they change with age. Furthermore, the authors explored if the regressing nasopharyngeal lymphatic plexus could be pharmacologically targeted for potential therapeutic enhancement of CSF drainage to deep cervical lymph nodes in neurological disorders.

This is an original study with exceptional relevance to our understanding of afferent pathways of CNS immunity. The authors have employed state-of-the-art methodology that is illustrated in each figure where necessary in detail and has referred to the shortcomings of studying CSF drainage pathways in anesthetized mice.

Response: We appreciate the reviewer's favorable comments and thorough, insightful, and careful review.

I have not identified major issues and have only some minor comments the authors may wish to consider.

1. Figure 8 d and e compare the tracer accumulation in deep cervical lymph nodes of young and aged mice. It would be informative to understand if the entire architecture of these aged lymph nodes has changed such that tracers would fail to accumulate.

Response: To address this issue we performed new experiments to compare lymph node size and lymphatic endothelial cell distribution in lymph nodes of adult and aged mice. New studies described on pages 13, lines 370-375, and Extended Data Fig. 24a-c revealed no significant change in either feature of lymph nodes in aged mice. These findings argue against structural changes in lymph nodes in aged mice as responsible for the reduction in tracer accumulation in deep cervical lymph nodes.

2. CSF drainage has in addition been observed to occur towards the superficial cervical or mandibular lymph nodes. It would be instructive if the authors did observe tracer drainage only to the deep cervical lymph nodes.

Response: At 30 min after infusion into the cisterna magna, TMR-Dextran was detected in both superficial and deep cervical lymph nodes. However, we found no evidence that superficial cervical lymph nodes receive TMR-Dextran drainage from the NPLP. TMR-Dextran drainage through the NPLP-to deep cervical lymphatics drained exclusively to deep

cervical lymph nodes. This feature is now mentioned in the Results (page 6, line 147-149, and Extended Data Fig. 5a-c).

3. The authors refer to the lateral afferent dc lymph node lymphatics to those connecting to the basolateral lymphatics through the jugular foramen. How do they envisage CSF drainage to dc lymph nodes from the dorsal dural lymphatics?

Response: We have addressed this comment by adding images from new histologic analyses of tissues removed after inserting a catheter into the jugular foramen to document the exit route and support the claim in the Results (page 9, lines 240-242 and Extended Data Fig. 16a-f). We revised the drawing of Extended Data Fig. 8c (now Extended Data Fig. 11) to show the connections of dorsal and basolateral dural lymphatics that pass through the jugular foramen and join lateral deep cervical lymphatics en route to deep cervical lymph nodes. Abbreviations in the Figure legend have been minimized for clarity.

4. In the discussion the authors highlight that modification of dural lymphatics may by enhancing immune cell migration from the CNS to the deep cervical lymph nodes ameliorate experimental autoimmune encephalomyelitis (EAE) citing a study from the Kipnis laboratory. However recent studies by the laboratories of Kari Alitalo (doi: 10.1172/JCI130308) and Odoardi/Flügel (doi: 10.1038/s41593-022-01108-3) have questioned these findings. As EAE generally serves as a model for multiple sclerosis the authors may wish to rephrase this statement.

Response: Good point! We revised the section dealing with EAE to reflect the recent findings mentioned (page 18, lines 519-523).

5. For the interested readers it would further be instructive if the authors included discussion of the recent findings by the Proulx laboratory on CSF outflow pathways at the cribriform plate (doi: 10.1016/j.ebiom.2023.104558) in the context of their findings.

Response: Yes, we now consider the recent findings by Spera *et al.* in the Introduction and Results sections (page 3, lines 82-84 and page 7, line 192).

6. In several figures arrowheads referred to are missing, and scale bars should be double checked. Figure 1 a – lateral view includes dashed lines not mentioned etc.

Response: Thank you for catching these errors. Missing arrowheads were added, all scale bars were double-checked, and dashed lines in Fig. 1a were removed.

This email has been sent through the Springer Nature Manuscript Tracking System NY-610A-SN&MTS

Reviewer Reports on the First Revision:

Referees' comments:

Referee #1 (Remarks to the Author):

The authors have done a superb job addressing my concerns. The additional experiments have added significant value to the paper and I would have no reservations now recommending this manuscript for publication.

I would recommend the authors consider making the additional minor changes to the text if accepted for publication:

1. Introduction, Line 99: Remove the word “fluorescent” from this sentence. This makes it sound like this plexus has endogenous fluorescence.
2. Results, Line 126: Add the word “collecting” before lymphatics. The authors are describing the typical features of these types of vessels in the periphery.
3. Results, Line 147-149: the authors may want to mention that the superficial lymph nodes had similar fluorescence levels and, if they are not downstream of the NPLP, that this is suggesting the presence of additional CSF outflow pathways not shown in this study.
4. Results Lines 182-187: The authors should correct the statement “a large lymphatic vessels”. Besides the typo, what do they mean by the unclear term “large”? I would suggest the authors be extra careful in their final version with the interpretation within this section of Extended Data 8a-d and Supplementary Video 4. Are the authors absolutely sure that this is a continuous lymphatic network? The reconstruction from the video appears to indicate a gap in the network in the cavernous sinus. These lymphatic “islands” appear to be common in the dural lymphatic network. Are the authors absolutely sure that the beads are being drained by these lymphatics? Do they have orthogonal projections that can show the beads with the lumen?
5. Results, Lines 290-292: This statement should be placed below the data describing the enhancement in the lymph nodes (Lines 292-294). Otherwise, the authors should change “CSF outflow” to “lymphatic diameter” or something to this effect.

Referee #2 (Remarks to the Author):

Overall the authors addressed well our comments. In response to our original comment 1: “Since the authors use a method that significantly alters blood flow during the surgery to expose the nasopharyngeal lymphatics and lymph nodes, could this affect the CSF and tracer flow through the lymphatic vessels? Blood flow is also thought to contribute to CSF movements in the brain”, the authors describe some limitations in Discussion section, but they do not really address blood flow changes. Please update Discussion by including a brief discussion about how alterations of blood flow during the surgery could possibly affect the nasopharyngeal lymphatics and the CSF and tracer flow through the lymphatic vessels.

Referee #3 (Remarks to the Author):

I am delighted to see the meticulous work performed to address in detail all my comments. I am convinced of the continuity of the lymphatic circuit from the olfactory meninges through the cribriform plate and the final data showing the therapeutic potential of a direct drug application on deep cervical lymphatics are very convincing.

The authors have performed a spectacular work that should become a key reference in the field of research on the lymphatic system of the central nervous system.

Author Rebuttals to First Revision:

Responses to Referees' comments

Reference: Nature 2023-03-05202A; The nasopharyngeal lymphatic plexus is a hub for cerebrospinal fluid drainage and is regressed with ageing

Referee #1 (Remarks to the Author):

The authors have done a superb job addressing my concerns. The additional experiments have added significant value to the paper and I would have no reservations now recommending this manuscript for publication. I would recommend the authors consider making the additional minor changes to the text if accepted for publication:

1. Introduction, Line 99: Remove the word “fluorescent” from this sentence. This makes it sound like this plexus has endogenous fluorescence.

Response: We agree and deleted “fluorescent”.

2. Results, Line 126: Add the word “collecting” before lymphatics. The authors are describing the typical features of these types of vessels in the periphery.

Response: We understand the reviewer’s comment and have revised the sentence to correct the problem. The revised sentence (now lines 126-128) reads: “Because of these features, lymphatics of the plexus were unlike individual initial lymphatics in most peripheral organs or large collecting lymphatics that have semilunar valves, long lymphangions, and smooth muscle coverage.”

3. Results, Line 147-149: the authors may want to mention that the superficial lymph nodes had similar fluorescence levels and, if they are not downstream of the NPLP, that this is suggesting the presence of additional CSF outflow pathways not shown in this study.

Response: This is an excellent suggestion. The sentence (now lines 150-152) was included to confirm that the CSF outflow that reached superficial lymph nodes did not flow through the NPLP.

4. Results Lines 182-187: The authors should correct the statement “a large lymphatic vessels”. Besides the typo, what do they mean by the unclear term “large”?

Response: We corrected the error and deleted “large”. The statement (now lines 187) now reads, “...identification of lymphatics near the pituitary...”

I would suggest the authors be extra careful in their final version with the interpretation within this section of Extended Data 8a-d and Supplementary Video 4. Are the authors absolutely sure that this is a continuous lymphatic network? The reconstruction from the video appears to indicate a gap in the network in the cavernous sinus. These lymphatic “islands” appear to be common in the dural lymphatic network. Are the authors absolutely

sure that the beads are being drained by these lymphatics? Do they have orthogonal projections that can show the beads with the lumen?

Response: We agree and carefully re-examined all raw images used for Extended Data 8a-d and Supplementary Video 4 and confirmed that lymphatics in these images were continuous and not islands of lymphatic endothelial cells. We also confirmed that the fluorescent beads were within the lumen of lymphatics. Changes were made in the text (now lines 190-192) to describe these additional findings.

5. Results, Lines 290-292: This statement should be placed below the data describing the enhancement in the lymph nodes (Lines 292-294). Otherwise, the authors should change “CSF outflow” to “lymphatic diameter” or something to this effect.

Response: We agree. To address this issue we changed this sentence (now lines 296-298) to read: “These findings provide evidence that the diameter of medial cervical lymphatics increased with NO-mediated dilatation and decreased with α 1-adrenergic-mediated constriction.”

Referee #2 (Remarks to the Author):

Overall the authors addressed well our comments. In response to our original comment 1: “Since the authors use a method that significantly alters blood flow during the surgery to expose the nasopharyngeal lymphatics and lymph nodes, could this affect the CSF and tracer flow through the lymphatic vessels? Blood flow is also thought to contribute to CSF movements in the brain”, the authors describe some limitations in Discussion section, but they do not really address blood flow changes. Please update Discussion by including a brief discussion about how alterations of blood flow during the surgery could possibly affect the nasopharyngeal lymphatics and the CSF and tracer flow through the lymphatic vessels.

Response: Yes, we agree with the reviewer that did not adequately address the contribution of cerebral blood flow and vascular pulsations to CSF circulation. Therefore, we added the following sentence to the Discussion (now lines 491-494): “Another potential limiting factor is the effect of surgical interventions on CSF drainage through lymphatics to the nasopharyngeal plexus, because cerebral blood flow and vascular pulsation contribute to CSF circulation, which in turn influences CSF outflow (Wang Y. et al. Cerebrovascular activity is a major factor in the cerebrospinal fluid flow dynamics. Neuroimage. 2022, 258:119362. PMID: 35688316).

Referee #3 (Remarks to the Author):

I am delighted to see the meticulous work performed to address in detail all my comments. I am convinced of the continuity of the lymphatic circuit from the olfactory meninges through the cribriform plate and the final data showing the therapeutic potential of a direct drug application on deep cervical lymphatics are very convincing. The authors have preformed a

spectacular work that should become a key reference in the field of research on the lymphatic system of the central nervous system.

Response: Thank you.